# SMI-Editor: Edit-based SMILES Language Model with Fragment-level Supervision

**Kangjie Zheng**[1,2,3] **Siyue Liang**[1,2,3] **Junwei Yang**[1,2,3] **Bin Feng**[4,1,2,3] **Zequn Liu**[1,2,3]
**Wei Ju**[*5] **Zhiping Xiao**[*6] **Ming Zhang**[*1,2,3]

[1] School of Computer Science, Peking University.
[2] National Key Laboratory for Multimedia Information Processing, Peking University.
[3] Peking University-Anker Embodied AI Lab, Peking University.
[4] International Digital Economy Academy (IDEA), Shenzhen, China.
[5] College of Computer Science, Sichuan University, Chengdu, China.
[6] Computer Science and Engineering Department, University of Washington, U.S.A.
`juwei@scu.edu.cn, patxiao@uw.edu, mzhang_cs@pku.edu.cn`

## Abstract

SMILES, a crucial textual representation of molecular structures, has garnered significant attention as a foundation for pre-trained language models (LMs). However, most existing pre-trained SMILES LMs focus solely on the single-token level supervision during pre-training, failing to fully leverage the substructural information of molecules. This limitation makes the pre-training task overly simplistic, preventing the models from capturing richer molecular semantic information. Moreover, during pre-training, these SMILES LMs only process corrupted SMILES inputs, never encountering any valid SMILES, which leads to a train-inference mismatch. To address these challenges, we propose SMI-Editor, a novel edit-based pre-trained SMILES LM. SMI-Editor disrupts substructures within a molecule at random and feeds the resulting SMILES back into the model, which then attempts to restore the original SMILES through an editing process. This approach not only introduces fragment-level training signals, but also enables the use of valid SMILES as inputs, allowing the model to learn how to reconstruct complete molecules from these incomplete structures. As a result, the model demonstrates improved scalability and an enhanced ability to capture fragment-level molecular information. Experimental results show that SMI-Editor achieves state-of-the-art performance across multiple downstream molecular tasks, and even outperforming several 3D molecular representation models. [1]

## 1 Introduction

With the widespread application of AI in molecular-related tasks, enhancing the modeling of SMILES data has become a key research focus. The textual nature of SMILES data makes it possible for us to leverage past experiences from text modeling to address challenges in SMILES representation. Additionally, the knowledge extracted from SMILES data often aligns well with textual knowledge. A large number of studies have attempted to design SMILES language models (LMs) to explore the knowledge inherent in SMILES sequences (Wang et al., 2019a; Chithrananda et al., 2020; Bagal et al., 2021; Ross et al., 2022). Significant efforts have also been made to align learned knowledge from SMILES with textual knowledge Edwards et al. (2022); Pei et al. (2023); Liu et al. (2023b), aiming at boosting the performance of downstream applications such as property prediction and molecular design. A critical issue in these model designs lies in how to efficiently mine molecular-related knowledge from SMILES data. This paper address this issue by presenting a SMILES LM with enhanced modeling capabilities.

---

[*] Corresponding Authors.
[1] Code is released at https://github.com/zhengkangjie/smi-editor

Current SMILES LMs often adopt strategies used in natural language processing, such as predicting missing tokens in corrupted SMILES sequences (e.g., Masked Language Modeling (MLM) and Causal Language Modeling (CLM)). However, this approach introduce several challenges:

(i) Unlike natural language, where individual tokens represent independent semantic units like subwords, words or phrases, SMILES data tokens correspond to single atoms, chemical bonds, or special symbols. Molecular functionality, however, is more closely tied to specific substructures (e.g., functional groups). The SMILES LMs focusing only on individual tokens in the SMILES context may fail to capture the semantic information of these substructures.

(ii) Predicting a single missing token in a given SMILES context is often trivial, leading to rapid saturation of model capacity during training. This limits the model's ability to acquire more nuanced molecular knowledge, and affects its scalability and its effectiveness in generalizing to diverse molecular data.

(iii) Since these models are trained on corrupted SMILES sequences containing special symbols like [MASK] – which do not appear in real-world SMILES data – they face challenges in modeling the semantic content of complete SMILES sequences.

To address these challenges, we propose an edit-based SMILES language model with fragment-level supervision:

(i) To enable the model to learn richer substructure-related molecular information, we introduce fragment-level supervision. By randomly removing substructures from molecules and train the model to recover the missing information, we encourage it to acquire more comprehensive fragment-level semantic knowledge.

(ii) We design an edit-based pre-training objective, allowing the model to process valid SMILES sequences and to restore missing substructures through an editing process.

In summary, our contributions are threefold:

- We analyze the behavior of current SMILES masked language models (MLMs) during pre-training phase and downstream tasks, identifying their rapid saturation problem in pre-training and their limited ability to capture the molecular substructure information. Previous studies lack a systematic exploration of these issues.

- We introduce the first edit-based pre-trained LM for SMILES, which transforms valid SMILES sequences into structurally related variants. This approach resolves the train-inference mismatch issue in existing SMILES LMs. The integration of fragment-level supervision further enhances our model's ability to learn richer semantic information from SMILES and improves its overall performance.

- Extensive experiments demonstrate that SMI-EDITOR achieves state-of-the-art performance across multiple molecular property prediction tasks, surpassing several 3D molecular representation models. Ablation studies confirm the effectiveness and scalability of SMI-EDITOR.

## 2 UNDERSTANDING THE BEHAVIOR OF MLM

The Masked Language Model (MLM) is a widely used approach for textual information modeling and has been extensively applied to SMILES representation learning (Wang et al., 2019a; Chithrananda et al., 2020; Ross et al., 2022). During the training process, tokens in a SMILES sequence – representing single atoms, chemical bonds, or special symbols – are randomly masked with a fixed masked ratio of 15%. The model is then trained to predict these masked tokens accurately, as is illustrated in Figure 1. To evaluate the effectiveness and capabilities of MLMs for SMILES data, we conducted a comprehensive series of experiments.

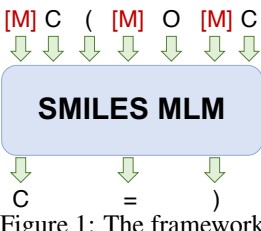

Figure 1: The framework of MLM for SMILES.

### 2.1 RAPID SATURATION PROBLEM

**Rapid Saturation During Pre-training.** To investigate whether the SMILES MLM model experiences rapid saturation during training and how this issue impacts its scalability, we trained MLMs

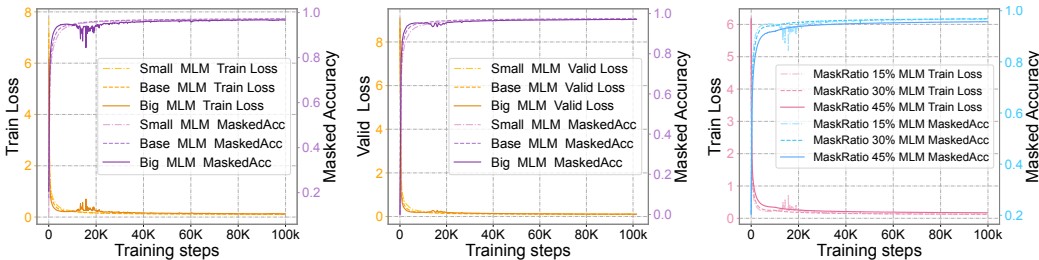

(a) Training Curves: Model Scales  (b) Validation Curves: Model Scales  (c) Training Curves: Mask Ratios

Figure 2: **Rapid Saturation Problem.** We train SMILES MLMs of various sizes and masking ratios using the dataset from Zhou et al. (2023). Figure (a) displays the training loss and masking prediction accuracy of different-sized models, showing a rapid decrease in loss and an increase in accuracy at the start of the training. Figure (b) presents similar trends for the validation set. Figure (c) illustrates the training loss and accuracy for models with different masking ratios, showing similar patterns.

of various scales and compared their training curves (Details of models with different scale can be found in Appendix D). As shown in Figure 2a, the training loss decrease rapidly, with the mask-prediction accuracy on the training set exceeding $90\%$ within the first 5,000 steps. By approximately 10,000 steps, all models achieved over $95\%$ accuracy on the training set. A similar rapid saturation phenomenon is observed on the validation set, as shown in Figure 2b, where the validation loss drops sharply after training begins and mask-prediction accuracy rises quickly. The rapid saturation phenomenon is consistent across all models, regardless of scale, including the small model with just 6.7M parameters. These findings indicate that the MLM pre-training task is overly simplistic, enabling even very small models to converge quickly, which restricts the models' capacity and scalability for more complex tasks. We also test the performance of MLMs of different sizes and different training steps on downstream tasks (detailed results are in Appendix E). The downstream-task results further confirm the limited scalability of SMILE MLMs.

**Different Mask Ratio Cannot Alleviate Rapid Saturation.**    One possible reason for the rapid saturation observed in MLM pre-training is the low masking ratio, with only $15\%$ of tokens masked during training. This might provide insufficient training information and make token prediction too easy, leading to rapid saturation. To explore this hypothesis, we trained large-scale MLMs with different mask ratios (i.e., $15\%$, $30\%$, $45\%$). The training curves, shown in Figure 2c, reveal that MLMs with different mask ratios all exhibit a sharp decline in training loss at the beginning of the training process, converging quickly to a very low level. Even with a mask ratio of $45\%$, the training loss drops rapidly, and by 10K steps, the mask-prediction accuracy already exceeds $92\%$. These findings indicate that increasing the mask ratio does not mitigate the rapid saturation problem or enhance the scalability of SMILES MLMs. Instead, the results suggest that the rapid saturation stems not from the masking ratio but from the inherent simplicity of the MLM pre-training task, which fails to provide sufficient complexity or information for modeling SMILES data effectively.

## 2.2    CHALLENGES IN MODELING SUBSTRUCTURE SEMANTICS

To evaluate the ability of MLMs to learn molecular substructure semantics, such as functional groups, we design experiments to analyze whether the models can accurately capture functional group information that is closely related to molecular properties. For this purpose, we use two molecular property prediction datasets, ESOL and FreeSolv (Wu et al., 2017), both of which are highly relevant to molecular hydrophilicity. Specifically, the ESOL dataset provides information on the water solubility of molecules, while the FreeSolv dataset focuses on the hydrogen free energy, both of which are strongly associated with hydrophilic groups within the molecules.

In our approach, we first fine-tune the MLM on these datasets using linear probing. Next, we traverse the SMILES of all molecules in the datasets and remove the hydrophilic groups (e.g., $-$OH, $-$COOH, $-$NH$_2$, etc.) identified in each molecule. We then compare the predicted molecular property values before and after the removal. As a control, we also randomly delete atoms from these molecules and analyzed the predicted changes in molecular properties.

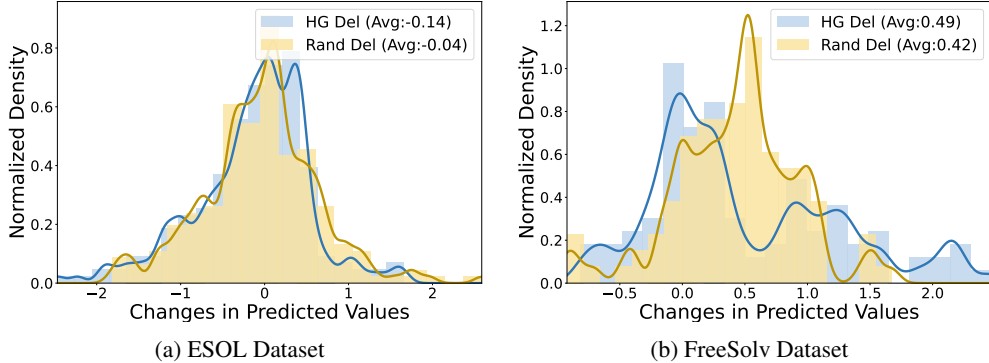

Figure 3: **Substructure Semantics Modeling.** We compared two molecular perturbation methods—removing hydrophilic groups and randomly deleting atoms—and their effects on the model's predictions of hydrophilicity and related properties. Figure (a) presents the impact of these perturbations on model predictions in the ESOL dataset, including the distribution of prediction changes. The average prediction change is similar for both methods (-0.14 vs. -0.04) and shows similar distributions. Figure (b) shows the effects on the FreeSolv dataset, also with similar average prediction change.

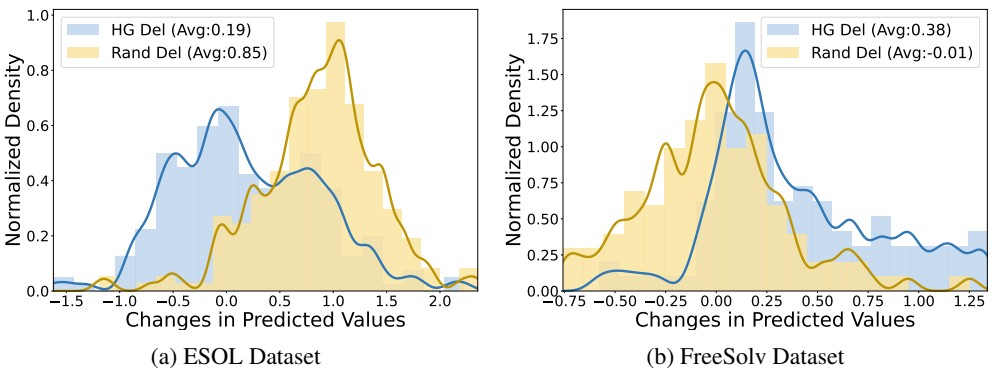

Figure 4: **Substructure Semantics Modeling from SMI-EDITOR.** We compared the effects of two molecular perturbation methods on the SMI-EDITOR's predictions of hydrophilicity and related properties. Figure (a) and Figure (b) show that the impact of deleting hydrophilic groups (HG Del) and randomly deleting atoms (Rand Del) on the model's predictions differs significantly, both in the average change in prediction values and their distributions.

As shown in Figure 3, the changes in predicted values after deleting hydrophilic groups (HG Del) are similar to those from random deletions (Rand Del) in both the ESOL and FreeSolv datasets. This indicates that the model struggles to differentiate between the impacts of removing hydrophilic groups and that of random atoms on molecular properties. These results further suggest that the MLM fails to effectively capture the semantic information of key molecular substructures in SMILES.

## 3 EDIT-BASED PRE-TRAINING FRAMEWORK

To overcome the limitations of MLM-based SMILES LMs, we propose a novel SMILES LM that employs an edit-based training objective. To enhance the model's ability to capture substructure semantics, we introduce fragment-level supervision during pre-training. This involves randomly discarding substructures and requiring the model to predict the missing components. This method enables the model to effectively learn substructure semantics. MLMs only operate on corrupted SMILES contexts with unreal [MASK] symbols, leading to inconsistencies between training and testing. In contrast, our approach inputs complete and valid SMILES sequences into the model, seeking to reconstruct the missing substructures through an editing-based approach. Moreover, the editing framework offers greater flexibility compared to MLM, as it imposes no specific restrictions

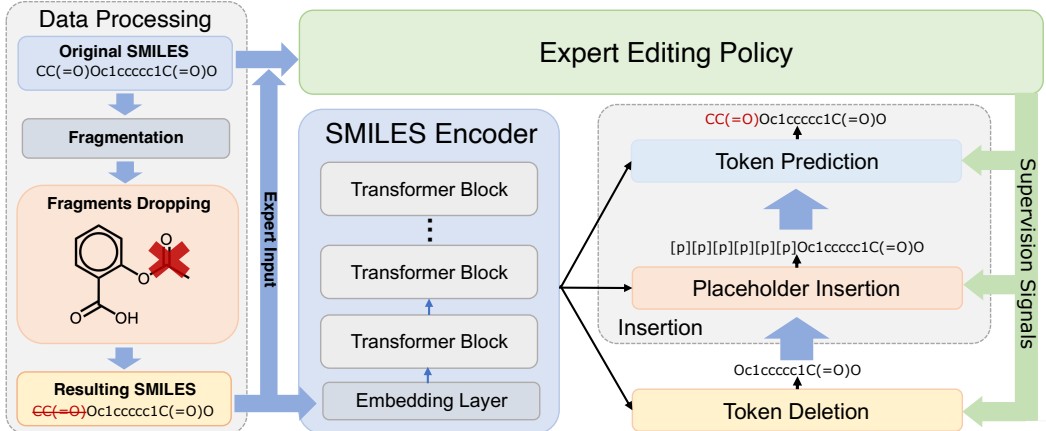

Figure 5: **Overall Framework of SMI-EDITOR**. The framework includes a data processing module, a SMILES encoder, and an edit-based pre-training process. In data processing, some fragments of the input molecule are randomly removed, and the resulting SMILES is fed into the model. The pre-training goal for the model is to edit the corrupted SMILES to recover the original SMILES. To enable this, we add three different heads for token deletion, placeholder insertion, and token prediction to the SMILES encoder (see Appendix B for details). An expert provides training signals for these operations to help the model learn how to recover the original SMILES through editing.

on input forms. This flexibility allows us to create more versatile model inputs by removing certain substructures from a molecule, converting it back to SMILES, and then feeding it into the model. In this section, we provide an in-depth discussion of SMI-EDITOR, focusing on its model design and its pre-training framework.

## 3.1 SMILES ENCODER WITH EDITING OPERATIONS

In the edit-based pre-training process, the model learn to model the editing operations. Specifically, given a SMILES sequence with missing substructures, the model needs to accurately predict the editing operations required to reconstruct these missing substructures. To achieve this, we designed a SMILES encoder capable of supporting editing-operation modeling.

**Model Architecture.**   The core architecture of the model is a Transformer Encoder composed of multiple stacked Transformer blocks. Each transformer block contains a multi-head self-attention layer and a feed-forward layer (Vaswani et al., 2017). The SMILES representations extracted by the Transformer Encoder are then passed to the Editing Operations Head, which predicts the necessary editing operations. Similar to existing Edit-based LMs (Gu et al., 2019), the model handles two types of editing operations: deletion and insertion. Specifically, it completes missing parts of a SMILES sequence by removing redundant parts and inserting missing substructures. To support these tasks, the architecture includes dedicated prediction heads for both editing operations, enabling the model to effectively model editing tasks.

**Deletion Operations Head.**   For a given input token, deletion is a binary classification problem – deciding whether to delete or retain the token. Let $\boldsymbol{x}_i^E$ denote the representation of the $i$-th input token extracted by the Encoder. The probability of deleting the $i$-th token, denoted as $\pi_\theta^{\text{del}}(i)$, can be expressed as:

$$\pi_\theta^{\text{del}}(i) = \text{Softmax}(\boldsymbol{W}_d^T \boldsymbol{x}_i^E),$$

where we have the weight matrix $\boldsymbol{W}_d$ of size $H \times 2$, and $H$ is the hidden size.

**Insertion Operations Heads.**   Modeling insertion operations is more complex than deletion. Inspired by LevT Gu et al. (2019), the insertion operation is split into two steps. In the first step, the model predicts the positions and number of tokens to insert into the original input sequence.

Placeholders [P] are then inserted at these positions, representing the locations where new tokens will be added. Later on, in the second step, the model predicts the actual tokens for each placeholder [P].

Given the ordinary length of SMILES sequences, we constrain the model to insert at most 255 tokens at a time. Therefore, this step is framed as a 256-class classification problem for each token position. The probability of inserting tokens at the $i$-th position, denoted as $\pi_\theta^{\text{ins}}(i)$, is defined as:

$$\pi_\theta^{\text{ins}}(i) = \text{Softmax}(\boldsymbol{W}_{\text{in}}^T \boldsymbol{x}_i^E),$$

where $\boldsymbol{W}_{\text{in}}$ is a matrix of size $H \times 256$.

In the second step, for each placeholder symbol [P], the model predicts the token to insert. This step resembles MLM pre-training tasks, where the model predicts the probability distribution over the vocabulary for the token at each placeholder. The probability distribution for the token at the $i$-th position, denoted as $\pi_\theta^{\text{tok}}(i)$, is defined as:

$$\pi_\theta^{\text{tok}}(i) = \text{Softmax}(\boldsymbol{W}_{\text{tok}}^T \boldsymbol{x}_i^E),$$

where $\boldsymbol{W}_{\text{tok}}$ is a matrix of size $H \times \text{vob}$, where $\text{vob}$ represents the size of the vocabulary.

## 3.2 Edit-based Pre-training with Fragment-level Supervision

After constructing the SMILES encoder with editing operations, the next crucial step is to develop an edit-based pre-training framework that incorporates fragment-level self-supervised training signals. Unlike traditional MLMs, our edit-based model transforms a valid SMILES input into the target SMILES by modeling the necessary editing operations. This process begins by fragmenting the input molecule using rule-based molecule fragmentation, and then randomly selected some fragments to discard from the original molecule. The resulting corrupted molecule is reconstructed into a SMILES representation and fed into the SMILES Encoder.

**Molecule Fragmentation and Fragments Dropping.** To provide the model with fragment-level training signals, we first split the input molecule $M$ into multiple fragments $\{f_1, f_2, ...\}$. A popular method for molecular fragmentation is the BRICS algorithm (Degen et al., 2008), which divides a molecule into fragments based on predefined rules such as functional groups. However, BRICS often generates relatively large fragments, and removing these fragments can overly disrupt the molecule, potentially eliminating critical structures like rings. To address this, we adopt a modified fragmentation approach on RMCF (Wang et al., 2022a), which further splits connections between rings and side chains on top of BRICS, resulting in smaller molecular fragments. After cutting the molecule, we randomly select and discard a subset of the fragments with a predefined probability. The remaining fragments are then reassembled into a corrupted molecule $\hat{M}$. The model is tasked with reconstructing the original molecule's SMILES $M$ from the given corrupted SMILES $\hat{M}$ through an edit-based approach.

**Edit-based Training Objective with Dual Loss.** The core pre-training task of the SMI-EDITOR model is to take the SMILES of a corrupted molecule $\hat{M}$ as input and attempt to reconstruct the SMILES of the original molecule $M$ through an editing process, by accurately modeling deletion and insertion operations. However, traditional edit-based models like LevT only provide insufficient training signals for deletion. It only teaches the model to remove incorrect tokens it inserted. This constraint hinders the model's ability to effectively learn how to delete the incorrect parts in the input SMILES. To overcome this problem, we introduce a dual deletion loss, which supervises the model to correctly remove erroneous tokens from the corrupted molecule input $\hat{M}$.

We adopt the imitation-learning method from LevT, which supervises the model to minimize the Levenshtein distance between the input and target output. The dual deletion objective is defined as:

$$\mathcal{L}_\theta^{\text{DualDel}} = - \sum_{\substack{y_i \in \hat{M} \\ d_i^* \in d^*}} \log \pi_\theta^{\text{del}}(d_i^* | i, \hat{M}),$$

where $d^*$ is the optimal deletion action determined by an expert to minimize the Levenshtein distance to the target output $y^*$ which is the SMILES of molecule $M$. This is formulated as $d^* = \text{argmin}_d \mathcal{D}(y^*, \varepsilon(\hat{M}, d))$, where $\mathcal{D}$ is the Levenshtein distance, $\pi_\theta^{\text{del}}$ is the Deletion Classifier, and $\varepsilon$ represents the environment in LevT's Markov Decision Process. $\varepsilon(\hat{M}, d)$ applies the deletion action $d$ to the initial input SMILES $\hat{M}$, removing selected tokens.

In addition to the dual deletion loss, we retain LevT's original training objective $\mathcal{L}_\theta^{\text{LevT}}$ (see Appendix B for details). This supervises both deletion and insertion actions by minimizing the Levenshtein distance between the input $\hat{M}$ and target $M$. The final training objective for SMI-EDITOR is a combination of these two losses:

$$\mathcal{L}_\theta = \mathcal{L}_\theta^{\text{DualDel}} + \mathcal{L}_\theta^{\text{LevT}}.$$

Through this edit-based pre-training process, we equip the SMI-EDITOR model with fragment-level training signals, enabling it to reconstruct the SMILES of a corrupted molecule $\hat{M}$ into the SMILES of the original molecule $M$ via fragment editing.

## 4 EXPERIMENTS

In this section, we first evaluate the performance of SMI-EDITOR on molecular property prediction tasks and compare it with baseline models (see Section 4.2). The results show that SMI-EDITOR outperforms both the MLM and 3D molecular models, achieving state-of-the-art performance. To further validate the model design and pre-training framework, we conduct ablation studies on training signals and editing operations (see Section 4.3). In addition, analytical experiments confirm that SMI-EDITOR has a stronger ability to capture the semantics of molecular substructures compared to MLMs. Analysis of the training curves also demonstrates that SMI-EDITOR mitigates the issue of rapid saturation and enhances the training stability (Section 4.4). We also provide details on the training data, baseline models, and implementation used in the experiments (see Section 4.1).

### 4.1 EXPERIMENT SETTINGS

**Datasets.** For pre-training, we use the large-scale molecular dataset provided by Zhou et al. (2023), which includes SMILES information for 19 million molecules. For fine-tuning, we employ the widely-recognized MoleculeNet benchmark (Wu et al., 2018) (see Appendix I for more details). We follow the same data split as used by Zhou et al. (2023) and tokenize SMILES sequences with the regular expression from Schwaller et al. (2018).

**Baselines.** We evaluate our approach against various supervised learning and pre-training baselines, including both SMILES-based and 3D molecular pre-training models. The supervised methods include D-MPNN (Yang et al., 2019) and AttentiveFP (Xiong et al., 2019), both of which are based on graph neural networks (GNNs). For 2D and 3D molecular pre-training, we consider baseline methods: N-gram (Liu et al., 2019a), GROVER (Rong et al., 2020), GraphMVP (Liu et al., 2021), MolCLR (Wang et al., 2022b), InfoGraph (Sun et al., 2019), Mole-BERT (Xia et al., 2023), 3D InfoMax (Stärk et al., 2022), and MoleculeSDE (Liu et al., 2023a). For a fair comparison, we train a SMILES model based on MLM pre-training, referred to as SMI-MLM, using the same training data, model architecture, and training hyperparameters as SMI-EDITOR.

**Implementation Details.** We use a Transformer block with a hidden size of 768 and 12 attention heads, comprising 12 layers in the SMILES encoder, which contains a total of 86.3 million trainable parameters. During pre-training, the fragment drop ratio is set to 0.15. For downstream tasks, we use the same fine-tuning dataset established by Uni-Mol. (cf. Appendix G for more details about hyper-parameter configuration.)

### 4.2 RESULTS ON MOLECULAR PROPERTY CLASSIFICATION TASKS

**Tasks Details.** We evaluate SMI-EDITOR on the MoleculeNet (Wu et al., 2017) benchmark and compare its performance with baseline models. We evaluate SMI-EDITOR on 7 widely-used

Table 1: The overall results on 7 molecule property classification datasets. We report ROC-AUC score (higher is better) under scaffold splitting. The best results are **bold**. The second-best results are underlined. For more detailed information about the dataset, please refer to Table 7.

| Datasets
# Molecules | BACE↑
1531 | BBBP↑
2039 | Tox21↑
7831 | SIDER↑
1427 | MUV↑
93087 | ClinTox↑
1478 | ToxCast↑
8575 | Mean↑
- |
|---|---|---|---|---|---|---|---|---|
| D-MPNN | 80.9 | 71.0 | 75.9 | 57.0 | 78.6 | 90.6 | 65.5 | 74.2 |
| Attentive FP | 78.4 | 64.3 | 76.1 | 60.6 | 76.6 | 84.7 | 63.7 | 72.1 |
| N-Gram$_{RF}$ | 77.9 | 69.7 | 74.3 | **66.8** | 76.9 | 77.5 | - | - |
| GROVER | **82.6** | 70.0 | 74.3 | 64.8 | 62.5 | 81.2 | 65.4 | 71.5 |
| GraphMVP | 81.2 | 72.4 | 75.9 | 63.9 | 77.7 | 79.1 | 63.1 | 73.3 |
| InfoGraph | 77.8 | 67.5 | 73.2 | 59.9 | 74.1 | 76.5 | 63.7 | 70.4 |
| MolCLR | 82.4 | 72.2 | 75.0 | 58.9 | 79.4 | 91.2 | **69.2** | 75.5 |
| Mole-BERT | 80.8 | 71.9 | 76.8 | 62.8 | 78.6 | 78.9 | 64.3 | 73.4 |
| 3D InfoMax | 79.7 | 69.1 | 74.5 | 60.6 | 74.4 | 79.9 | 64.4 | 71.8 |
| MoleculeSDE | 80.4 | 73.2 | 76.5 | 59.6 | 79.9 | 86.6 | 65.2 | 74.5 |
| SMI-MLM | 77.8 | 68.6 | 75.1 | 61.2 | 75.1 | 89.8 | 64.9 | 73.2 |
| SMI-EDITOR | 80.3 | **77.4** | **77.1** | 63.0 | **80.2** | 98.9 | 67.4 | **77.8** |

molecular property prediction tasks ( see Appendix H for details). For all the seven tasks, we take the normalized SMILES information as model input and fine-tuning on each task separately. We use ROC-AUC as the evaluation metric, and the results are summarized in Table 1.

**Results.**   SMI-EDITOR achieves state-of-the-art (SOTA) performance on 4 out of 7 tasks, and closely matches the SOTA models on the remaining tasks. Compared to the MLM model SMI-MLM, whose training settings, training data, and evaluation processes for downstream tasks are all identical to ours, SMI-EDITOR demonstrates superior performance across all seven tasks. These results validate the effectiveness of our pre-training framework. Additionally, SMI-EDITOR outperforms several molecular representation learning models that utilize 3D information, indicating that SMILES has the potential of revealing important and rich semantic information related to the spatial molecular properties, and that SMI-EDITOR effectively captures such information. SMI-EDITOR also demonstrated the strongest average performance across all 7 tasks, indicating that it outperforms other baseline models in these prediction tasks overall.

## 4.3   ABLATION STUDIES

### 4.3.1   ABLATION STUDIES ON FRAGMENT-LEVEL SUPERVISION

Table 2: **Ablation Studies on Fragment-level Supervision.** Fragment-level supervision provide more informative and useful training signals than atom-level supervision and are crucial for helping the model learn multi-level semantic information in molecules.

| Method | BACE↑ | BBBP↑ | Tox21↑ | SIDER↑ | ToxCast↑ | Mean↑ |
|---|---|---|---|---|---|---|
| SMI-EDITOR-AtomsDropping | 80.0 | 73.4 | 76.5 | 59.2 | 66.6 | 71.1 |
| SMI-EDITOR-AtomsMasking | **80.4** | 73.2 | 75.0 | 58.3 | 64.6 | 70.3 |
| SMI-MLM | 77.8 | 68.6 | 75.1 | 61.2 | 64.9 | 69.5 |
| SMI-EDITOR | 80.3 | **77.4** | **77.1** | **63.0** | **67.4** | **73.0** |

**Experimental Settings.**   To explore the impact of fragment-level supervision signals versus the atom-level signals on model performance, we additionally train SMI-EDITOR's variants using two different pre-training strategies. The first model, SMI-EDITOR-AtomsDropping, replaces the fragment dropping process in pre-training with random atom dropping. After discarding certain atoms, we input the modified SMILES into the model, asking it to restore the original SMILES through an editing approach. The second model, SMI-EDITOR-AtomsMasking, uses random token masking similar to MLM, where selected tokens are replaced with `[MASK]`, and the model is tasked

with restoring the original SMILES via editing. The performance of these models is presented in Table 2.

**Results Analysis.** The results show a significant decline in performance when fragment-dropping is replaced with random-atom-dropping (SMI-EDITOR-AtomsDropping vs. SMI-EDITOR), indicating that the fragment-level supervision signal enables the model to learn more important and nuanced semantic information. Furthermore, when random atom dropping is replaced with random token masking, performance decreases again (SMI-EDITOR-AtomsMasking vs. SMI-EDITOR-AtomsDropping). This suggests that while both random-atom-masking and random-atom-dropping introduce atom-level training signals, the introduction of the unrealistic special symbol [MASA] through token masking adversely affects model performance. Compared to these two models, SMI-MLM exhibits even poorer performance, demonstrating that this editing training approach effectively helps the model learn richer semantic knowledge.

### 4.3.2 ABLATION STUDIES ON EDITING OPERATIONS

Table 3: **Ablation Studies on Editing Operations.** The placeholder insertion process, which is absent in MLMs, enables the model to learn richer and more diverse semantic information.

| Method | BACE↑ | BBBP↑ | Tox21↑ | SIDER↑ | ToxCast↑ | Mean↑ |
|---|---|---|---|---|---|---|
| w/o PlhIns | 76.1 | 69.7 | 76.9 | 55.5 | 66.2 | 68.9 |
| w/o TokPred | 79.8 | 69.2 | 75.4 | 57.4 | 65.9 | 69.5 |
| w/o TokDel | 79.0 | 73.5 | **77.3** | 61.9 | 64.9 | 71.3 |
| w/o DualDel | 78.4 | 70.1 | 76.4 | 59.5 | 64.4 | 69.8 |
| SMI-EDITOR | **80.3** | **77.4** | 77.1 | **63.0** | **67.4** | **73.0** |

**Experimental Settings.** To investigate the impact of different training signals from the editing operations in the SMI-EDITOR model on its performance, we train four additional variants of the SMI-EDITOR model. These models are obtained by removing the training signals for placeholder insertion, token prediction, token deletion, and dual token deletion (setting the training loss weight to zero), corresponding to the three editing operations in the original LevT model and the dual deletion loss. The detailed results are presented in Table 3.

**Results Analysis: Why SMI-EDITOR is Better than SMI-MLM.** The results indicate that the ablation of any of these four editing operations leads to a significant decline in model performance. Notably, removing the placeholder insertion operation results in the largest performance drop. This operation primarily models the position of missing fragments within the SMILES, highlighting the importance of teaching the model to predict the locations of these fragments for capturing critical semantic information and improving performance. In contrast, the MLM model attempts to predict masked tokens based on their given positions, which simplifies the pre-training task and limits the model's exposure to important semantic information, ultimately affecting its performance. Moreover, SMI-EDITOR provides supervision signals for each token in the sequence, but the MLM model only provides supervision signals for [MASK] tokens, limiting the semantic richness of the model.

**Results Analysis: Dual Deletion Loss is More Useful.** The ablation of the dual deletion operation also causes a significant decline in model performance, with a more pronounced drop than when token deletion is removed. This indicates that the dual-deletion-objective of SMI-EDITOR provides more useful and richer training signals than token-deletion objective in LevT.

### 4.4 ANALYTICAL EXPERIMENTS

**SMI-EDITOR Understands Substructure Semantics.** Similar to the analysis in Section 2.2, we tested SMI-EDITOR's response to two different molecular perturbation methods on the ESOL and FreeSolv datasets. As shown in Figure 4, compared it to the results in Figure 3, SMI-EDITOR exhibits distinct prediction changes for the two perturbation methods on both the ESOL and FreeSolv datasets. This indicates that SMI-EDITOR can clearly differentiate between the impact of removing

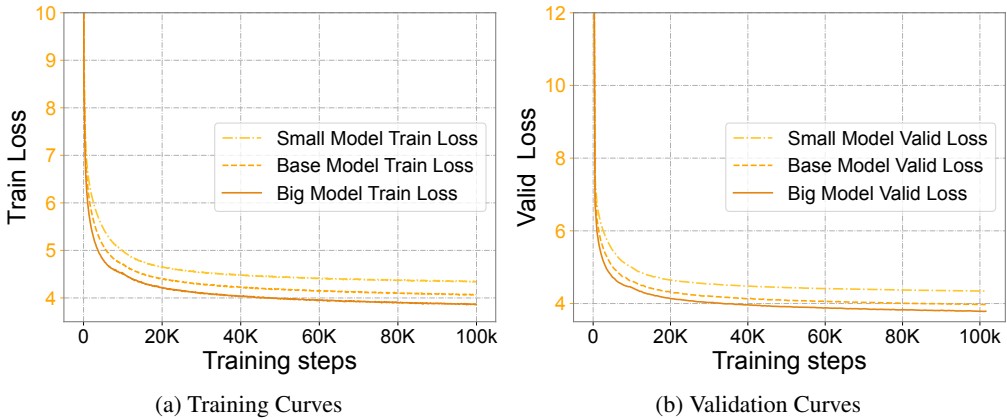

(a) Training Curves          (b) Validation Curves

Figure 6: The training and validation loss curves of different-sized SMI-EDITOR models. The loss curves consistently show a stable downward trend throughout the training process, and the model loss gradually decreases as the model size increases.

hydrophilic groups and randomly-selected atoms on molecular properties, demonstrating that it models the semantics of key molecular substructures more effectively than the MLM.

**SMI-EDITOR Enhances Training Stability and Model Scalability.** We train SMI-EDITOR of different sizes and compare their training curve variations. As shown in Figure 6, the losses of the SMI-EDITOR models consistently exhibit a more pronounced downward trend throughout the training process compared to the MLMs (see Figure 2a), further alleviating the rapid saturation problem. Additionally, unlike the MLM, the training loss of the SMI-EDITOR shows more distinct differences across sizes. As the model size increases, the loss steadily decreases, with the larger model (Big Model) converging more stably than the MLM, indicating better scalability for SMI-EDITOR. We also analyze the training and validation loss curves for the three types of editing operations in SMI-EDITOR, confirming the model's scalability during pre-training (see Appendix C for details). Additionally, we evaluate the performance of SMI-EDITOR models of different sizes on downstream tasks, demonstrating that SMI-EDITOR exhibits better scalability and stability compared to the MLM (i.e., SMI-MLM) (see Appendix F for details).

## 5    CONCLUSIONS

In this paper, we analyze the behavior and shortcomings of masked language models (MLMs) on SMILES data. By examining the training curves, we demonstrate that training MLMs on SMILES data encounters rapid saturation issues. Further analytical experiments reveal that MLMs struggle to effectively capture the semantics of important molecular substructures. To address these issues, we propose the edit-based pre-training molecular representation learning framework SMI-EDITOR, which is specifically designed to capture substructure semantics by learning how to recover the missing fragments through edit operations. Extensive experiments on molecular property prediction tasks validate the effectiveness of SMI-EDITOR, and ablation studies confirm the advantages of its design over traditional MLMs in modeling molecular substructure semantics and training stability.

## 6    ACKNOWLEDGEMENTS

We would like to thank Kuangqi Zhou from Bytedance for his insightful discussions on the project. We also thank other members from Dlib in Peking University for their valuable feedback given during the internal discussions. This paper is partially supported by National Natural Science Foundation of China with Grant Number 62276002.

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

## A    RELATED WORKS

Simplified Molecular Input Line Entry System (SMILES) has emerged as a cornerstone sequential representation for molecular data, making it a focal point in molecular representation learning. A large range of Pre-trained SMILES LMs (Wang et al., 2019a; Chithrananda et al., 2020; Ross et al., 2022) have been developed to address challenges in SMILES-based molecular modeling, demonstrating their effectiveness across diverse downstream tasks (Bagal et al., 2021; Tong et al., 2021; Feng et al., 2024; Yang et al., 2024a). These models typically leverage techniques like MLM and autoregressive pre-training to capture the complex syntax and semantics embedded in SMILES sequences. In parallel, edit-based generative models, another important approach to sequence modeling, have gained prominence in broader sequence modeling tasks, including machine translation, text summarization, and grammatical error correction. This section provides a comprehensive overview of these two prominent approaches. We first introduce representative work in Pre-trained SMILES LMs, highlighting their architectures and applications. Then, we discuss edit-based LMs, detailing their methodologies and their relevance to sequence modeling tasks.

### A.1    PRE-TRAINED SMILES LANGUAGE MODEL

Similar to text, SMILES sequences encode information in a sequential format. Early pre-trained SMILES LMs adopted methods techniques originally designed for text modeling. Wang et al. (2019a) introduced SMILES-BERT, inspired by the BERT model (Devlin et al., 2018), employing its MLM objective, which has been widely applied in various data representation learning tasks (Lin et al., 2023; Zheng et al., 2024; Yang et al., 2024b; Hayes et al., 2025). Their results demonstrated significant improvements in molecular property prediction tasks. Likewise, Chithrananda et al. (2020) developed ChemBERTa, leveraging the advanced RoBERTa architecture (Liu et al., 2019b) to enhance SMILES semantics modeling through MLM pre-training. Further advancing this line of research, Ross et al. (2022) proposed Molformer, trained on a larger dataset with MLM training objective, showing that SMILES LMs can effectively capture both molecular properties and structural patterns. These contributions have established MLM-based methods as a dominant approach in SMILES representation learning. In parallel, generative pre-training approaches have also gained traction in this domain. MolGPT (Bagal et al., 2021) employs an autoregressive mechanism, while Tong et al. (2021) applied generative models to drug design tasks. More recently, Liu et al. (2023b) introduced MolXPT, unifying SMILES and textual data using a generative pre-training strategy. Overall, these works highlight the growing importance of pre-trained SMILES LMs, particularly those based on the MLM objective.

### A.2    EDIT-BASED LANGUAGE MODEL

Edit-based sequence generation offers a faster and more flexible alternative to traditional autoregressive methods. Malmi et al. (2019) introduced the LASERTAGGER model, which uses tags (i.e., keep, delete, add) to edit sequences. The Felix model (Mallinson et al., 2020) extended this idea by integrating a pointer-based mechanism with an MLM backbone to handle insertion and deletion tasks more efficiently. Recognizing that edit operations from an input sequence to a target output can be diverse and difficult to compute directly, Gu et al. (2019) developed the Levenshtein Transformer (LevT) model, which calculates the minimum levenshtein distance between the input and target sequences. By leveraging this metric, LevT generates an optimal sequence of edit operations, improving performance on tasks such as machine translation and post-editing. LevT was further applied to lexically constrained translation tasks with notable success (Susanto et al., 2020). To resolve inconsistencies between training and inference, Zheng et al. (2023) introduced a dual training objective, improving the performance of edit-based models in applications such as summarization and grammatical error correction. With their efficiency and adaptability, edit-based LMs have emerged as a highly promising paradigm for sequence modeling across diverse tasks.

## B    DETAILS OF LEVENSHTEIN TRANSFORMER

The Levenshtein Transformer (LevT) is a non-autoregressive edit-based generation model that employs a three-step editing process: token deletion, placeholder insertion, and token prediction. LevT is trained using imitation-learning with a dual policy: (i) learning to insert tokens by predicting

those that have been randomly deleted from the complete sequences, and (ii) learning to delete tokens by identifying incorrect tokens generated by an insertion module. Below are more details about the training objective of LevT, denoted as $\mathcal{L}_\theta^{\mathrm{LevT}}$.

**Placeholder Insertion Loss.** In this step, the model needs to determine how many placeholders [P] should be inserted at specific positions in the original input, which will later be replaced by concrete tokens in subsequent steps. This operation is modeled as a classification task that predicts how many words need to be inserted after each token in the input sequence. For practical implementation, LevT limits the maximum number of words that can be inserted after each token to 255. Thus, this step essentially becomes a 256-class classification task at each token, predicting the number of words (0–255) to insert after each token. This process can be represented as follows:

$$\mathcal{L}_\theta^{\mathrm{ins}} = - \sum_{\substack{y_i \in Y_0 \\ p_i^* \in p^*}} \log \pi_\theta^{\mathrm{ins}}(p_i^*|i, Y_0),$$

where $p_i^*$ is the optimal placeholder insertion action found by the expert that minimizes the Levenshtein distance to the target output $y^*$ which is the SMILES of molecule $M$, and can be formalized as $p_i^* = \mathrm{argmin}_p \mathcal{D}(Y^*, \varepsilon(Y_0, p))$, $Y_0$ is the initial input of the model which is the SMILES of molecule $\hat{M}$, $\mathcal{D}$ is the Levenshtein distance measurement, $\pi_\theta^{\mathrm{del}}$ is LevT's Deletion Classifier, and $\varepsilon$ is the environment in the Markov Decision Process of LevT which receives editing actions and returns the modified sequence, and $\varepsilon(Y_0, p)$ means applies the insertion action $p$ to the initial input sequence $Y_0$ (e.g., insert some placeholders in $Y_0$). Details of $\varepsilon$ can be found in LevT's framework (Gu et al., 2019).

**Token Prediction Loss.** In this step, the task is to predict an exact token for each placeholder [P] in the sequence $Y_1 = \varepsilon(Y_0, p^*)$ that has had placeholders inserted. This process closely resembles that of MLM, as it essentially entails a classification problem where the number of classes is equal to the size of the vocabulary.

$$\mathcal{L}_\theta^{\mathrm{tok}} = - \sum_{\substack{y_i \in Y_1, t_i^* \in t^* \\ y_i = <[\mathrm{P}]>}} \log \pi_\theta^{\mathrm{tok}}(t_i^*|i, Y'),$$

where $t_i^*$ is the optimal insertion action found by the expert that minimizes the Levenshtein distance to the target output $Y^*$, $Y_1$ is the modified sequence by applying the optimal placeholder action $p^*$ to the input sequence $Y_0$, and these terms can be formalized as: $t_i^* = \mathrm{argmin}_t \mathcal{D}(y^*, \varepsilon(Y_1, t))$, $y_1 = \varepsilon(Y_0, p^*)$, $d^*$ or $p^* = \mathrm{argmin}_{d,p} \mathcal{D}(Y^*, \varepsilon(Y_0, \{d, p\}))$. $\pi_\theta^{\mathrm{tok}}$ is token classifier.

**Token Deletion Loss.** In the previous insertion step, the model may have inserted incorrect tokens. So in this step, it needs to predict which of the previously-inserted tokens are incorrect and should be deleted. Essentially, this step involves learning how to "correct" the errors made during the insertion phase. Specifically, the input to this step is the output from the insertion module, $Y_2 = \varepsilon(Y_1, t)$, where $t$ represents the actions predicted by the model in the token prediction step. Since the task is to decide whether each token in $Y_2$ should be deleted, this step is essentially a binary classification task on each token, represented as:

$$\mathcal{L}_\theta^{\mathrm{del}} = - \sum_{\substack{y_i \in Y_2 \\ d_i^* \in d^*}} \log \pi_\theta^{\mathrm{del}}(d_i^*|i, Y_2),$$

where $d_i^*$ is the optimal delete action found by the expert that minimizes the Levenshtein distance to the target output $Y^*$ which is the SMILES of molecule $M$, and can be formalized as $d_i^* = \mathrm{argmin}_d \mathcal{D}(y^*, \varepsilon(Y_2, d))$, $\pi_\theta^{\mathrm{del}}$ is LevT's deletion classifier.

**Total Loss.** Since the editing process of LevT consists of three steps—token deletion, placeholder insertion, and token prediction—the overall training objective of LevT is the sum of the training objectives for these three processes:

$$\mathcal{L}_\theta^{\mathrm{LevT}} = \mathcal{L}_\theta^{\mathrm{ins}} + \mathcal{L}_\theta^{\mathrm{tok}} + \mathcal{L}_\theta^{\mathrm{del}}.$$

In summary, unlike MLMs that provide training signals only for each [MASK] symbol in the input sequence, the LevT model offers training signals for every token in both the Placeholder Insertion and

Token Deletion steps. This design requires the model to determine whether each token in the input sequence should be deleted and whether new tokens should be inserted after each existing token, thus providing richer semantic information to the model.

## C    MORE TRAINING CURVES OF SMI-EDITOR

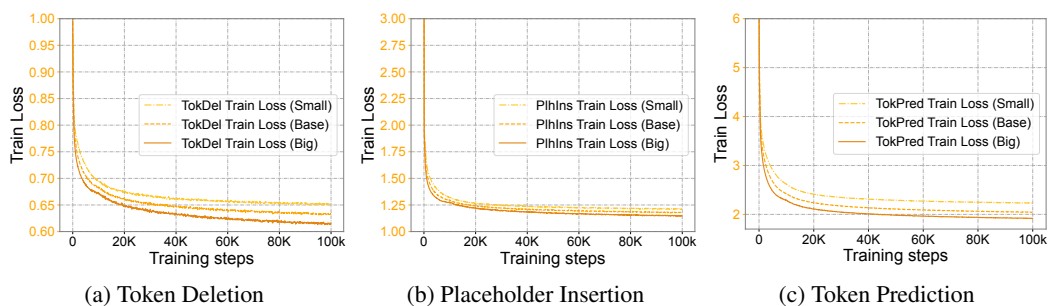

(a) Token Deletion          (b) Placeholder Insertion          (c) Token Prediction

Figure 7: **Training Loss Curves of Editing Operations.** We train SMI-EDITOR models of varying sizes and compare their loss curves during training for three different editing operations. As shown in the results, the loss for the token prediction process represented in Figure (c) is consistently the highest among the three type of losses, while the loss for token deletion is the lowest. Furthermore, as the model size increases, all three types of loss exhibit a stable downward trend.

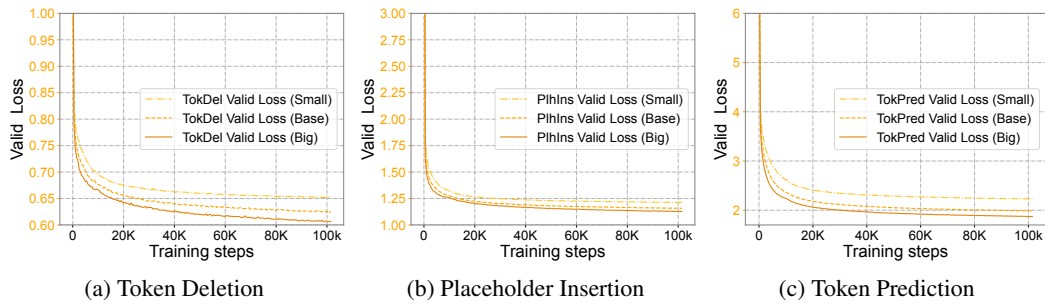

(a) Token Deletion          (b) Placeholder Insertion          (c) Token Prediction

Figure 8: **Validation Loss Curves of Editing Operations.** Similar to the training loss curves, the validation loss for the token prediction process shown in Figure (c) consistently remains the highest among the three types, while the loss for token deletion is the lowest. Additionally, as the model size increases, the validation loss for all three editing operations exhibits a stable downward trend.

We present the changes in training and validation loss curves for SMI-EDITOR models of varying sizes during training. As shown in Figure 7 and Figure 8, both training and validation losses for the three types of editing operations exhibit a stable downward trend as model scale increases. The loss from the token prediction process consistently constitutes the largest portion of the overall training loss. As is mentioned before, during the edit-based pre-training, the token prediction task is similar to that of MLM, as it involves predicting the real tokens corresponding to each placeholder token [P], aiming to restore the complete target SMILES. However, unlike the results in Figure 2, the token prediction loss in the SMI-EDITOR pre-training does not exhibit a rapid saturation phenomenon in the early stages of training. Even in later training phases, the token prediction loss continues to decline steadily. This highlights the advantage of incorporating fragment-level training signals; by removing substructures and requiring the model to predict them, rather than randomly masking tokens, we establish a training task with improved scalability.

## D    HYPER-PARAMETERS FOR MODELS OF VARYING SCALES

In Table 4, we present the specific training hyperparameters for the models of different sizes (i.e., Big, Base, Small) used in this study. Notably, despite the differences in training objectives between

MLM and SMI-EDITOR, all other model settings, including the training hyperparameters (as listed in Table 4) and training datasets, remain consistent to ensure a fair comparison of results.

Table 4: Hyper-parameters for pre-train models with different scales.

| Model | Max Tokens | Layers | Attn Heads | Embed Dim | FFN Dim | Dropout | Num of Paras |
|-------|-----------|--------|-----------|-----------|---------|---------|--------------|
| Big   | 64K       | 9      | 12        | 768       | 2048    | 0.1     | 50.5M        |
| Base  | 64K       | 6      | 8         | 512       | 2048    | 0.1     | 19.4M        |
| Small | 64K       | 3      | 8         | 512       | 1024    | 0       | 6.8M         |

# E  PERFORMANCE OF MLMS ON DOWNSTREAM TASK

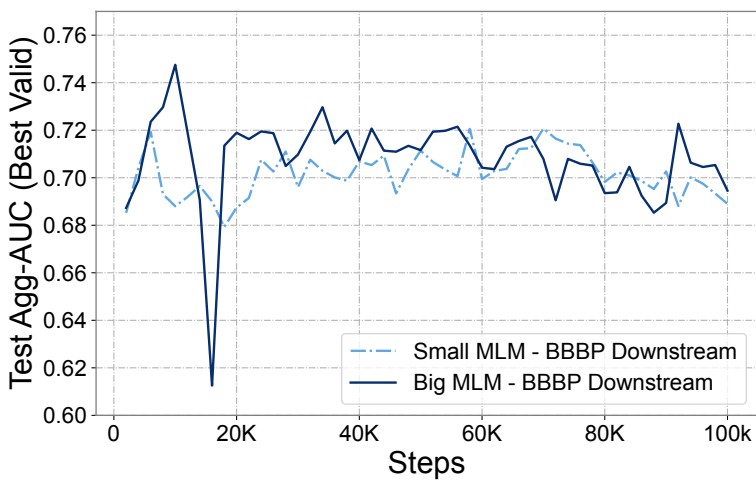

Figure 9: The performance of MLMs of different sizes and training steps on BBBP task.

We test the performance of MLMs (i.e., SMI-MLM) of different sizes and different numbers of training steps on the BBBP task. To reduce variability and ensure accuracy, we evaluate each checkpoint on the downstream tasks five times and take the mean results. As shown in Figure 9, increasing the model's scale does not consistently improve the model's performance on the downstream tasks. In many cases, small models even outperform larger ones. This indicates that the semantic information captured by larger MLMs does not necessarily translate into better downstream task performance. Instead, larger models exhibit greater variability in their performance compared to small ones. These findings highlight the limited scalability and stability of MLMs.

# F  PERFORMANCE OF SMI-EDITOR ON DOWNSTREAM TASK

We also test the performance of SMI-EDITOR of different sizes and training steps on the BBBP task, and the results are shown in Figure 10. Again, we evaluate each checkpoint on the downstream tasks five times and take the mean result to ensure accuracy. Compared to the performance of the MLM (Figure 9), the larger SMI-EDITOR model (i.e., Big Model) consistently outperforms the smaller models (i.e., Small Model). As the number of training steps increases, the performance gap between large and small models becomes more pronounced for SMI-EDITOR. In contrast, MLMs do not exhibit this trend, with models of different sizes achieving similar performance on downstream tasks. Moreover, larger MLMs show greater performance fluctuations compared to their smaller counterparts. Notably, the larger SMI-EDITOR model demonstrates superior performance stability, as it does not exhibit increased fluctuations compared to smaller SMI-EDITOR models. These results indicate that the SMI-EDITOR model offers better training stability and model scalability than the MLM.

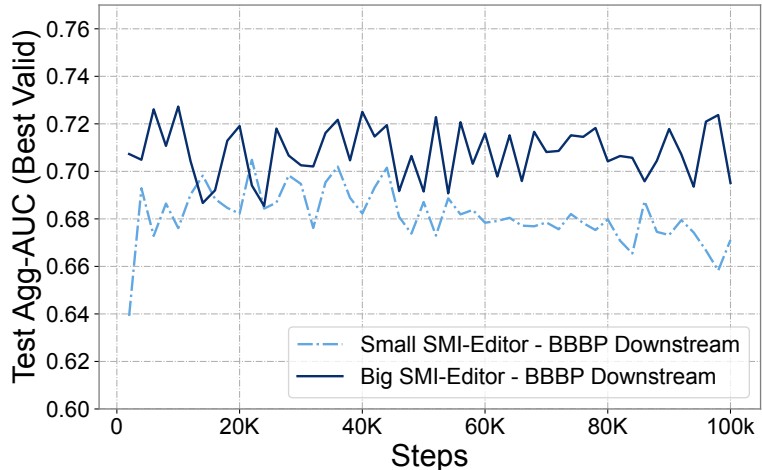

Figure 10: The performance of SMI-EDITOR of different sizes and training steps on BBBP task.

## G   HYPER-PARAMETER CONFIGURATION FOR PRE-TRAINING

We implement SMI-EDITOR using 12 stacked Transformer layers, each with 12 attention heads. The model dimension and feedforward dimension of each Transformer layer are 768 and 3,072, respectively. The total number of SMI-EDITOR's parameters achieves 86.3M. We use Adam (Kingma & Ba, 2014) optimizer and polynomial learning rate scheduler to train SMI-EDITOR, and we set the learning rate as $5e-4$ and warmup step as 10K. The total training step is 120K and each batch has 64k tokens at maximum. We implement the SMI-EDITOR model using the Fairseq library [2] and train SMI-EDITOR on four RTX3090 GPUs for about 1 day.

For more pre-training hyper-parameters, please refer to Table 5.

Table 5: SMI-EDITOR hyper-parameters for pre-training.

| Hyper-parameters | Value |
| --- | --- |
| Learning rate | 5e-4 |
| LR scheduler | polynomial_decay |
| Warmup updates | 10K |
| Max updates | 120K |
| Max tokens | 64k |
| FFN dropout | 0.1 |
| Attention dropout | 0.1 |
| Activation dropout | 0 |
| Num of layers | 12 |
| Num of attention heads | 12 |
| Encoder embedding dim | 768 |
| Encoder FFN dim | 3072 |
| Adam $(\beta_1, \beta_2)$ | (0.9,0.98) |
| Fragments Drop ratio | 0.15 |
| Vocabulary size | 369 |
| Activation function | GELU |
| Weight Decay | 0.0 |
| Clip Norm | 1.0 |

---

[2]https://fairseq.readthedocs.io/en/latest/

## H  HYPER-PARAMETER CONFIGURATION FOR FINE-TUNING

In different downstream task, we use different hyper-parameters. For detailed fine-tuning hyper-parameters, please refer to Table 6.

Table 6: SMI-EDITOR hyper-parameters for fine-tuning.

| Tasks | Epochs | Batch size | Learning rate | Warmup Ratio | Dropout | Pooler-dropout |
|---|---|---|---|---|---|---|
| BACE | 60 | 64 | 1e-4 | 0.06 | 0.1 | 0.2 |
| BBBP | 40 | 128 | 4e-4 | 0.06 | 0.1 | 0.1 |
| TOX21 | 80 | 128 | 1e-4 | 0.06 | 0.1 | 0.1 |
| SIDER | 100 | 32 | 5e-4 | 0.4 | 0.1 | 0 |
| MUV | 40 | 128 | 2e-5 | 0.2 | 0.1 | 0.1 |
| ClinTox | 100 | 256 | 5e-5 | 0.1 | 0.1 | 0.5 |
| ToxCast | 80 | 64 | 1e-4 | 0.06 | 0.1 | 0.1 |

## I  DETAILS OF FINE-TUNING DATASETS

We perform a comprehensive set of experiments on the MoleculeNet(Wu et al., 2018) benchmark, focusing on the molecular property prediction task. MoleculeNet has emerged as one of the most widely recognized and utilized benchmarks in the field of molecular property prediction, providing a standardized platform for evaluating machine learning models' performances on evaluating molecular properties. Its datasets encompass a broad range of molecular tasks, and address diverse and practical scientific problems such as drug discovery, toxicity prediction and so on.

In this section, we provide a detailed summary of the statistics and fundamental characteristics of the MoleculeNet datasets we use in Table 7. This table offers information about the dataset sizes, task types, and compositions, providing readers with essential background information to better understand the experimental setup and subsequent analysis.

Table 7: Summary information of the MoleculeNet benchmark datasets.

| Dataset | Tasks | Task type | Molecules (train/valid/test) | Describe |
|---|---|---|---|---|
| ESOL | 1 | Regression | 902/113/113 | Water solubility |
| FreeSolv | 1 | Regression | 513/64/64 | Hydrogen free energy |
| Lipo | 1 | Regression | 3,360/420/420 | Octanol/water distribution ratio, coefficient |
| BACE | 1 | Classification | 1,210/151/151 | Binding results of human BACE-1 inhibitors |
| BBBP | 1 | Classification | 1,631/204/204 | Blood-brain barrier penetration |
| ClinTox | 2 | Multi-label classification | 1,182/148/148 | Clinical trial toxicity and FDA approval status |
| Tox21 | 12 | Multi-label classification | 6,264/783/783 | Qualitative toxicity measurements |
| ToxCast | 617 | Multi-label classification | 6,860/858/858 | Toxicology data based on in vitro screening |
| SIDER | 27 | Multi-label classification | 1,141/143/143 | Adverse drug reactions to the 27 systemic organs |
| MUV | 17 | Multi-label classification | 74,469/9,309/9,309 | A subset of PubChem BioAssay |

## J  PERFORMANCE OF SMI-EDITOR ON DEEPCHEM DATA

We evaluated the performance of SMI-EDITOR (pre-trained on datasets provided by Ross et al. (2022)) on various downstream tasks of MoleculeNet benchmark using the data splits provided by DeepChem [3]. In our previous experiments, our results were based on a different data split, which made it less convincing to compare our model against others built on this dataset. Therefore, we re-tested SMI-EDITOR on DeepChem splits and included comparisons with more baseline models. Detailed results are presented in Table 8. As shown in Table 8, SMI-EDITOR achieves significant performance gains over baseline models, reaching state-of-the-art levels with noticeable mean performance improvements. Below is a detailed analysis of these results:

- SMI-EDITOR outperforms models trained with various paradigms: Measured by the mean performance, SMI-EDITOR surpasses molecular representation learning models like Mol-CLR and $DMP_{TF}$, which use contrastive pretraining, as well as models like ChemBerta and SMI-MLM, which use masked language modeling. It also outperforms autoregressive LMs like Galactica and graph-based models like MolCLR, MGSSL, and MoMu. These results highlight the potential of SMILES LMs.

- SMI-EDITOR achieves competitive performance with less training data: SMI-EDITOR outperforms $DMP_{TF}$, which is trained on over 100 million compounds, despite using only 19 million compounds for training. This demonstrates SMI-EDITOR's higher data efficiency, enabled by its ability to effectively leverage substructure information from SMILES sequences.

Table 8: Mean results on MoleculeNet datasets using DeepChem splits. ROC-AUC scores (higher is better) are reported for all tasks. The best results are **bolded**

| Method | BBBP↑ | Tox21↑ | ClinTox↑ | HIV↑ | BACE↑ | SIDER↑ | Mean↑ |
|---|---|---|---|---|---|---|---|
| GEM | 72.4(0.4) | 78.1(0.1) | 90.1(1.3) | 80.6(0.9) | 85.6(1.1) | 67.2(0.4) | 79.0 |
| ChemBerta | 64.3 | - | 90.6 | 62.2 | - | - | - |
| MolCLR | 73.6(0.5) | 79.8(0.7) | 93.2(1.7) | 80.6(1.1) | 89.0(0.3) | 68.0(1.1) | 80.7 |
| MGSSL | 70.5(1.1) | 76.5(0.4) | 80.7(2.2) | 79.5(1.1) | 79.7(0.8) | 61.8(0.7) | 74.8 |
| $DMP_{TF}$ | 78.1(0.5) | 78.8(0.5) | 95.0(0.5) | 81.0(0.7) | 89.3(0.9) | 69.2(0.7) | 81.9 |
| MoMu | 70.5(2.0) | 75.6(0.3) | 79.9(4.1) | 76.2(0.9) | 77.1(1.4) | 60.5(0.9) | 73.3 |
| SMI-MLM | 89.4(1.9) | 76.2(1.6) | 90.6(1.8) | 79.8(1.2) | 86.6(0.4) | 66.5(0.5) | 81.5 |
| SMI-EDITOR | **93.5**(2.2) | **81.4**(1.1) | **95.2**(1.3) | **81.6**(0.7) | **89.9**(0.2) | **69.8**(0.6) | **85.2** |

## K  PERFORMANCE ADVANTAGES OF SMI-EDITOR OVER AUTO-REGRESSIVE MODELS

To comprehensively compare SMI-EDITOR with autoregressive models, we trained a decoder-only model with identical architecture and size to SMI-EDITOR using an autoregressive language-modeling objective, referred to as SMI-GPT. We evaluated SMI-GPT's performance across several downstream tasks, with results shown in Table 9. The findings indicate that SMI-EDITOR can perform better than SMI-GPT. Below is an analysis of these results:

### K.1  IMPLEMENTATION DETAILS FOR SMI-GPT(NT) AND SMI-GPT(EMB)

**SMI-GPT(NT)**: This approach uses next-token prediction for downstream classification tasks by appending a special token (e.g., $Label_0$, $Label_1$) at the end of each SMILES sequence to denote the classes of sample's label. The model learns to predict the correct label token during fine-tuning. **SMI-GPT(Emb)**: The representations of each token in the SMILES string extracted by the SMI-GPT model are processed using mean pooling. The resulting pooled representation is then fed into a classification head, which predicts the class of the SMILES.

---

[3]https://github.com/deepchem/deepchem

Table 9: Results of SMI-EDITOR and SMI-GPT on MoleculeNet datasets using DeepChem splits. ROC-AUC scores (higher is better) are reported for all tasks The best results are **bolded**

| Method | BBBP↑ | Tox21↑ | ClinTox↑ | HIV↑ | BACE↑ | SIDER↑ | Mean↑ |
|---|---|---|---|---|---|---|---|
| SMI-GPT(NT) | 88.5 | 74.3 | 88.9 | 68.8 | 76.2 | 63.7 | 76.7 |
| SMI-GPT(Emb) | 91.2 | 75.1 | 91.4 | 79.4 | 86.2 | 67.1 | 81.7 |
| MoMu | 70.5 | 75.6 | 79.9 | 76.2 | 77.1 | 60.5 | 73.3 |
| SMI-MLM | 89.4 | 76.2 | 90.6 | 79.8 | 86.6 | 66.5 | 81.5 |
| SMI-EDITOR | **93.5** | **81.4** | **95.2** | **81.6** | **89.9** | **69.8** | **85.2** |

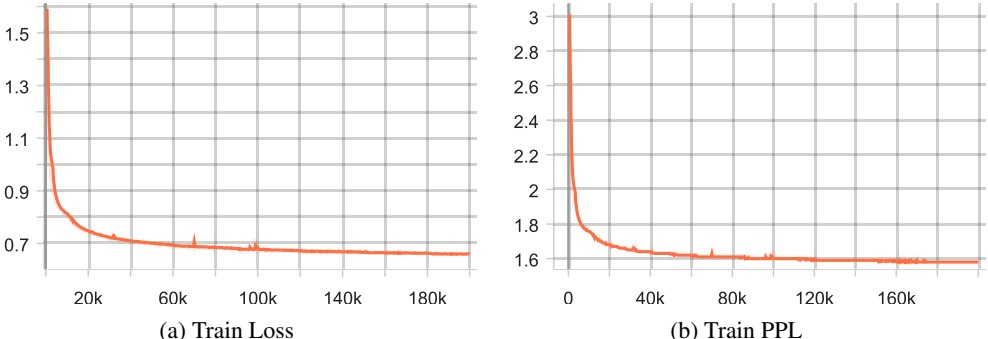

(a) Train Loss  (b) Train PPL

Figure 11: The training loss and perplexity (PPL) curves of the SMI-GPT model.

## K.2   ADVANTAGES OF THE ENCODER-ONLY SMI-EDITOR ARCHITECTURE

As shown in Table 9, SMI-EDITOR consistently outperforms SMI-GPT(Emb) and SMI-GPT(NT), highlighting its superior semantic learning capabilities. SMI-GPT(Emb) achieves better performance than SMI-GPT(NT), suggesting that pretraining-based feature transfer is preferable for molecular property prediction tasks. Therefore, the encoder-only pre-trained model is highly suitable for molecular property prediction tasks.

## K.3   RAPID CONVERGENCE IN AUTOREGRESSIVE LMS

We provide the training curve of the SMI-GPT model in Figure 11, which shows that the loss decreases rapidly during the early stages of training. Similarly, the perplexity also drops quickly, reaching approximately 1.6 at the 40K training step. By the end of training, the model's Perplexity falls below 1.6, which is significantly lower than the perplexity typically observed for GPT models trained on text data.

## K.4   WHY DOES THIS PHENOMENON OCCUR?

For auto-regressive LMs, each time a new token is generated, it receives all preceding tokens as prefix input. This means that when the model generates tokens at later positions, it has access to more comprehensive contextual information (i.e., a longer prefix and more complete sequence information). As a result, the prediction difficulty for tokens in later positions is significantly reduced, allowing the model to converge more easily. A key difference between SMI-EDITOR and SMI-GPT is that in SMI-EDITOR, each discarded token is predicted independently, with equal importance assigned to the prediction of each token. This enables SMI-EDITOR to more effectively capture the complete semantic information encoded in the tokens.

In summary, compared to LLMs on text data, GPT models on SMILES data converge significantly faster and achieve much lower perplexity. This indicates that SMILES data is inherently easier to fit than natural-language text. Therefore, it is crucial to design effective methods to extract richer semantic information from SMILES. Our SMI-EDITOR represents a meaningful and successful exploration

in this direction, highlighting the importance of leveraging substructural fragment information within SMILES data.

## L    PERFORMANCE OF SMI-EDITOR WITH FRAGMENT CORRECTION

### L.1    TRAINING SMI-EDITOR TO CORRECT ERRORS AND REMOVE EXTRANEOUS COMPONENTS DID NOT IMPROVE PERFORMANCE

We implemented a version of SMI-EDITOR that learns to correct erroneous functional groups and remove extraneous substructures, referred to as SMI-EDITOR-Cor. However, SMI-EDITOR-Cor did not outperform the original SMI-EDITOR on downstream tasks. Table 10 below compares the performance of SMI-EDITOR and SMI-EDITOR-Cor, showing that their performance is similar, demonstrating the limited benefit of incorporating these tasks.

### L.2    ANALYSIS OF SMI-EDITOR-COR'S PERFORMANCE

We attribute SMI-EDITOR-Cor's lack of improvement to the following reasons:

- **Correcting errors and removing extraneous components provide limited additional training signals:** SMI-EDITOR's training comprises two major steps: deletion and insertion. During deletion, erroneous functional groups and extraneous substructures are removed, while the insertion step involves learning to recover the correct tokens in the appropriate positions. Thus adding erroneous functional groups or extraneous substructures affects only the deletion step, which is a simpler task providing limited information. Moreover, as shown in Table 3 of the main text, ablating the token deletion (TokDel) step has minimal performance impact.

- **Identifying erroneous functional groups and extraneous structures is too simple for the model:** SMI-EDITOR-Cor constructs erroneous inputs through random substitutions, often resulting in chemically invalid SMILES that are easy for the model to identify. Consequently, the simplicity of the training task limits further performance improvement.

Table 10: Performance comparison between SMI-EDITOR-Cor and SMI-EDITOR.

| Method | BACE↑ | BBBP↑ | SIDER↑ | Tox21↑ | ToxCast↑ | Mean↑ |
|---|---|---|---|---|---|---|
| SMI-EDITOR-Cor | 80.6 | 77.1 | 62.2 | 76.8 | 68.0 | 72.9 |
| SMI-EDITOR | 80.3 | 77.4 | 63.0 | 77.1 | 67.4 | 73.0 |

## M    HOW THE FRAGMENT DROP RATIO AFFECT SMI-EDITOR

To investigate the impact of the fragment drop ratio on SMI-EDITOR, we trained SMI-EDITOR models with different drop ratios (i.e., 15%, 30%, 45%) and analyzed their training curves and downstream task performance. The results indicate that increasing the drop ratio significantly raises training loss for SMI-EDITOR, indicating that its pretraining task is more challenging than MLM. Below are the detailed findings:

### M.1    IMPACT ON SMI-EDITOR'S CONVERGENCE

We plotted the training and validation loss curves for SMI-EDITOR with varying drop ratios in Figure 12. **The results show that as the drop ratio increases, both training and validation losses rise significantly.** Compared to Figure 2c of the paper, the loss increase for SMI-EDITOR is more pronounced than for MLM, confirming that SMI-EDITOR's task is inherently more challenging.

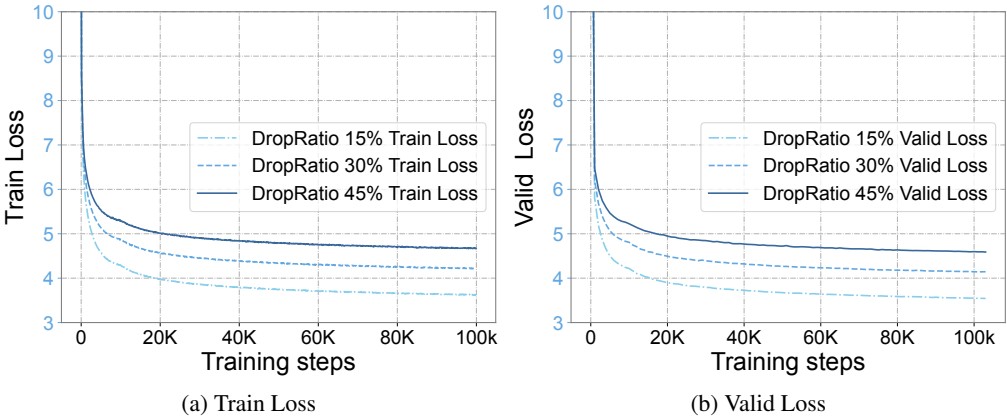

(a) Train Loss  (b) Valid Loss

Figure 12: The training loss and valid loss of the SMI-EDITOR with different fragment drop ratios.

## M.2 IMPACT ON DOWNSTREAM TASK PERFORMANCE

We evaluated the performance of SMI-EDITOR and MLMs with varying drop or mask ratios. The results are summarized in Table 11. From Table 11, it can be observed that as the mask ratio increases, the average performance of the SMI-MLMmodel shows no significant change, while the performance of the SMI-EDITOR model declines as the drop ratio increases. This indicates that SMI-EDITOR represents a more challenging training task.

**Here is a more detailed explanation:**

- SMI-EDITOR discards chemically-meaningful substructures that often serve as standalone semantic units. This also makes predicting the discarded fragments more difficult than predicting individual masked tokens. Dropping more substructures severely disrupts the molecular structure, making it harder for the model to reconstruct the original molecule.

- MLM, on the other hand, randomly masks tokens in SMILES sequences. Since SMILES tokens often represent individual atoms or bonds, masking does not typically disrupt the molecular semantics significantly. For instance, masking one or two atoms of a functional group like $-COOH$ still leaves enough contextual information to reconstruct it. Additionally, the probability of masking an entire functional group is low due to MLM's token-based masking mechanism. This explains why MLM performance is less sensitive to mask ratio increases, as also reflected in Figure 2c of the paper: Different Mask Ratios Cannot Alleviate Rapid Saturation.

Table 11: Performance of SMI-EDITOR and SMI-MLMwith different drop or mask ratios on downstream tasks.

| Method | BACE↑ | BBBP↑ | SIDER↑ | Tox21↑ | ToxCast↑ | Mean↑ |
|---|---|---|---|---|---|---|
| SMI-MLM(15%) | 77.8 | 68.6 | 61.2 | 75.1 | 64.9 | 69.5 |
| SMI-MLM(30%) | 78.3 | 70.2 | 58.2 | 76.0 | 63.7 | 69.3 |
| SMI-MLM(45%) | 78.4 | 66.1 | 59.3 | 76.4 | 65.5 | 69.1 |
| SMI-EDITOR(15%) | 80.3 | **77.4** | **63.0** | 77.1 | **67.4** | **73.0** |
| SMI-EDITOR(30%) | **81.6** | 73.3 | 59.6 | 77.0 | 66.8 | 71.7 |
| SMI-EDITOR(45%) | 79.3 | 72.2 | 61.1 | **77.8** | 67.1 | 71.5 |

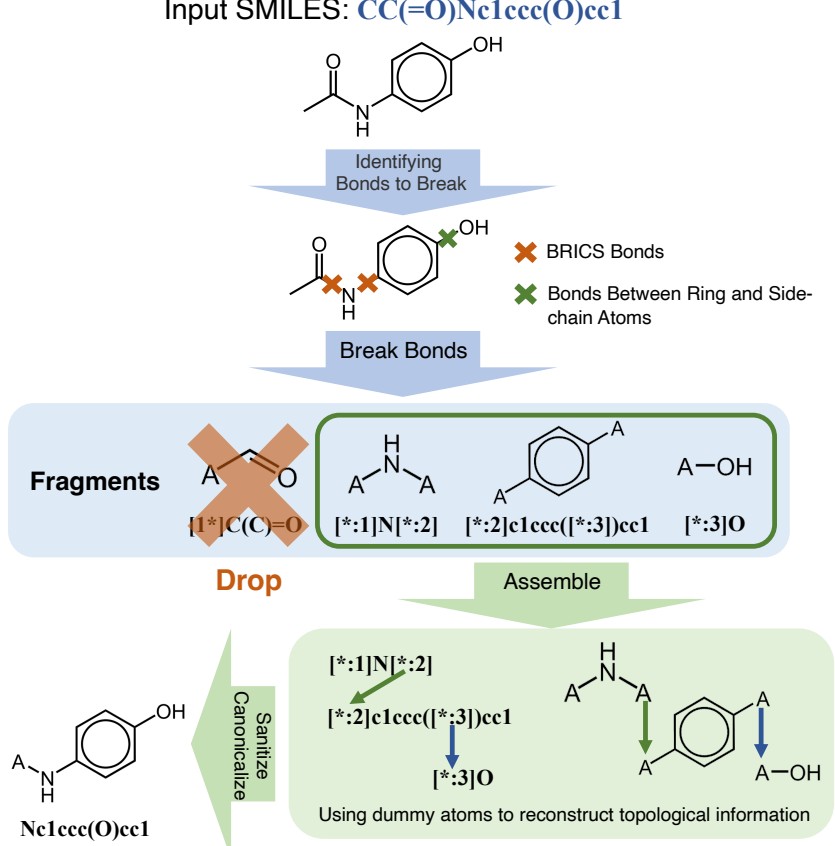

Figure 13: An Example Workflow of Molecule Fragmentation and Assemble with Paracetamol

## N PERFORMANCE OF SMI-EDITOR ON MOLECULAR PROPERTY REGRESSION TASKS

We evaluated the model's performance on three molecular property regression tasks, as shown in Table 12. SMI-EDITOR achieved the best performance compared to baseline models and significantly outperformed the MLM model.

Table 12: Performance of SMI-EDITOR on molecular property regression tasks.

| Method | ESOL↓ | FreeSolv↓ | Lipo↓ |
|---|---|---|---|
| MPNN | 0.58 | 1.150 | 0.7190 |
| DMP$_{TF}$ | 0.700 | - | - |
| A-FP | 0.503 | 0.736 | 0.578 |
| SMI-MLM | 0.576 | 0.709 | 0.642 |
| SMI-EDITOR | **0.362** | **0.524** | **0.565** |

## O A CASE STUDY FOR FRAGMENTS ASSEMBLE

We provide an example workflow of data processing in pre-training with Paracetamol (SMILES: CC(=O)Nc1ccc(O)cc1) in Figure 13.

## P   THE SCALABILITY OF SMI-EDITOR

We added results showing the performance of SMI-EDITOR and SMI-MLM models of varying sizes on downstream tasks, which further demonstrate SMI-EDITOR's strong scalability. These results are shown in Table 13. It is evident that while increasing model size has minimal impact on MLMs, larger SMI-EDITOR models show more consistent performance gains. **This confirms the claim that SMI-EDITOR has better scalability compared to MLMs.**

Table 13: Performance of SMI-EDITOR and SMI-MLMwith different scales on downstream tasks.

| Method | BACE↑ | BBBP↑ | SIDER↑ | Tox21↑ | ToxCast↑ | Mean↑ |
|---|---|---|---|---|---|---|
| SMI-MLM(Small) | 76.8 | 69.6 | 60.5 | 75.3 | 64.2 | 69.2 |
| SMI-MLM(Base) | 76.6 | 69.3 | 59.9 | 75.3 | 64.4 | 69.1 |
| SMI-MLM(Big) | 77.4 | 68.7 | 60.8 | 75.1 | 65.3 | 69.4 |
| SMI-EDITOR(Small) | 78.3 | 72.6 | 59.4 | 75.6 | 65.1 | 70.2 |
| SMI-EDITOR(Base) | 79.2 | 73.2 | **61.0** | 75.7 | 65.8 | 71.0 |
| SMI-EDITOR(Big) | **79.3** | **74.2** | 60.9 | **76.7** | **66.4** | **71.5** |

## Q   THE TRAINING COST OF SMI-EDITOR

We measured that the training cost of SMI-EDITOR is approximately three times that of MLMs (i.e., SMI-MLM) for the same model size, training hyperparameters, and data. However, the training cost of SMI-EDITOR remains acceptable. To better analyze the impact of training cost, we trained an MLM with equivalent computational cost (SMI-MLM(More)). Results showed that SMI-MLM(More) performed worse than the original SMI-MLMand significantly lagged behind SMI-EDITOR, highlighting that merely increasing MLM training cost does not yield better results.

### Q.1   REASONS FOR HIGHER TRAINING COST IN SMI-EDITOR

SMI-EDITOR requires computing expert actions (using a computationally expensive dynamic programming algorithm) and modeling three different editing operations, which introduces additional overhead.

### Q.2   ACCEPTABLE TRAINING COST

Training SMI-EDITOR on a dataset with 19M compounds using four RTX 3090 GPUs took approximately 24.6 hours. Scaling SMI-EDITOR to larger datasets (e.g., more than 100M compounds) is feasible, demonstrating its potential for broader applications.

### Q.3   SMI-EDITOR PERFORMS BETTER UNDER THE SAME TRAINING COST WITH MLM

We trained SMI-MLM(More) with the same computational cost as SMI-EDITOR by increasing its training steps from 120K to 360K. Table 14 shows that SMI-MLM(More) performs worse than the SMI-EDITOR and original SMI-MLM. This is due to rapid saturation issues in MLM training on SMILES data. **This also indicates that the speed of model training is not the most important factor; what matters more is whether the model can efficiently extract high-quality semantic representations**. This highlights the importance of designing more powerful training schemes like SMI-EDITOR to effectively extract meaningful information from SMILES.

### Q.4   HIGHER PERFORMANCE CEILING FOR SMI-EDITOR

Although the inclusion of Experts slows down the training speed of the SMI-EDITOR model, it also enriches the semantic information the model learns. This gives SMI-EDITOR greater scalability and a higher performance ceiling compared to SMI-MLM. As shown in Table 14, SMI-EDITOR benefits more from more computational resources. This makes SMI-EDITOR a better choice when given sufficient training budget.

Table 14: Performance of SMI-EDITOR and SMI-MLM with training steps on downstream tasks.

| Method | BACE↑ | BBBP↑ | SIDER↑ | Tox21↑ | ToxCast↑ | Mean↑ |
|---|---|---|---|---|---|---|
| SMI-MLM(More) | 74.3 | 66.2 | 49.5 | 73.3 | 62.3 | 65.1 |
| SMI-MLM | 77.8 | 68.6 | 61.2 | 75.1 | 64.9 | 69.5 |
| SMI-EDITOR | **80.3** | **77.4** | **63.0** | **77.1** | **67.4** | **73.0** |

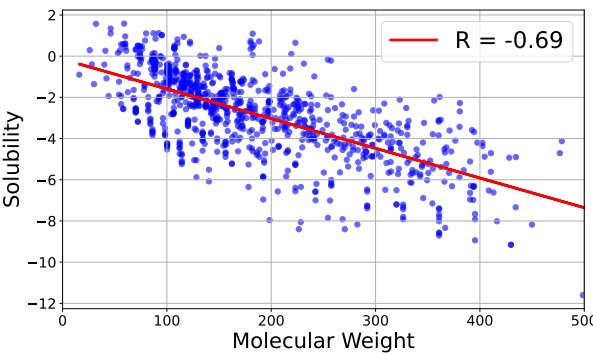

Figure 14: The relationship between molecular weight and solubility in the ESOL training set

## R  A MORE DETAILED ANALYSIS OF THE MODEL'S SUBSTRUCTURE MODELING CAPABILITY

The observed trends for the FreeSolv dataset are fully consistent with our expectations and align with the definition of its physical properties. On the other hand, the performance on the ESOL dataset is influenced by additional factors such as molecular weight. We also designed more analytical experiments to further investigate the behavior of the SMI-EDITOR model, and the results demonstrate that the model's behavior aligns with expectations. Detailed explanations are as follows.

For the FreeSolv dataset, the observed trends align with its physical property definitions. FreeSolv reflects the hydration free energy of compounds, defined as the free energy change when a compound transitions from a non-dissolved state to a dissolved state. When hydrophilic groups in a molecule are reduced, the change in hydration free energy increases, leading to higher hydration free energy. Therefore, when we remove hydrophilic groups from the molecule, the model predicts an increase in hydration free energy, consistent with the trend observed in Figure 4b, which matches our expectations.

For the ESOL task, the model predictions are significantly influenced by molecular weight The ESOL dataset reflects compound solubility, which is strongly negatively correlated with molecular weight: the larger the molecular weight, the lower the solubility. We plotted a scatter diagram (Figure 14) showing the relationship between molecular weight and solubility in the ESOL training set. A clear negative correlation with a coefficient of $R = -0.69$ is observed. Consequently, when functional groups or atoms are removed from a molecule, its molecular weight decreases, leading the model to predict an increase in solubility. This explains why, in Figure 4a, the model predicts increased solubility regardless of whether hydrophilic groups or random groups are removed. The increase is more significant with random deletions, demonstrating the model's ability to distinguish between hydrophilic group deletions and random deletions.

To eliminate the influence of molecular weight, we designed a hydrophilic group replacement scheme (HG Rep). We replaced all hydrophilic groups in a molecule with non-hydrophilic groups of similar molecular weight (e.g., methyl, ethyl, propyl) and compared this hydrophilic group replacement scheme (HG Rep) with a random group replacement scheme (Rand Rep), where random groups were replaced with others of similar weight. The results, shown in Figure 15, reveal that SMI-EDITOR effectively distinguishes between HG Rep and Rand Rep, demonstrating its ability to model key

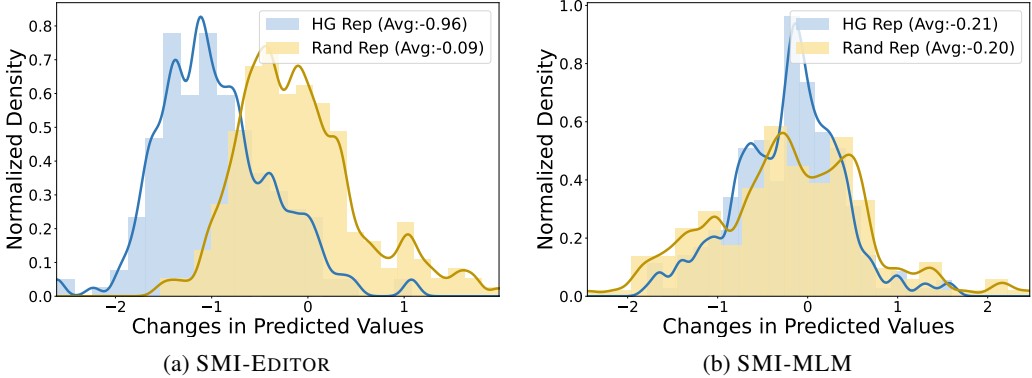

(a) SMI-EDITOR  (b) SMI-MLM

Figure 15: **Substructure Semantics Modeling on ESOL Dataset.** We compared the effects of two molecular perturbation methods on the SMI-EDITOR's and SMI-MLM's predictions of hydrophilicity. Figure 15a show that the impact of replacing hydrophilic groups (HG Rep) and randomly replacing atoms (Rand Rep) on the model's predictions differs significantly, both in the average change in prediction values and their distributions.

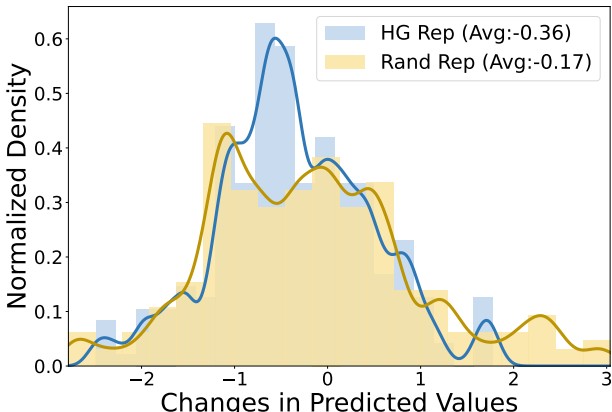

Figure 16: Substructure Semantics Modeling on ESOL Dataset of Auto-regressive LM.

molecular group semantics. It also correctly predicts that replacing hydrophilic groups reduces molecular solubility.

Furthermore, we plotted the distribution of predicted changes for MLMs and Auto-regressive LMs before and after these replacement operations in Figure 16. The results show that these models perform significantly worse than the SMI-EDITOR in distinguishing between random replacements and hydrophilic group replacements. This further highlights the superiority of the SMI-EDITOR in modeling the semantics of molecular substructures.

## S   SMI-EDITOR'S PERFORMANCE ON RETROSYNTHESIS PREDICTION TASKS

Considering that the original SMI-EDITOR is an encoder-only model and cannot be directly applied to generative tasks, we further pretrained a model based on an encoder-decoder architecture, referred to as **SMI-EDITOR-Gen**. We tested its performance on the retrosynthesis prediction task, where it achieved state-of-the-art results. Below is a detailed discussion:

**Model Details of SMI-EDITOR-Gen.** SMI-EDITOR-Gen adopts a transformer architecture with a base-sized scale (Vaswani et al., 2017). During pretraining, the input to the encoder consists of SMILES strings with missing molecular fragments, while the decoder's pretraining task is to reconstruct the original SMILES. Following approaches commonly used in machine translation (Vaswani et al., 2017), the features extracted by the encoder are passed to the decoder through

encoder-decoder attention (Vaswani et al., 2017) . Compared to SMI-EDITOR, the most significant difference is that the encoder-decoder architecture enables SMI-EDITOR-Gen to perform sequence-to-sequence generative tasks, allowing us to explore the model's capabilities in such tasks.

**SMI-EDITOR-Gen Exhibits Strong Performance in Retrosynthesis Prediction Tasks.** Following the experimental setup of EditRetro(Han et al., 2024), we evaluated SMI-EDITOR-Gen on the retrosynthesis prediction task. During fine-tuning, we applied the same fine-tuning strategies and data augmentation techniques as EditRetro. The experimental results, shown in Table 15, demonstrate that SMI-EDITOR-Gen achieved strong performance on the USPTO-50K dataset. This validates that the pretraining approach proposed by SMI-EDITOR also exhibits excellent performance and great potential in generative tasks.

Table 15: Top-$k$ exact match accuracy of SMI-EDITOR and baselines on the USPTO-50k dataset.

|  | Top-1 ↑ | Top-3 ↑ | Top-5 ↑ | Top-10 ↑ |
|---|---|---|---|---|
| RetroPrime | 51.4% | 70.8% | 74.0% | 76.1% |
| Transformer | 42.4% | 58.6% | 63.8% | 67.7% |
| SCROP | 43.7% | 60.0% | 65.2% | 68.7% |
| MEGAN | 48.1% | 70.7% | 78.4% | 86.1% |
| GTA | 51.1% | 67.6% | 74.8% | 81.6% |
| Retroformer | 53.2% | 71.1% | 76.6% | 82.1% |
| Graph2Edits | 55.1% | 77.3% | 83.4% | 89.4% |
| R-SMILE | 56.3% | 79.2% | 86.2% | **91.0%** |
| EditRetro | 60.8% | 80.6% | 86.0% | 90.3% |
| SMI-EDITOR | **61.2%** | **80.9%** | **86.4%** | 89.7% |

## T SPE TOKENIZER DOES NOT IMPROVE SMILES MLM PERFORMANCE

We trained a SMILES MLM with SPE tokenizer, **SMI-MLM(SPE)**, using the same architecture and hyperparameters as SMI-EDITOR, and evaluated it on multiple tasks. As shown in Table 16, SMI-MLM(SPE) performs similarly to SMI-MLMand significantly worse than SMI-EDITOR. This demonstrates that introducing SPE cannot replicate the effectiveness of SMI-EDITOR. The reasons are as follows:

- **Limited Fragment Diversity**: SPE relies on a fixed vocabulary, limiting the diversity of fragment-level information it can capture. In contrast, SMI-EDITOR dynamically fragments molecules using the BRICS algorithm, capturing a wider variety of molecular substructures.

- **Topology Information Leakage**: SPE-based models retain token position information, which is tied to molecular topology in SMILES, making the prediction task easier but less effective.

- **Lack of Chemical Context**: SMI-EDITOR fragments molecules based on chemical rules, allowing it to capture substructure information more relevant to molecular properties, unlike SPE, which relies on character pair frequencies.

- **Superior Performance with Fragment-Level Supervision**: A MLM trained with fragment-level supervision, SMI-MLM(Frag), outperforms SMI-MLM(SPE), as shown in Table 16. This validates the effectiveness of SMI-EDITOR's training approach.

## U MASKED SPAN LM DOES NOT IMPROVE SMILES LM PERFORMANCE

To highlight the differences between SMI-EDITOR and Masked Span LMs (MSLMs), we trained a SMILES model using MSLM, which randomly masks continuous sequences in SMILES and predicts the missing parts (similar to SpanBERT(Joshi et al., 2020)). This model, referred to as **SMI-MLM(SPAN)**, shows performance comparable to SMI-MLMbut significantly worse than SMI-EDITOR (see Table 16). This further demonstrates SMI-EDITOR's advantages over traditional MSLMs. Reasons for Poor Performance of Traditional MSLMs:

Table 16: Performance comparison of MLMs with different pretraining strategies.

| | BACE↑ | BBBP↑ | SIDER↑ | Tox21↑ | ToxCast↑ | Mean↑ |
|---|---|---|---|---|---|---|
| SMI-MLM | 77.8 | 68.6 | 61.2 | 75.1 | 64.9 | 69.5 |
| SMI-MLM(SPE) | 76.7 | 71.1 | 59.3 | 74.7 | 65.3 | 69.4 |
| SMI-MLM(SPAN) | 78.6 | 67.2 | 59.4 | 76.1 | 62.3 | 68.7 |
| SMI-MLM(Frag) | 79.4 | 73.3 | 62.1 | 74.0 | 64.8 | 70.7 |
| SMI-EDITOR | **80.3** | **77.4** | **63.0** | **77.1** | **67.4** | **73.0** |

- **Differences between Text Data and SMILES Data.** Unlike text, molecular data has complex topological structures. In text, adjacent tokens often have strong semantic relevance, and continuous spans convey related information, making span masking effective for learning local semantics. However, **SMILES lacks such locality**; a single functional group may not appear contiguous, and adjacent tokens may lack strong relevance. For example, aromatic rings with multiple substituents often appear discontinuous in SMILES (we provide a specific case **CASE1**). This limits the effectiveness of applying span masking directly to SMILES data.

- **Traditional MSLM (e.g., T5(Raffel et al., 2020)) and SMI-EDITOR Have Different Implementations; Traditional MSLM is Unsuitable for SMILES Data.** Text data's semantic continuity enables models like T5 to use random span masking, where continuous text segments are masked for prediction. In contrast, SMILES lacks this continuity, so SMI-EDITOR uses a fragmentation algorithm to split molecules into chemically meaningful fragments. The model predicts missing fragments, which may not correspond to continuous SMILES segments. Unlike traditional MSLM, SMI-EDITOR focuses on masking chemically significant fragments, a key difference in its design.

- **Better Performance of SMI-MLM(Frag).** The improved performance of SMI-MLM(Frag) over SMI-MLM(SPAN) highlights the superiority of SMI-EDITOR's fragment-level supervision. While SMI-MLM(SPAN) uses the traditional MSLM approach, SMI-MLM(Frag) incorporates supervision signals similar to SMI-EDITOR, enabling it to better capture molecular substructure information.

**CASE1** When does SMILES exhibit discontinuity: SMILES is a linearized representation of graph-structured molecules, which inherently causes discrepancies between molecular topology and sequence-level representation. For example, when a ring contains multiple substituents, its representation in SMILES often becomes discontinuous. Consider Glibenclamide, a drug used for diabetes treatment, with the canonical SMILES: COc1ccc(Cl)cc1C(=O)NCC**c2ccc**(S(=O)(=O)NC(=O)NC3CCCCC3)**cc2**. Here, the bolded atoms originate from the same aromatic ring, but due to the multiple substituents, this ring is represented discontinuously in SMILES. Additionally, the aromatic carbon **cc2** is adjacent to CCC3 atoms from a distant cycloalkane ring. Such discontinuities are common in SMILES and adversely affect Masked Span LMs.

## V  COMPARISON BETWEEN SMI-EDITOR AND CONTRASTIVE LEARNING

**Similarities**: Both contrastive learning and SMI-Editor aim to learn alignment.

- **Contrastive learning aligns representations of different views.** The core idea of contrastive learning is to bring the representations of different views of the same sample (positive pairs) closer while pushing representations of different samples (negative pairs) apart. Essentially, this process learns the correct alignment between views of the same sample.

- **SMI-Editor aligns representations of missing substructures and contexts.** As Fu et al. (2022) noted, MLMs align the representations of contexts and missing words during training. Similarly, SMI-Editor aligns the representations of missing substructures and their contexts. For example, given the input Nc1ccc(O)cc1, the model need to predict the complete molecule

CC(=O)Nc1ccc(O)cc1. SMI-Editor can effectively align the representation of the missing fragment CC(=O) with the context Nc1ccc(O)cc' through this process.

**Differences**: The alignment targets differ between the two paradigms.

- **Contrastive learning focuses on global information**: The representations to be aligned often correspond to different augmented views of the same molecule, such as through atom deletion, bond deletion, or subgraph deletion. These views typically preserve the molecule's overall structure and thus contain global information.

- **SMI-Editor emphasizes aligning local substructure information with global context**: In SMI-Editor, the context typically corresponds to the molecule's backbone, representing global information, while the missing substructures contain local information.

- **SMI-Editor is more sensitive to local structure information**: By aligning local substructures with global context, SMI-Editor learns finer-grained semantics from SMILES data, making it better suited to capturing detailed molecular information than contrastive learning.

## W  K-FOLD CROSS-VALIDATION OF THE SMI-EDITOR MODEL.

Using a 5-fold setup, we evaluated SMI-EDITOR's performance on the training sets of BACE, BBBP, SIDER, Tox21, ToxCast, ClinTox and MUV. The results are shown in Table 17. These results demonstrate that SMI-EDITOR exhibits strong performance and stability across downstream tasks.

Each dataset was evenly divided into five parts. In each run, one part was selected as the validation set, while the remaining four parts were used as the training set. The model was trained and evaluated on the validation set. This process was repeated five times to complete all runs.

Table 17: 5-fold cross-validation results of the SMI-EDITOR model.

| | BACE↑ | BBBP↑ | SIDER↑ | Tox21↑ | ToxCast↑ | ClinTox↑ | MUV↑ |
|---|---|---|---|---|---|---|---|
| Run 1 | 91.92 | 97.64 | 62.59 | 83.69 | 75.83 | 99.76 | 77.23 |
| Run 2 | 91.86 | 96.27 | 66.89 | 84.09 | 73.31 | 98.73 | 79.49 |
| Run 3 | 90.82 | 98.53 | 62.60 | 84.87 | 73.52 | 99.6 | 77.4 |
| Run 4 | 91.13 | 98.77 | 63.32 | 83.95 | 74.60 | 99.82 | 77.58 |
| Run 5 | 90.68 | 97.84 | 63.50 | 85.83 | 75.51 | 98.61 | 79.39 |
| **Mean** | **91.28** | **97.81** | **63.78** | **84.48** | **74.55** | **99.30** | **78.23** |
| **Std** | **0.58** | **0.97** | **1.78** | **0.87** | **1.13** | **0.59** | **1.12** |

## X  BROAD APPLICATIONS OF ATOM-LEVEL TOKENIZERS

Currently, many SMILES LMs, including MLM and autoregressive LMs, rely on atom-level tokenizers to process molecular representations. Atom-level tokenizers break down SMILES strings into individual atomic units or tokens, such as atoms and simple symbols (e.g., "C", "O", "="). This approach simplifies the tokenization process and aligns well with the intrinsic atomic structure of molecules, enabling models to capture fine-grained atomic interactions and features. For example, MolXPT (Liu et al., 2023b) and Dual-view Molecular Pre-training (Zhu et al., 2023) explicitly leverage atom-level tokenization to enhance the granularity of molecular representations, facilitating downstream tasks such as molecule generation and property prediction.

Atom-level tokenization has the advantage of maintaining a straightforward correspondence between the SMILES representation and the underlying molecular structure, making it easier for the model to interpret local chemical environments. This granularity is particularly beneficial for tasks that require precise predictions. For instance, studies such as ChemBERTa (Chithrananda et al., 2020), Molecular Transformer (Schwaller et al., 2019), and SMILES-BERT (Wang et al., 2019b) demonstrate that atom-level tokenization can achieve good performance in molecular property prediction tasks.

