# OpenReview forum: "SMI-Editor: Edit-based SMILES Language Model with Fragment-level Supervision"
_ICLR.cc/2025/Conference — ICLR 2025 Poster_

### Official Review · Reviewer_AGu3 · 2024-10-22

**Soundness:** 2
**Presentation:** 2
**Contribution:** 2
**Rating:** 3
**Confidence:** 4

**Summary:**

In this paper, the authors propose SMILES-Editor, an edit-based pre-trained SMILES language model by introducing a new pre-training strategy that randomly corrupts SMILES strings during the pre-training process and lets the LLM restore the original SMILES strings. Experimental results show that the new proposed pre-training strategy achieves better performance than the previous MLM task in various downstream tasks.

**Strengths:**

1. The idea of focusing on the atoms and fragments is meaningful and enables better learning of SMILES representations.
2. The new pre-training task may lead to an improvement in the valid generation of molecule SMILES representations.
3. The SMILES-Editor shows competitive performance compared to the original MLM, bringing an improvement in the pre-training of SMILES Language Models.

**Weaknesses:**

1. The authors claim that "most existing pre-trained SMILES language models (LMs) only provide supervision at the single-token level during pre-training and fail to fully leverage substructural information of molecules", which lacks enough support. In works like  [1], the tokenizers are initialized with SMILES Pair Encoding. Meanwhile, the sentence-piece method can also find common fragments in molecule SMILES strings, and these common fragments are usually critical functional groups. In this case, I do not agree with this claim.
2. The novelty of SMILES-Editor is limited. The main contribution of this paper is to propose a new pre-training strategy, while this strategy is not novel, and there is not a significant difference compared to the previous Masked Span Language Modeling used by T5. Furthermore, more advanced LLMs are now using decoder-only structures, while SMILES-Editor can only be adopted in LLMs with encoder structures, which further harms the novelty.
3. The experiments on the classification tasks do not incorporate the k-fold experiments, which raises concerns about the stability of the method.
4. The experiment results do not seem satisfying enough. And the comparison should include more pre-training strategies.
5. In Table 2, the performance of SMI-EDITOR-AtomsMasking in BACE is even better than SMILES-Editor.

#### References
[1] Li, X., & Fourches, D. (2021). SMILES pair encoding: a data-driven substructure tokenization algorithm for deep learning. Journal of chemical information and modeling, 61(4), 1560-1569.

**Questions:**

1. Could the authors find more evidence to support their claims in Weakness 1?
2. Could the authors explain more about Weakness 2?
3. I am wondering about the comparison between different pre-training strategies. Could the authors compare the auto-regressive pre-training strategies in identical decoder-only structures?
4. What is the time cost or complexity of SMILES-Editor compared to the previous MLM?

---

> ### Author Response · Authors · 2024-11-19
> **Response to Reviewer AGu3 (Part 1)**
>
> We appreciate the insightful suggestions from Reviewer AGu3. In the following sections, we will address all your concerns regarding comparison with SPE tokenizers, model's novelty, k-fold experiments, effectiveness of our model, the ablation studies of SMI-Editor, comparison between different pre-training strategies, time cost of SMILES-Editor . And we have also supplemented new experimental results (performance comparison between MLM with SPE and SMI-Editor, performance and architecture comparison with more pre-training strategies, k-fold experiment results of SMI-Editor, performance comparison on more datasets and with more baseline models, more ablation study results of SMI-Editor, statistics and analysis of training time cost for SMI-Editor.) to address some of the issues you have raised. We hope the replies can make our paper more clear. Any further comments are welcome!
>
> ---
>
> ### Reponse to: Tokenizers such as SMILES Pair Encoding can also provide fragment-level training signals for SMILES language models, so the claim in this paper is inaccurate.
>
> **A:** Thank you for your suggestion. We have added more discussions and comparisons with the SMILES Pair Encoding (SPE) method in the revised paper in Appendix S. Many SMILES language models use atom-level tokenizers, which are common and foundational in this field, making them a reasonable basis for discussion. Our experiments also show that using an SPE tokenizer has minimal impact on MLM performance, and its training signals differ significantly from the fragment-level supervision proposed in our paper. We have also added these resuts to Appendix S in the revised paper. Below is a summary of our findings:
>
> 1. **Atom-Level Tokenization is Common in SMILES LM Research**:
>    Many recent works [1-13] adopt atom-level tokenization in SMILES LM, highlighting its widespread application and impact. It's very reasonable and scientifically meaningful to base our discussions on an atom-level tokenizer. We have added more discussions on these works with atom-level tokenizers in Appendix W.
>
> 2. **SPE Tokenizer Has Limitations**:
>
>    - **Limited Fragment Diversity**: SPE relies on a fixed vocabulary, limiting the diversity of fragment-level information it can capture. Moreover, to prevent the long-tail problem of the vocabulary, the size of the SPE vocabulary is also very limited (usually less than 5k). This further prevents SPE from capturing more diverse fragment-level information. In contrast, SMI-Editor dynamically fragments molecules using the BRICS algorithm, capturing a wider variety of molecular substructures (as shown in Table F2 in part 2).
>    - **Limited Generalization Capability**: SPE merges frequently occurring fragments into single tokens and represents different fragments with distinct tokens. This approach prevents the model from directly capturing the semantic relationships between these fragments at the atomic level. A typical example is that SPE represents the functional groups `C(=O)`, `C(=O)N`, and `NC(=O)` with three different tokens. Although these fragments share a high degree of similarity at the atomic level, a model based on the SPE vocabulary struggles to directly capture this relationship.
>
>    - Given the limitations of SPE, **exploring the introduction of fragment-level supervision signals to models while maintaining atomic-level representations is a highly meaningful research question.**
>
> 3. **SPE Tokenizer Does Not Improve SMILES MLM Performance**:
>    We trained a SMILES MLM model with SPE tokenizer, **SMI-MLM(SPE)**, using the same architecture and hyperparameters as SMI-Editor, and evaluated it on multiple tasks. As shown in Table F1 in part 2, SMI-MLM(SPE) performs similarly to SMI-MLM and significantly worse than SMI-Editor. This demonstrates that introducing SPE cannot replicate the effectiveness of SMI-Editor. The reasons are:
>    - **Topology Information Leakage**: SPE-based masked language models also retain token position information, which is tied to molecular topology in SMILES, making the prediction task easier but less effective.
>    - **Lack of Chemical Context**: SMI-Editor fragments molecules based on chemical rules, allowing it to capture substructure information more relevant to molecular properties, unlike SPE, which relies on character pair frequencies.
>    - **Fragment-Level Supervision Performs Better than SPE**: A MLM model trained with fragment-level supervision, **SMI-MLM(Frag)**, outperforms SMI-MLM(SPE), as shown in Table F1 in part 2. This validates the effectiveness of SMI-Editor’s training approach. (details of **SMI-MLM(Frag)** is displayed in part 2)

---

> ### Author Response · Authors · 2024-11-19
> **Response to Reviewer AGu3 (Part 2)**
>
> In conclusion, we will expand the discussion on SPE in the revised paper. Our findings highlight that SMI-Editor’s fragment-level supervision is more effective and represents a fundamentally different approach to SMILES data modeling.
>
> **Details of SMI-MLM(Frag)**: We train a SMILES MLM model, referred to as SMI-MLM(Frag), under the exact same training settings as SMI-Editor, incorporating a fragment-level supervision similar to that of SMI-Editor. During training, a substructure from the molecule is selected, and all atoms and bonds within this substructure are masked, requiring the model to predict the entire masked fragment. To ensure comparability with SMI-Editor, we employed the identical molecular fragmentation scheme as used in SMI-Editor.
>
>
> Table F1. Performance comparison of MLM models with different pretraining strategies.
>
> |    | BACE↑    | BBBP↑    | SIDER↑   | Tox21↑   | ToxCast↑ | Mean↑    |
> | ------ | ------ | -------- | -------- | -------- | -------- | -------- |
> | SMI-MLM       | 77.8     | 68.6     | 61.2     | 75.1     | 64.9     | 69.5     |
> | SMI-MLM(SPE)  | 76.7     | 71.1     | 59.3     | 74.7     | 65.3     | 69.4     |
> | SMI-MLM(SPAN) | 78.6     | 67.2     | 59.4     | 76.1     | 62.3     | 68.7     |
> | SMI-MLM(Frag) | 79.4     | 73.3     | 62.1     | 74.0     | 64.8     | 70.7     |
> | SMI-Editor    | **80.3** | **77.4** | **63.0** | **77.1** | **67.4** | **73.0** |
>
>
>
> Table F2. The number of distinct molecular fragments observed during training for SPE and the fragment-level supervision of SMI-Editor.
>
> |      | SPE  | Fragment-level supervision of SMI-Editor |
> | ------ | ---- | -------- |
> | Number of Different Fragments | 3152 | 1585019                                  |
>
> ---
>
> ### Reponse to: There is not a significant difference compared to the previous Masked Span Language Modeling used by T5.
>
> **A:** Thank you for your suggestion. While works like T5 and SpanBERT have explored Masked Span Language Models (MSLMs) on text data, their motivations, methods, and outcomes differ fundamentally from SMI-Editor. Our experiments also confirm that MSLMs do not perform well for SMILES data. We have also added these resuts to Appendix T in the revised paper. Below is a detailed discussion and comparison:
>
> 1. **Traditional MSLMs Perform Poorly in SMILES Modeling**:
>    To highlight the differences between SMI-Editor and MSLMs, we trained a SMILES model using MSLM, which randomly masks continuous sequences in SMILES and predicts the missing parts (similar to SpanBERT). This model, referred to as **SMI-MLM(SPAN)**, shows performance comparable to SMI-MLM but significantly worse than SMI-Editor (see Table F1). This further demonstrates SMI-Editor's advantages over traditional MSLMs.
>
> 2. **Reasons for Poor Performance of Traditional MSLMs**:
>    1. **Differences between Text Data and SMILES Data:**
>       Unlike text, molecular data has complex topological structures. In text, adjacent tokens often have strong semantic relevance, and continuous spans convey related information, making span masking effective for learning local semantics. However, **SMILES lacks such locality**; a single functional group may not appear contiguous, and adjacent tokens may lack strong relevance. For example, aromatic rings with multiple substituents often appear discontinuous in SMILES (we provide a specific case **CASE1** in part 3). This limits the effectiveness of applying span masking directly to SMILES data.
>    2. **Traditional MSLM (e.g., T5) and SMI-Editor Have Different Implementations; Traditional MSLM is Unsuitable for SMILES Data:**
>       Text data's semantic continuity enables models like T5 to use random span masking, where continuous text segments are masked for prediction. In contrast, SMILES lacks this continuity, so SMI-Editor uses a fragmentation algorithm to split molecules into chemically meaningful fragments. The model predicts missing fragments, which may not correspond to continuous SMILES segments. Unlike traditional MSLM, SMI-Editor focuses on masking chemically significant fragments, a key difference in its design.
>    3. **Better Performance of SMI-MLM(Frag)**:
>       The improved performance of **SMI-MLM(Frag)** over SMI-MLM(SPAN) highlights the superiority of SMI-Editor’s fragment-level supervision (results are shown in Table 1).  While SMI-MLM(SPAN) uses the traditional MSLM approach, SMI-MLM(Frag) incorporates supervision signals similar to SMI-Editor, enabling it to better capture molecular substructure information.

---

> ### Author Response · Authors · 2024-11-19
> **Response to Reviewer AGu3 (Part 3)**
>
> 3. **Conclusion**:
>    MSLMs differ significantly from SMI-Editor in core ideas, implementation, and performance. These findings confirm that SMI-Editor represents a novel SMILES LM training approach, distinct from traditional MSLMs.
>
> **CASE1** When does SMILES exhibit discontinuity: SMILES is a linearized representation of graph-structured molecules, which inherently causes discrepancies between molecular topology and sequence-level representation. For example, when a ring contains multiple substituents, its representation in SMILES often becomes discontinuous. Consider **Glibenclamide**, a drug used for diabetes treatment, with the canonical SMILES:  COc1ccc(Cl)cc1C(=O)NCC**c2ccc**(S(=O)(=O)NC(=O)NC3CCCCC3)**cc2**. Here, the bolded atoms originate from the same aromatic ring, but due to the multiple substituents, this ring is represented discontinuously in SMILES. Additionally, the aromatic carbon **cc2** is adjacent to CCC3 atoms from a distant cycloalkane ring. Such discontinuities are common in SMILES and adversely affect Masked Span Language Models.
>
> ---
>
> ### Reponse to: The experiment results do not seem satisfying enough. And the comparison should include more pre-training strategies.
>
> **A:** Thank you for raising this insightful concern. We re-evaluated our model's performance on different downstream tasks using the dataset split provided by DeepChem. Previously, our model used a different test data split, which made it difficult to compare its performance with other works based on DeepChem’s split. Thus, we tested SMI-Editor on the DeepChem Split data, and the detailed results are presented in **General Response Table A1** and Appendix I in the revised draft. **The results show that SMI-Editor achieves significant performance improvements over baseline models, reaching SOTA levels**. Below is a further analysis of these results:
>
> 1. **SMI-Editor Outperforms Models with Different Training Paradigms**:
>    SMI-Editor’s average performance on these tasks surpasses molecular representation learning models based on contrastive learning pretraining, such as MolCLR and DMP\_{TF}. It also outperforms models trained with masked language models, such as ChemBERTa and SMI-MLM, as well as the autoregressive language model Galactica. Furthermore, SMI-Editor surpasses models based on molecular graph data, such as MolCLR, MGSSL, and MoMu, highlighting the potential of SMILES-based language models.
>
> 2. **SMI-Editor Achieves Strong Performance with Smaller Training Data**:
>    SMI-Editor outperforms DMP\_{TF}, which was trained on a dataset of over 100M compounds, despite being trained on a dataset of only 19M compounds. This demonstrates SMI-Editor’s ability to more effectively extract substructure information from SMILES, leading to significantly higher data efficiency.
>
> ---
>
>
> ### Reponse to: The comparison betweenand the auto-regressive pre-training strategies in identical decoder-only structures.
>
> **A:** Thank you for your suggestions and questions. We agree that this comparison is highly meaningful. To better evaluate the performance differences between the SMI-Editor model and auto-regressive LMs, we trained a decoder-only model, **SMI-GPT**, using the auto-regressive LM training objective. This model has the same size and architecture as SMI-Editor. We tested SMI-GPT’s performance across multiple downstream tasks and the results are shown in **General Response Part 2**.
>
> In summary, **SMI-Editor outperformed SMI-GPT, and the results from various tasks demonstrated that the encoder-only architecture of SMI-Editor is more suitable for representation learning tasks**. Additionally, we identified the reasons behind SMI-GPT’s performance disadvantages. For more detailed analyses and results, please refer to General Response Part 2 or Appendix J in the revised draft. These experimental findings highlight the effectiveness and potential of the SMI-Editor model design.

---

> ### Author Response · Authors · 2024-11-19
> **Response to Reviewer AGu3 (Part 4)**
>
> ### Reponse to: More advanced LLMs are now using decoder-only structures, while SMILES-Editor can only be adopted in LLMs with encoder structures, which harms the novelty.
>
> **A:** Thank you for your suggestions and questions. **The pretraining framework proposed by SMI-Editor demonstrates broad applicability and significant potential for further development**. We extended this framework to an Encoder-Decoder architecture, achieving excellent results in molecular retrosynthesis prediction tasks. This highlights SMI-Editor's potential in generative tasks. Below are further discussions and detailed results:
>
> 1. **Encoder Architecture Exhibits Superior Representation Learning Capability**:
>    Experiments and analysis in General Response Part 2 show that encoder-based architectures outperform decoder-only architectures in representation learning, effectively capturing bidirectional semantic dependencies from SMILES data. Therefore, using an encoder-based model for downstream molecular modeling tasks is both reasonable and effective. Furthermore, numerous molecular modeling models are designed with encoder-based architectures, achieving strong performance across various tasks [2,4,8,9,12,19,20], further validating the significance and value of exploring encoder-based designs.
>
> 2. **SMI-Editor’s Training Approach Has Significant Potential for Generative Tasks**:
>    SMI-Editor introduces fragment-level supervision during training, which is crucial for generative tasks. To explore this potential, we trained an Encoder-Decoder-based version of SMI-Editor, referred to as **SMI-Editor-Gen**, and tested it on retrosynthesis prediction tasks. The model achieved SOTA performance. As shown in Table F3, SMI-Editor-Gen delivered strong results on the USPTO-50K dataset (details provided in the supplementary materials). This demonstrates that the pretraining method proposed by SMI-Editor also excels in generative tasks, showcasing substantial potential.
>
> 3. **Non-Autoregressive Generative Models Like SMI-Editor Have Significant Potential in AI4Science**:
>    While unidirectional autoregressive language models have achieved great success in text tasks, the semantic dependency patterns in molecular and biological sequencing data differ significantly from those in text, with strong bidirectional dependencies. Generative models capable of modeling such bidirectional semantics are crucial for tasks like molecular generation and design. Recent works have explored non-autoregressive generative models for biological molecular data, achieving notable progress in retrosynthesis prediction [14], protein design [15,16,17], and single-cell data modeling [18]. This highlights the cutting-edge and meaningful nature of SMI-Editor’s research.
>
> 4. **SMI-Editor’s Success Can Inspire Pretraining Design for Decoder-Only Models**:
>    SMI-Editor demonstrates the importance of modeling fragment-level information in SMILES. However, traditional autoregressive language models lack fragment-level supervision during training. A promising direction for future research is to enable autoregressive models to effectively capture molecular substructure semantics. This could significantly improve their performance in tasks like molecular property prediction and design.
>
>
>
> Table F3. Top-k exact match accuracy of SMI-Editor and baselines on USPTO-50k dataset.
>
> |             | Top-1↑    | Top-3↑    | Top-5↑    | Top-10↑   |
> | ----------- | --------- | --------- | --------- | --------- |
> | RetroPrime  | 51.4%     | 70.8%     | 74.0%     | 76.1%     |
> | Transformer | 42.4%     | 58.6%     | 63.8%     | 67.7%     |
> | SCROP       | 43.7%     | 60.0%     | 65.2%     | 68.7%     |
> | MEGAN       | 48.1%     | 70.7%     | 78.4%     | 86.1%     |
> | GTA         | 51.1%     | 67.6%     | 74.8%     | 81.6%     |
> | Retroformer | 53.2%     | 71.1%     | 76.6%     | 82.1%     |
> | Graph2Edits | 55.1%     | 77.3%     | 83.4%     | 89.4%     |
> | R-SMILE     | 56.3%     | 79.2%     | 86.2%     | **91.0%** |
> | EditRetro   | 60.8%     | 80.6%     | 86.0%     | 90.3%     |
> | SMI-Editor  | **61.2%** | **80.9%** | **86.4%** | 89.7%     |

---

> ### Author Response · Authors · 2024-11-19
> **Response to Reviewer AGu3 (Part 5)**
>
> P.S. Details of USPTO-50K dataset: USPTO-50K is a high-quality dataset comprising approximately 50,000 chemical reactions extracted from U.S. patent literature. These reactions feature precise atom-to-atom mappings, clearly defining the correspondence between reactants and products. The dataset is categorized into 10 distinct reaction types, enabling detailed analysis and comparison with other existing methods. Widely used in prior research, this dataset is ideal for benchmarking our proposed method against current state-of-the-art approaches.  For our experiments on the USPTO-50K dataset, we adhere to the well-established data split scheme reported by Coley et al. Specifically, the dataset is divided into training, validation, and test sets comprising 40,000, 5,000, and 5,000 samples, respectively. This consistent division ensures comparability with previous studies and allows for a rigorous evaluation of our method. For the evaluation metric, we utilize the top-k exact match accuracy as our primary evaluation metric. This metric provides a rigorous assessment by comparing the canonical SMILES of the predicted reactants to the ground truth reactants in the test dataset.
>
> ---
>
> ### Reponse to: The experiments on the classification tasks do not incorporate the k-fold experiments, which raises concerns about the stability of the method.
>
> **A:** Thank you for your suggestion. We have added k-fold experimental results for classification tasks. Using a 5-fold setup, we evaluated SMI-Editor’s performance on the training sets of BACE, BBBP, SIDER, Tox21, and ToxCast. The results are shown in Table F4. These results demonstrate that SMI-Editor exhibits strong performance and stability across downstream tasks. We have also added these resuts to Appendix V in the revised paper.
>
> Implementation Details: Each dataset was evenly divided into five parts. In each run, one part was selected as the validation set, while the remaining four parts were used as the training set. The model was trained and evaluated on the validation set. This process was repeated five times to complete all runs.
>
>
>
> Table F4. 5-fold cross-validation results of the SMI-Editor model.
>
> |          | BACE↑ | BBBP↑ | SIDER↑ | Tox21↑ | ToxCast↑ |
> | -------- | ----- | ----- | ------ | ------ | -------- |
> | Run 1    | 91.92 | 97.64 | 62.59  | 83.69  | 75.83    |
> | Run 2    | 91.86 | 96.27 | 66.89  | 84.09  | 73.31    |
> | Run 3    | 90.82 | 98.53 | 62.6   | 84.87  | 73.52    |
> | Run 4    | 91.13 | 98.77 | 63.32  | 83.95  | 74.6     |
> | Run 5    | 90.68 | 97.84 | 63.5   | 85.83  | 75.51    |
> | **Mean** | 91.28 | 97.81 | 63.78  | 84.48  | 74.55    |
> | **Std**  | 0.58  | 0.97  | 1.78   | 0.87   | 1.13     |

---

> ### Author Response · Authors · 2024-11-19
> **Response to Reviewer AGu3 (Part 6)**
>
> ### Reponse to: What is the time cost or complexity of SMILES-Editor compared to the previous MLM?
>
> **A:** Thank you for raising this insightful concern. The use of experts does increase training costs. We measured that the training cost of SMI-Editor is approximately three times that of MLM models (SMI-MLM) for the same model size, training hyperparameters, and data. However, **the training cost of SMI-Editor remains acceptable**. To better analyze the impact of training cost, we trained an MLM model with equivalent computational cost (SMI-MLM(More)). Results showed that SMI-MLM(More) performed worse than the original SMI-MLM and significantly lagged behind SMI-Editor, highlighting that merely increasing MLM training cost does not yield better results. We have also added these resuts to Appendix P in the revised paper. Below is a detailed analysis:
>
> 1. **Training cost comparison:**
>    Table F5 compares the training costs (measured in GPU hours) of SMI-Editor and SMI-MLM models with the same size and hyperparameters. SMI-Editor’s cost is approximately three times that of SMI-MLM.
>
> 2. **Reasons for higher training cost in SMI-Editor:**
>    SMI-Editor requires computing expert actions (using a computationally expensive dynamic programming algorithm) and modeling three different editing operations, which introduces additional overhead.
>
> 3. **Acceptable training cost:**
>    Training SMI-Editor on a dataset with 19M compounds using four RTX 3090 GPUs took approximately 24.6 hours. Scaling SMI-Editor to larger datasets (e.g., 100M+ compounds) is feasible, demonstrating its potential for broader applications.
>
> 4. **SMI-Editor performs better under the same training cost with MLM:**
>    We trained SMI-MLM(More) with the same computational cost as SMI-Editor by increasing its training steps from 120K to 360K. Table F6 shows that SMI-MLM(More) performs worse than the SMI-Editor and  original SMI-MLM . This is due to rapid saturation issues in MLM training on SMILES data. **This also indicates that the speed of model training is not the most important factor; what matters more is whether the model can efficiently extract high-quality semantic representations**. This highlights the importance of designing more powerful training schemes like SMI-Editor to effectively extract meaningful information from SMILES.
>
> 5. **Higher performance ceiling for SMI-Editor:**
>    Although the inclusion of Experts slows down the training speed of the SMI-Editor model, it also enriches the semantic information the model learns. This gives SMI-Editor greater scalability and a higher performance ceiling compared to SMI-MLM. In Table F7, we compare the performance of MLM models and SMI-Editor models of different sizes. The table shows that the performance gains from increasing model size and training costs are more significant for SMI-Editor. This makes SMI-Editor a better choice when given the same and enough training budget.
>
>
>
> Table F5. Training time comparison (RTX 3090 GPUs) between SMI-Editor and SMI-MLM.
>
> |              | SMI-MLM | SMI-Editor |
> | ------------ | ------- | ---------- |
> | GPU Time (h) | 33.2    | 98.35      |
>
>
>
> Table F6. Performance of MLM models with equivalent training costs (SMI-MLM(More)).
>
> |               | BACE↑    | BBBP↑    | SIDER↑   | Tox21↑   | ToxCast↑ | Mean↑    |
> | ------------- | -------- | -------- | -------- | -------- | -------- | -------- |
> | SMI-MLM(More) | 74.3     | 66.2     | 49.5     | 73.3     | 62.3     | 65.1     |
> | SMI-MLM       | 77.8     | 68.6     | 61.2     | 75.1     | 64.9     | 69.5     |
> | SMI-Editor    | **80.3** | **77.4** | **63.0** | **77.1** | **67.4** | **73.0** |
>
>
>
>  Table F7. Performance of SMI-Editor and SMI-MLM models of varying sizes on downstream tasks.
>
> |                   | BACE↑    | BBBP↑    | SIDER↑   | Tox21↑   | ToxCast↑ | Mean↑    |
> | ----------------- | -------- | -------- | -------- | -------- | -------- | -------- |
> | SMI-MLM(Small)    | 76.8     | 69.6     | 60.5     | 75.3     | 64.2     | 69.2     |
> | SMI-MLM(Base)     | 76.6     | 69.3     | 59.9     | 75.3     | 64.4     | 69.1     |
> | SMI-MLM(Big)      | 77.4     | 68.7     | 60.8     | 75.1     | 65.3     | 69.4     |
> | SMI-Editor(Small) | 78.3     | 72.6     | 59.4     | 75.6     | 65.1     | 70.2     |
> | SMI-Editor(Base)  | 79.2     | 73.2     | **61.0** | 75.7     | 65.8     | 71.0     |
> | SMI-Editor(Big)   | **79.3** | **74.2** | 60.9     | **76.7** | **66.4** | **71.5** |

---

> ### Author Response · Authors · 2024-11-19
> **Response to Reviewer AGu3 (Part 7)**
>
> ### Reponse to: In Table 2, the performance of SMI-EDITOR-AtomsMasking in BACE is even better than SMILES-Editor.
>
> **A:** Thank you for your suggestion. We have conducted additional experiments to further compare the performance differences between the two models. To better evaluate the differences between **SMI-EDITOR-AtomsMasking (SMI-EDITOR-AM)** and **SMI-EDITOR**, we assessed their performance across more tasks, as shown in Table F8. The results indicate that while the two models perform **similarly** on the BACE task (80.4 vs. 80.3), there are notable performance differences on the other six tasks (74.6 vs. 77.8). The average performance of the two models also shows a significant gap.
>
> These results further validate the effectiveness of SMI-Editor’s pretraining strategy and demonstrate that SMI-Editor has a clear performance advantage over simply performing atom masking and prediction.
>
>
>
> Table F8. Performance comparison of SMI-EDITOR-AtomsMasking, SMI-EDITOR-AtomsDroping, and SMI-Editor on more downstream tasks.
>
> |               | BACE↑    | BBBP↑    | Tox21↑   | SIDER↑   | MUV↑     | ClinTox↑ | ToxCast↑ | Mean↑    |
> | ------------- | -------- | -------- | -------- | -------- | -------- | -------- | -------- | -------- |
> | SMI-EDITOR-AD | 80.0     | 73.4     | 76.5     | 59.2     | 77.3     | 98.4     | 66.6     | 75.9     |
> | SMI-EDITOR-AM | **80.4** | 73.2     | 75.0     | 58.3     | 73.7     | 97.2     | 64.6     | 74.6     |
> | SMI-MLM       | 77.8     | 68.6     | 75.1     | 61.2     | 75.1     | 89.8     | 64.9     | 73.2     |
> | SMI-Editor    | 80.3     | **77.4** | **77.1** | **63.0** | **80.2** | **98.9** | **67.4** | **77.8** |
>
>
>
>
>
> [1] Liu, Zequn, et al. "MolXPT: Wrapping Molecules with Text for Generative Pre-training." ACL. 2023.
>
> [2] Zhu, Jinhua, et al. "Dual-view molecular pre-training." *Proceedings of the 29th ACM SIGKDD Conference on Knowledge Discovery and Data Mining*. 2023.
>
> [3] Frey, Nathan C., et al. "Neural scaling of deep chemical models." *Nature Machine Intelligence* 5.11 (2023): 1297-1305.
>
> [4] Ross, Jerret, et al. "Large-scale chemical language representations capture molecular structure and properties." *Nature Machine Intelligence* 4.12 (2022): 1256-1264.
>
> [5] Taylor, Ross, et al. "Galactica: A large language model for science." *arXiv preprint arXiv:2211.09085* (2022).
>
> [6] Skinnider, Michael A., et al. "Chemical language models enable navigation in sparsely populated chemical space." *Nature Machine Intelligence* 3.9 (2021): 759-770.
>
> [7] Bagal, Viraj, et al. "MolGPT: molecular generation using a transformer-decoder model." *Journal of Chemical Information and Modeling* 62.9 (2021): 2064-2076.
>
> [8] Fabian, Benedek, et al. "Molecular representation learning with language models and domain-relevant auxiliary tasks." *arXiv preprint arXiv:2011.13230* (2020).
>
> [9] Chithrananda, Seyone, Gabriel Grand, and Bharath Ramsundar. "ChemBERTa: large-scale self-supervised pretraining for molecular property prediction." *arXiv preprint arXiv:2010.09885* (2020).
>
> [10] Schwaller, Philippe, et al. "Molecular transformer: a model for uncertainty-calibrated chemical reaction prediction." *ACS central science* 5.9 (2019): 1572-1583.
>
> [11] Honda, Shion, Shoi Shi, and Hiroki R. Ueda. "Smiles transformer: Pre-trained molecular fingerprint for low data drug discovery." *arXiv preprint arXiv:1911.04738* (2019).
>
> [12] Wang, Sheng, et al. "Smiles-bert: large scale unsupervised pre-training for molecular property prediction." *Proceedings of the 10th ACM international conference on bioinformatics, computational biology and health informatics*. 2019.
>
> [13] Schwaller, Philippe, et al. "“Found in Translation”: predicting outcomes of complex organic chemistry reactions using neural sequence-to-sequence models." *Chemical science* 9.28 (2018): 6091-6098.
>
> [14] Han, Yuqiang, et al. "Retrosynthesis prediction with an iterative string editing model." *Nature Communications* 15.1 (2024): 6404.
>
> [15] Hayes, Tomas, et al. "Simulating 500 million years of evolution with a language model." *bioRxiv* (2024): 2024-07.
>
> [16] Wang, Xinyou, et al. "Diffusion Language Models Are Versatile Protein Learners." *arXiv preprint arXiv:2402.18567* (2024).
>
> [17] Wang, Xinyou, et al. "DPLM-2: A Multimodal Diffusion Protein Language Model." *arXiv preprint arXiv:2410.13782* (2024).
>
> [18] Cui, Haotian, et al. "scGPT: toward building a foundation model for single-cell multi-omics using generative AI." *Nature Methods* (2024): 1-11.
>
> [19] Zhou, Gengmo, et al. "Uni-mol: A universal 3d molecular representation learning framework." (2023).
>
> [20] Luo, Shengjie, et al. "One transformer can understand both 2d & 3d molecular data." *The Eleventh International Conference on Learning Representations*. 2022.

---

> ### Author Response · Authors · 2024-11-21
> **Seeking more discussion with Reviewer AGu3**
>
> Dear Reviewer AGu3,
>
> We are grateful for your insightful feedback, which has been instrumental in improving our manuscript. We have also provided an additional summary here to further facilitate our discussion.
>
> First, in response to your questions regarding SPE tokenizers and related methods, we analyzed the limitations of SPE tokenizers and the differences between them and our model. This analysis demonstrated that exploring the introduction of fragment-level supervision signals to models while maintaining atomic-level representations is a highly meaningful research direction. We also trained an MLM model based on SPE tokenizers, and the experimental results showed that MLM with SPE did not achieve performance improvements, further highlighting the fundamental differences between this model and SMI-Editor.
>
> For the comparison with the Masked Span Language Model, we trained an MLM model based on this approach, and the results indicated that the Masked Span Language Model is not suitable for modeling SMILES data. We provided a detailed analysis to discuss these results.
>
> Regarding your concerns about experimental results, we presented SMI-Editor’s performance on DeepChem downstream task datasets, demonstrating that SMI-Editor outperforms baseline models with different pre-training strategies. For the comparison with autoregressive pre-training strategies, we trained a SMILES model with a fully consistent setup using an autoregressive pre-training approach and compared its performance with SMI-Editor, showing that SMI-Editor has superior representation learning capabilities.
>
> Additionally, we extended SMI-Editor’s training strategy to Encoder-Decoder models and applied it to generative tasks. The model's performance demonstrated the significant application potential of the SMI-Editor pre-training strategy for generative tasks. We also provided k-fold experimental results of SMI-Editor on various downstream tasks, which confirmed SMI-Editor’s robustness and stability in downstream tasks.
>
> Regarding the computational cost of SMI-Editor, we compared the training time costs of SMI-Editor and SMI-MLM under the same training settings.  Additionally, we present the performance of both models on downstream tasks with equivalent computational costs. Finally, we evaluated SMI-Editor-AtomsMasking on all downstream tasks and demonstrated that SMI-Editor achieves consistent performance improvements over SMI-Editor-AtomsMasking.
>
> **Please let us know if we have adequately addressed your concerns, and we are always open to further suggestions.**
>
> Thank you for your time and consideration.
>
> Authors

---

> ### Author Response · Authors · 2024-11-23
> **Looking Forward to Your Valuable Feedback**
>
> We greatly appreciate your time and effort in reviewing our paper, as your feedback is incredibly valuable in helping us improve the quality of our paper and also crucial to the evaluation process. We have carefully addressed all the concerns and comments you raised. As the deadline for this discussion is approaching, we kindly request you to prioritize submitting your feedback at your earliest convenience. Thank you again for your valuable suggestions. We look forward to any further discussions.

---

> ### Author Response · Authors · 2024-11-24
> **Eagerly seeking more discussion. We need your valuable feedback !**
>
> Thank you for providing us with valuable suggestions. Considering that only **two days** remain until the end of the discussion period and we have yet to receive any feedback, we would like to confirm once again whether our response has addressed all your concerns. We welcome any further discussion.
>
> As one of the reviewers for this year's ICLR conference as well, I deeply understand that reading others' papers and providing suggestions is a time-consuming and arduous task. Therefore, I sincerely thank every reviewer who thoughtfully provided feedback on my paper; it is your suggestion that helped make my work better. At the same time, as one of the authors submitting to this conference, I also deeply understand the hard work and dedication that every author puts into preparing their manuscripts and responding to each question raised by the reviewers. Therefore, I will take responsibility and actively engage with the author discussions for every paper I review.  **The paper you have read and the responses we have provided are the result of our team members' utmost effort, dedication, and hard work. It holds great significance for us, and we simply hope to be treated fairly.**
>
> Thanks for your understanding. Wishing you all the best !

---

> ### Comment · Reviewer_AGu3 · 2024-11-25
>
> Thanks for your rebuttal. I still have the following concerns after reading your responses.
> * The prior is too strong for the learning of SMILES. Compared Masked Span Language Modeling, the proposed method may help learn better representation of SMILES strings, but it can not help LMs learn the grammar of SMILES and thus fail to generalize to the generation tasks of molecules. On the other side, if you ultimately want to augment the SMILES representation, graph is way more better. It naturally shows the connections between different substructures. For example, there are works like Fragment-based Pretraining and Finetuning on Molecular Graphs [1]. Ultimately, this work is still a variant of Masked Span Language Modeling, I don't see the workload or novelty is solid enough for qualifying acceptance.
> * Although the authors claim their methods as SOTA, the actual performance is not the case. SMI-Editor achieves the highest everage performance, but fails to outperform several previous baselines in several downstream tasks. Works like MolCA [2] seem to perform better in many downstream tasks. Meanwhile, I am quite curious that why the authors did not provide the 5-fold results of Clintox and MUV?
> * I am still not sure why the authors conduct experiments on MoleculeNet by choosing the best experiment results initially, instead following the k-fold experiment settings. Although they did conduct 5-fold experiments in their rebuttal, they did not align their experiment settings with the previous baselines.
> * The training cost we saw from the response is unacceptable and seems to make the method unable to scale up.
> * It makes me feel more weird that there are two reviewers give a rating of 8, but they did not raise any of the concerns at all?
>
> Given the above reasons, I will keep my ratings.
>
>
> ## **References**
> [1] Luong, K. D., & Singh, A. K. (2024). Fragment-based pretraining and finetuning on molecular graphs. Advances in Neural Information Processing Systems, 36.
>
> [2] Liu, Z., Li, S., Luo, Y., Fei, H., Cao, Y., Kawaguchi, K., ... & Chua, T. S. (2023). MolCA: Molecular graph-language modeling with cross-modal projector and uni-modal adapter. arXiv preprint arXiv:2310.12798.

---

> > ### Author Response · Authors · 2024-11-25
> > **Response to Reviewer AGu3 (Part 8)**
> >
> > Thank you very much for your response; it is very valuable in helping us further improve our paper. Regarding the questions and concerns you just raised, we have provided our responses below, and we look forward to further discussions.
> >
> > ---
> >
> > ### Response to: This method can not help LMs learn the grammar of SMILES and thus fail to generalize to the generation tasks of molecules.
> >
> > Thank you for your interesting question. Our response to this point is as follows:
> >
> > - **The model learns SMILES grammar during the process of reconstructing incomplete SMILES into complete and valid SMILES:** The core of SMI-Editor's training process is to restore SMILES with missing fragments back to the original complete SMILES. In this process, the supervision signals provided to the model require it to correctly reconstruct the original and grammar-valid SMILES. As the model's ultimate generation targets during training are valid and follow SMILES grammar, the model is able to capture grammatical information inherent to SMILES.
> >
> > - **SMI-Editor performs well on generation tasks**: As shown in the results of Table F3, SMI-Editor achieves strong performance on the molecular retrosynthesis prediction task. The core process of molecular retrosynthesis prediction involves requiring the model to correctly generate reactants that can produce the target molecule. This task is an excellent way to test the model's understanding of molecular grammar and its molecule generation capability. SMI-Editor's strong performance on this task demonstrates that the model can accurately generate outputs that comply with SMILES grammar and can be effectively applied to molecule generation tasks.
> >
> > ---
> >
> >
> > ### Response to: If you ultimately want to augment the SMILES representation, graph is way more better.
> >
> > Thank you for your question. Our response to this point is as follows:
> >
> > - **SMILES and molecular graphs are two different representations of molecular information, but they contain the same amount of information**: Although SMILES and molecular graphs differ in their forms of expression, they hold equivalent information. A SMILES representation can be losslessly converted into its corresponding molecular graph representation. From this perspective, modeling SMILES can enable models to learn the same knowledge as modeling molecular graphs.
> >
> > - **SMILES modeling is a crucial research direction**: As mentioned earlier, there is currently a large body of work focusing on SMILES modeling [1-13], and SMILES is widely used as a molecular data representation in numerous biological and chemical applications[21, 22]. Therefore, studying better SMILES modeling methods is a highly meaningful research problem. This work introduces fragment-level supervision into SMILES modeling, which addresses a significant gap in the current field of SMILES language modeling.
> >
> > - **Our approach is significantly different from introducing fragment-level supervision signals into graphs**: Although SMILES and molecular graphs contain equivalent information, their different data formats lead to different modeling challenges. This makes the edit-based generation training method of SMI-Editor difficult to directly transfer to graphs. This is because SMI-Editor requires the model to recover the information of deleted fragments. However, deleting fragments in a graph can severely disrupt its structure, making it ill-posed (e.g., resulting in a disconnected graph after fragment removal), which makes it difficult for GNN models to handle such information. This is also an advantage of SMILES models. Even if the resulting molecular graph becomes disconnected after deletion, we can still concatenate the SMILES strings of the resulting molecular fragments and input them into the model, requiring the model to correctly recover and assemble these fragments.
> >
> >
> >
> > [21] M Veselinovic, Aleksandar, et al. "Application of SMILES notation based optimal descriptors in drug discovery and design." *Current topics in medicinal chemistry* 15.18 (2015): 1768-1779.
> >
> > [22] Hanson, Robert M. "Jmol SMILES and Jmol SMARTS: specifications and applications." *Journal of Cheminformatics* 8 (2016): 1-20.

---

> ### Author Response · Authors · 2024-11-25
> **Response to Reviewer AGu3 (Part 9)**
>
> ### Response to: This work is still a variant of Masked Span Language Modeling, I don't see the workload or novelty is solid enough.
>
> Thank you for your question. Our response to this issue is as follows:
>
> - **SMI-Editor and Masked Span Language Modeling differ in three key aspects**: First, SMI-Editor uses fragment deletion instead of fragment masking, which prevents the issue of topological information leakage in SMILES. Second, SMI-Editor better defines what constitutes chemically meaningful fragments in SMILES. Third, SMI-Editor employs a fundamentally different modeling approach from traditional Masked Span Language Modeling, utilizing an edit-based modeling approach. This enables the model to better capture the detailed differences between the input and target output (e.g., missing molecular fragments).
> - **Our experimental results also demonstrate that the SMI-Editor model achieves significant performance improvements compared to Masked Span Language Modeling** (shown in Table F1, part 2). This further confirms that SMI-Editor is a more suitable method for modeling semantic information at the molecular fragment level, rather than merely being a variant of Masked Span Language Modeling.
>
> Based on the above results, there are significant differences between SMI-Editor and Masked Span Language Modeling. Whether in model design, pretraining methods, or downstream task performance, the two models exhibit clear distinctions.
>
> ---
>
>
> ### Response to: MolCA seems to perform better in many downstream tasks. Meanwhile, I am quite curious that why the authors did not provide the 5-fold results of Clintox and MUV?
>
> Thank you for your question. Our response to this issue is as follows:
>
> - **The MolCA model leverages textual data during pre-training, giving it an unfair comparative advantage**: The test data for the MoleculeNet benchmark is collected from past academic literature. MolCA, however, uses Galactica which is pre-trained on a large amount of scientific literature as its backbone, making it easier for such models to extract textual knowledge relevant to downstream test tasks from their pre-training data. This potentially leads to issues of test data leakage. So the comparison between SMI-Editor and MolCA is not fair.
>
> - **We are currently conducting 5-fold testing of the model on the Clintox and MUV datasets, and the results will be included in the final version of the manuscript:** The reason we initially did not report the 5-fold results for Clintox and MUV datasets is that 5-fold testing is relatively time-consuming. Given that we added many new experimental results during the Rebuttal stage, computational resources were extremely limited. As a result, we prioritized running 5-fold tests on the five tasks originally selected for analysis in the paper. The 5-fold results for Clintox and MUV will be added to the manuscript later.
>
> P.S.: After approximately 12 hours of testing, we have obtained the 5-fold experimental results of the SMI-Editor model on the ClinTox and MUV datasets, as shown in Table F9. The results demonstrate that SMI-Editor also exhibits good performance stability on these datasets. Additionally, we have included this information in Appendix V of the paper.
>
> Table F9. 5-fold cross-validation results of the SMI-Editor model.
>
> |          | BACE↑ | BBBP↑ | SIDER↑ | Tox21↑ | ToxCast↑ | **ClinTox↑** | **MUV↑** |
> | -------- | ----- | ----- | ------ | ------ | -------- | ------------ | -------- |
> | Run 1    | 91.92 | 97.64 | 62.59  | 83.69  | 75.83    | 99.76        | 77.23    |
> | Run 2    | 91.86 | 96.27 | 66.89  | 84.09  | 73.31    | 98.73        | 79.49    |
> | Run 3    | 90.82 | 98.53 | 62.6   | 84.87  | 73.52    | 99.6         | 77.4     |
> | Run 4    | 91.13 | 98.77 | 63.32  | 83.95  | 74.6     | 99.82        | 77.58    |
> | Run 5    | 90.68 | 97.84 | 63.5   | 85.83  | 75.51    | 98.61        | 79.39    |
> | **Mean** | 91.28 | 97.81 | 63.78  | 84.48  | 74.55    | 99.30        | 78.23    |
> | **Std**  | 0.58  | 0.97  | 1.78   | 0.87   | 1.13     | 0.59         | 1.12     |

---

> ### Author Response · Authors · 2024-11-25
> **Response to Reviewer AGu3 (Part 10)**
>
> ### Response to: I am still not sure why the authors conduct experiments on MoleculeNet by choosing the best experiment results initially. They did not align their experiment settings with the previous baselines.
>
> We adopted the exact same testing setting as the baseline models for downstream tasks, running each task three times with three different random seeds and then calculating the average. Additionally, we have provided the variance of our model on these tasks. And this information will also be included in the final version of our paper. Below are our experimental results, which demonstrate that SMI-Editor exhibits good stability in performance:
>
> Table F10. Performance of the SMI-Editor model.
> |            | BBBP↑        | Tox21↑       | ClinTox↑     | HIV↑         | BACE↑        | SIDER↑       | Mean↑    |
> | ---------- | ------------ | ------------ | ------------ | ------------ | ------------ | ------------ | -------- |
> | GEM        | 72.4±0.4     | 78.1±0.1     | 90.1±1.3     | 80.6±0.9     | 85.6±1.1     | 67.2±0.4     | 79.0     |
> | ChemBerta  | 64.3         | -            | 90.6         | 62.2         | -            | -            | -        |
> | MolCLR     | 73.6±0.5     | 79.8±0.7     | 93.2±1.7     | 80.6±1.1     | 89.0±0.3     | 68.0±1.1     | 80.7     |
> | MGSSL      | 70.5±1.1     | 76.5±0.4     | 80.7±2.2     | 79.5±1.1     | 79.7±0.8     | 61.8±0.7     | 74.8     |
> | DMP_{TF}   | 78.1±0.5     | 78.8±0.5     | 95.0±0.5     | 81.0±0.7     | 89.3±0.9     | 69.2±0.7     | 81.9     |
> | Galactica  | 66.1         | 68.9         | 82.6         | 74.5         | 61.7         | 63.2         | 69.5     |
> | MoMu       | 70.5±2.0     | 75.6±0.3     | 79.9±4.1     | 76.2±0.9     | 77.1±1.4     | 60.5±0.9     | 73.3     |
> | SMI-MLM    | 89.4±1.9     | 76.2±1.6     | 90.6±1.8     | 79.8±1.2     | 86.6±0.4     | 66.5±0.5     | 81.5     |
> | SMI-Editor | **93.5±2.2** | **81.4±1.1** | **95.2±1.3** | **81.6±0.7** | **89.9±0.2** | **69.8±0.6** | **85.2** |
>
> ---
>
> ### Response to: The training cost we saw from the response is unacceptable and seems to make the method unable to scale up.
>
> Thank you very much for your valuable suggestions. In fact, **the primary metric for defining the scalability of a model should be whether its performance improves significantly when more computational resources are provided** (e.g., training a larger model), rather than just focusing on the absolute amount of computational cost. From this perspective, SMI-Editor demonstrates better scalability compared to MLM. This means that when we allocate more computational resources to the SMI-Editor model, its performance improves significantly, whereas MLM struggles to achieve the same. Therefore, scaling up the SMI-Editor model is not only more meaningful but also holds greater potential, while MLM, due to the inherent limitations of its training objective, finds it harder to scale effectively. Moreover, the training cost of the SMI-Editor model is entirely acceptable. The current version of the SMI-Editor model requires only about one day of training on four 3090 GPUs. Compared to the well-established practices in the model scaling field, this training cost is entirely reasonable.
>
> ---
>
> Thank you once again for your suggestions and questions regarding our paper, as well as for your time. We also welcome any further discussions.

---

> ### Author Response · Authors · 2024-11-27
>
> Dear Reviewer AGu3,
>
> Hi, thank you for taking the time to provide feedback on our manuscript. We deeply appreciate your insights, and we have tried our best to address your concerns in our response. Please kindly check it out.
>
> As the final deadline for discussion is approaching, we would be very grateful for any further feedback you might have. Please don’t hesitate to reach out if you have additional questions—we’d be happy to provide further clarifications!
>
> Looking forward to hearing from you, and many many thanks!
>
> Best,
>
> Authors

---

> ### Author Response · Authors · 2024-12-02
> **Kindly Request for Your Feedback Before Upcoming Discussion Deadline**
>
> Dear Reviewer AGu3,
>
> Thank you so much for taking the time to review our manuscript and for providing your valuable guidance! And we have tried our best to address all your concerns in our previous response. As the discussion deadline is just **one day** away, we are very eager to hear your feedback or suggestions so that we can make any necessary improvements promptly. If you have any thoughts or a convenient time to discuss, please feel free to let us know—we are happy to engage in further discussion on any issues you may have.
>
> Thank you again for your support and help!
>
> Best,
> Authors

---

### Official Review · Reviewer_Rnmw · 2024-10-28

**Soundness:** 2
**Presentation:** 3
**Contribution:** 2
**Rating:** 6
**Confidence:** 4

**Summary:**

The paper introduces SMI-Editor, a novel edit-based SMILES language model with fragment-level supervision to improve molecular representation learning. SMI-Editor tries to address the limitations of masked language models (MLMs), such as rapid saturation and limited substructure semantics modeling, by using an LevT-based modeling.

**Strengths:**

- The analysis identifies interesting issues: the MLM model struggles to distinguish between random deletion and hydrophilic deletion, and it quickly saturates on single-token masking.
- The paper writing is clear and well-illustrated.

**Weaknesses:**

- LevT isn’t actually an MLM model; it’s a non-autoregressive generative model. From this perspective, the authors' solution for handling corrupted SMILES input is somewhat trivial, as other sequence-based generative models could also address this.
- There is a discrepancy between the motivation and solution. Neither corrupted SMILES nor fragment-level supervision is directly related to the way LevT models molecules.
- Limited effectiveness. While I understand the authors' choice of the MoleculeNet baseline without considering powerful graph-based models like Uni-Mol, DVMP_{TF} in Jinhua Zhu et al. is a SMILES-based Transformer encoder and appears to perform much better than the reported results.
- Figure 5 is confusing; if the model can distinguish hydrophilic groups, wouldn’t performance degrade noticeably after HG deletion?

**Questions:**

- Why not include the performance on FreeSolv and ESOL?
- What about applying MLM on fragment spans? This could also help alleviate the saturation problem in single-token prediction.

---

> ### Author Response · Authors · 2024-11-19
> **Response to Reviewer Rnmw (Part 1)**
>
> We appreciate the insightful suggestions from Reviewer Rnmw. In the following sections, we will address all your concerns regarding motivation, effectiveness of our model, explanations of analytical experiments, performance on ESOL and FreeSolv tasks, MLM with fragment spans. And we have also supplemented new experimental results(further explanation of the model's motivation, performance comparison on more datasets and with more baseline models, more detailed analytical experiments, results of SMI-Editor on ESOL and FreeSolv tasks, the performance of MLM with fragment spans.) to address some of the issues you have raised. We hope the replies can make our paper more clear. Any further comments are welcome!

---

> ### Author Response · Authors · 2024-11-19
> **Response to Reviewer Rnmw (Part 2)**
>
> ### Reponse to：What is the motivation for using the LevT model? What are its advantages?
>
> **A:** Thank you for raising this question, which helps clarify the paper’s structure. Your concerns can be summarized as follows:
>
> 1. The solution for handling corrupted SMILES input appears somewhat trivial, as other sequence-based generative models could also address this.
> 2. There is a disconnect between the stated motivation and the solution. Neither corrupted SMILES nor fragment-level supervision directly relates to how LevT models molecules.
>
> Therefore, we will address these two questions to explain why we chose the LevT model and what advantages this type of model offers compared to others.
>
> #### Part 1: Advantages of SMI-Editor
>
> We chose an edit-based model like LevT as the core architecture because **its design aligns perfectly with our goal of effectively modeling molecular substructure information**. The advantages of this approach are as follows:
>
> 1. **SMI-Editor focuses on identifying and correcting errors, making it well-suited for restoring corrupted SMILES.**
>    The pretraining task involves transforming corrupted SMILES into their correct forms, where the input and target sequences share high similarity. This means the model only needs to identify and fix minor errors (e.g., deleting or inserting specific tokens) without recreating the entire sequence.
>    - Unlike autoregressive models, which attempt to generate every token (even all the correct ones) from scratch, SMI-Editor only learns the necessary edits. This makes it more sensitive to subtle errors in the input sequence, which is crucial for effectively modeling molecular substructures.
>
> 2. **The edit-based training approach enables the model to better capture fragment-level semantics.**
>    SMI-Editor learns from corrupted SMILES that lack any explicit information about the positions and lengths of missing fragments. This forces the model to infer richer semantic information.
>    - Ablation experiments (Table 3 in the paper) show that predicting the locations and lengths of missing fragments (placeholder insertion) significantly enhances the model's understanding of molecular semantics. This capability stems from the flexible generative ability of edit-based models, which many other sequence generation approaches lack.
>    - For instance, models like non-autoregressive CMLM[1] (or Mask-Predict) require a predefined target sequence length during decoding. This leakage of structural information limits their performance.
>
> #### Part 2: Comparison with other sequence generation models
>
> To further demonstrate the advantages of SMI-Editor, we compared its performance against several sequence generation models:
>
> 1. **Comparison with autoregressive language models (SMI-GPT):**
>    We trained an autoregressive language models SMI-GPT with the same size and architecture as SMI-Editor. The results (General Response, Table A2) show that SMI-Editor significantly outperforms SMI-GPT across downstream tasks.
>    - **Reason for the difference:** As explained in the General Response, SMI-GPT use all preceding tokens as context for predicting the next token. This makes it easier to predict tokens at later positions due to richer context, resulting in faster convergence but limited ability to extract richer information.
>
> 2. **Comparison with non-autoregressive sequence generation models:**
>    We trained a Masked Span Language Model (SMI-MLM(SPAN)) that works similarly to classical non-autoregressive models like CMLM[1]. Table E1 shows that SMI-Editor outperforms SMI-MLM(SPAN) across all tasks.
>    - **Reason for the difference:** CMLM requires predefined masked regions (including the positions and lengths of these regions) during training, leading to the leakage of topological information about the fragments' locations and sizes. This makes the task easier and limits the semantic richness learned by the model.
>
> In summary, SMI-Editor improves representation learning, mitigates rapid saturation issues, and extracts richer substructure semantics from SMILES, providing distinct advantages over other pretraining approaches. The extensive experimental results demonstrate its effectiveness, making it a novel and suitable pretraining model for SMILES.
>
> [1] Ghazvininejad, Marjan, et al. "Mask-predict: Parallel decoding of conditional masked language models." *arXiv preprint arXiv:1904.09324* (2019).
>
> Table E1. Performance comparison among SMI-MLM(SPAN), SMI-MLM(Frag), and SMI-Editor.
>
> |    | BACE↑    | BBBP↑    | SIDER↑   | Tox21↑   | ToxCast↑ | Mean↑    |
> | ------- | ---- | ---- | -------- | -------- | -------- | -------- |
> | SMI-MLM       | 77.8     | 68.6     | 61.2     | 75.1     | 64.9     | 69.5     |
> | SMI-MLM(SPAN) | 78.6     | 67.2     | 59.4     | 76.1     | 62.3     | 68.7     |
> | SMI-MLM(Frag) | 79.4     | 73.3     | 62.1     | 74.0     | 64.8     | 70.7     |
> | SMI-Editor    | **80.3** | **77.4** | **63.0** | **77.1** | **67.4** | **73.0** |

---

> ### Author Response · Authors · 2024-11-19
> **Response to Reviewer Rnmw (Part 3)**
>
> ### Reponse to: Figure 5 (Figure 4 in the revised draft) is confusing; If the model can distinguish hydrophilic groups, wouldn’t performance degrade noticeably after HG deletion?
>
> **A:** Thank you for this question! Regarding FreeSolv, the observed trends align with physical properties as expected. In contrast, ESOL results are influenced by molecular weight and other factors. **Further analyses confirm that SMI-Editor's behavior is consistent with expectations**. We have also added these resuts to Appendix Q in the revised paper. Below are detailed explanations:
>
> 1. **For the FreeSolv dataset, the observed trends align with its physical property definitions.** FreeSolv reflects the hydration free energy of compounds, defined as the free energy change when a compound transitions from a non-dissolved state to a dissolved state. When hydrophilic groups in a molecule are reduced, the change in hydration free energy increases, leading to higher hydration free energy. Thus, when we remove hydrophilic groups from the molecule, the model predicts an increase in hydration free energy, consistent with the trend observed in Figure 5(b), which matches our expectations.
>
> 2. **For the ESOL task, the model predictions are significantly influenced by molecular weight.** The ESOL dataset reflects compound solubility, which is strongly negatively correlated with molecular weight: the larger the molecular weight, the lower the solubility. We plotted a scatter diagram (figure line: [here](https://anonymous.4open.science/r/ICLR25-21E2/esol_vis.pdf), or the Figure 14 in our revised draft) showing the relationship between molecular weight and solubility in the ESOL training set. A clear negative correlation with a coefficient of \( R = -0.69 \) is observed. Consequently, when functional groups or atoms are removed from a molecule, its molecular weight decreases, leading the model to predict an increase in solubility. **This explains why, in Figure 5(a), the model predicts increased solubility regardless of whether hydrophilic groups or random groups are removed**. The increase is more significant with random deletions, demonstrating the model's ability to distinguish between hydrophilic group deletions and random deletions.
>
> 3. **To eliminate the influence of molecular weight, we designed a hydrophilic group replacement scheme (HG Rep).** We replaced all hydrophilic groups in a molecule with non-hydrophilic groups of similar molecular weight (e.g., methyl, ethyl, propyl) and compared this hydrophilic group replacement scheme (HG Rep) with a random group replacement scheme (Rand Rep), where random groups were replaced with others of similar weight. The results, shown in Figure 15 in our revised draft (or used links to the figure: [SMI-Editor](https://anonymous.4open.science/r/ICLR25-21E2/SMI_Editor_ESOL_Rep.pdf), [SMI-MLM](https://anonymous.4open.science/r/ICLR25-21E2/MLM_ESOL_Rep.pdf)), reveal that **SMI-Editor effectively distinguishes between HG Rep and Rand Rep, demonstrating its ability to model key molecular group semantics.** It also correctly predicts that replacing hydrophilic groups reduces molecular solubility.
>
> Furthermore, we plotted the distribution of predicted changes for MLM models and auto-regressive language models (Auto-regressive LM) before and after these replacement operations (link to the figure: [here](https://anonymous.4open.science/r/ICLR25-21E2/AR_ESOL_Rep.pdf), or Figure 16 in our revised draft). **The results show that these models perform significantly worse than the SMI-Editor in distinguishing between random replacements and hydrophilic group replacements.** This further highlights the superiority of the SMI-Editor in modeling the semantics of molecular substructures.
>
> ---
>
> ### Reponse to: DVMP_{TF} in Jinhua Zhu et al. is a SMILES-based Transformer encoder and appears to perform much better than the reported results.
>
> A: Thank you for raising this insightful concern. In fact, the reason we did not include a comparison with the DMP\_{TF} model in our previously submitted paper is that the test data split used by the DMP\_{TF} model differs from ours. The DMP\_{TF} model followed the DeepChem split, while our model used the Uni-Mol split. We have now evaluated the performance of  SMI-Editor on the DeepChem split test data. The detailed results can be found in General Response Table A1. From these results, it can be observed that under the same dataset split, SMI-Editor outperforms DMP\_{TF} on all tasks. Furthermore, **SMI-Editor achieved the best average performance among all baseline models, showing significant improvement compared to them.** This demonstrates the immense potential of SMI-Editor in terms of performance.

---

> ### Author Response · Authors · 2024-11-19
> **Response to Reviewer Rnmw (Part 4)**
>
> ### Reponse to: Why not include performance on FreeSolv and ESOL?
>
> **A:** Thank you for the suggestion! We have added these results in Table E2. **SMI-Editor achieves state-of-the-art performance on these molecular property regression tasks, significantly outperforming SMI-MLM.** These results align with the visualization analyses in the paper. We have also added these resuts to Appendix M in the revised paper.
>
>
>
> Table E2. Performance of SMI-Editor on molecular property regression tasks.
>
> |            | ESOL↓     | FreeSolv↓ | Lipo↓     |
> | ---------- | --------- | --------- | --------- |
> | MPNN       | 0.58      | 1.150     | 0.7190    |
> | DMP_{TF}   | 0.700     | -         | -         |
> | A-FP       | 0.503     | 0.736     | 0.578     |
> | SMI-MLM    | 0.576     | 0.709     | 0.642     |
> | SMI-Editor | **0.362** | **0.524** | **0.565** |

---

> ### Author Response · Authors · 2024-11-19
> **Response to Reviewer Rnmw (Part 5)**
>
> ### Reponse to: What about applying MLM on fragment spans? This could also help alleviate the saturation problem in single-token prediction.
>
> Thank you for your valuable suggestions. We implemented two different approaches for applying MLM on fragment spans (random span selection and fragment-based span selection) and conducted experiments to compare the performance of SMI-Editor with these models. The results demonstrate that **SMI-Editor’s pretraining strategy allows the model to capture molecular substructure-related information more effectively**. We have also added these resuts to Appendix T in the revised paper. Here are more details.
>
> - **Random Span Masking: Exploring the Performance of Masked Span Language Model (MSLM) on SMILES**
>   - We trained a SMILES model using a Masked Span Language Model, referred to as **SMI-MLM(SPAN)**, with identical model size and hyperparameters to SMI-Editor. This model randomly masks a continuous sequence in the SMILES string and requires the model to predict the missing portion (similar to SpanBERT). The performance of SMI-MLM(SPAN) is presented in Table E1, which shows that **its performance is similar to SMI-MLM but significantly inferior to SMI-Editor**. This further validates the advantages of SMI-Editor over traditional MSLM.
>
>   - **Reasons for the Poor Performance of Traditional MSLM:**
>     - **Differences between Text Data and SMILES Data:**
>       Unlike text, molecular data has complex topological structures. In text, adjacent tokens often have strong semantic relevance, and continuous spans convey related information, making span masking effective for learning local semantics. However, **SMILES lacks such locality**; a single functional group may not appear contiguous, and adjacent tokens may lack strong relevance. For example, aromatic rings with multiple substituents often appear discontinuous in SMILES (we provide a specific case **CASE1** in part 6). This limits the effectiveness of applying span masking directly to SMILES data.
>
>     - **Traditional MSLM (e.g., T5) and SMI-Editor Have Different Implementations; Traditional MSLM is Unsuitable for SMILES Data:**
>       Text data's semantic continuity enables models like T5 to use random span masking, where continuous text segments are masked for prediction. In contrast, SMILES lacks this continuity, so SMI-Editor uses a fragmentation algorithm to split molecules into chemically meaningful fragments. The model predicts missing fragments, which may not correspond to continuous SMILES segments. Unlike traditional MSLM, SMI-Editor focuses on masking chemically significant fragments, a key difference in its design.
>
>   - **Summary:**
>     MSLM differs from SMI-Editor in motivation, implementation, and experimental results. These findings confirm that SMI-Editor is not merely a variant of MSLM on fragment spans but a novel training approach tailored to SMILES data.

---

> ### Author Response · Authors · 2024-11-19
> **Response to Reviewer Rnmw (Part 6)**
>
> - **Fragment-Based Span Masking: Exploring the Performance of Fragment-Level Supervised SMILES MLM**
>
>   - We trained a fragment-level supervised SMILES MLM model, referred to as **SMI-MLM(Frag)**, with identical model size and hyperparameters to SMI-Editor. This model masks all atoms and bonds within a selected molecular substructure during training. The molecular fragmentation scheme used is identical to SMI-Editor for comparability. The performance of SMI-MLM(Frag) is shown in Table E1, where it outperforms SMI-MLM and SMI-MLM(SPAN) but still lags behind SMI-Editor.
>
>   - **Analysis of Results:**
>     - **SMI-MLM(Frag) Captures Richer Molecular Substructure Semantics:**
>       The introduction of supervision signals related to molecular substructures during training allows SMI-MLM(Frag) to outperform models that cannot directly capture such information, like SMI-MLM and SMI-MLM(SPAN).
>
>     - **SMI-MLM(Frag) Suffers from Severe Molecular Topology Information Leakage:**
>       Unlike SMI-Editor, SMI-MLM(Frag) retains the positions and sizes of masked substructures in its input, making the pretraining task simpler and limiting its performance compared to SMI-Editor.
>
>   - **Summary:**
>     SMI-Editor significantly outperforms pretraining approaches based on MLM with span information, highlighting that SMI-Editor is not a variant of MLM on fragment spans but a fundamentally new approach for SMILES modeling.
>
>
>
> **CASE1** When does SMILES exhibit discontinuity: SMILES is a linearized representation of graph-structured molecules, which inherently causes discrepancies between molecular topology and sequence-level representation. For example, when a ring contains multiple substituents, its representation in SMILES often becomes discontinuous. Consider **Glibenclamide**, a drug used for diabetes treatment, with the canonical SMILES:  COc1ccc(Cl)cc1C(=O)NCC**c2ccc**(S(=O)(=O)NC(=O)NC3CCCCC3)**cc2**. Here, the bolded atoms originate from the same aromatic ring, but due to the multiple substituents, this ring is represented discontinuously in SMILES. Additionally, the aromatic carbon **cc2** is adjacent to CCC3 atoms from a distant cycloalkane ring. Such discontinuities are common in SMILES and adversely affect Masked Span Language Models.
>
>
>
> Table E1. Performance comparison among SMI-MLM(SPAN), SMI-MLM(Frag), and SMI-Editor.
>
> |               | BACE↑    | BBBP↑    | SIDER↑   | Tox21↑   | ToxCast↑ | Mean↑    |
> | ------------- | -------- | -------- | -------- | -------- | -------- | -------- |
> | SMI-MLM       | 77.8     | 68.6     | 61.2     | 75.1     | 64.9     | 69.5     |
> | SMI-MLM(SPAN) | 78.6     | 67.2     | 59.4     | 76.1     | 62.3     | 68.7     |
> | SMI-MLM(Frag) | 79.4     | 73.3     | 62.1     | 74.0     | 64.8     | 70.7     |
> | SMI-Editor    | **80.3** | **77.4** | **63.0** | **77.1** | **67.4** | **73.0** |

---

> > ### Comment · Reviewer_Rnmw · 2024-11-25
> > **Response to Reviewer's Rebuttal**
> >
> > Thank you for the detailed analysis and thorough discussion, which have addressed most of my concerns and furthered my understanding of the biochemical aspects of this work. While I still maintain my initial perspective regarding the method's design—particularly the treatment of LevT as a sequence-based model—I appreciate the authors' comprehensive evidence and analysis demonstrating SMI-Editor’s effectiveness in capturing the semantics of functional groups. These efforts underscore the paper's contributions and its value to the community. As a result, I have raised my score to 6, reflecting my belief that this work is deserving of acceptance.

---

> ### Author Response · Authors · 2024-11-21
> **Seeking more discussion with Reviewer Rnmw**
>
> Dear Reviewer Rnmw,
>
> We are grateful for your insightful feedback, which has been instrumental in improving our manuscript. We have also provided an additional summary here to further facilitate our discussion.
>
> First, in response to your concerns about the motivation behind the model, we analyzed the advantages and disadvantages of the SMI-Editor model compared to other types of sequence generation models. Additionally, we demonstrated the advantages and potential of this edit-based model through performance comparisons with both autoregressive and non-autoregressive sequence generation models.
>
> Regarding the issue with Figure 5 in the paper, we analyzed the reasons behind the model's behavior and provided further detailed analyses to illustrate the consistency between the SMI-Editor model's performance on these two types of data and the underlying physical principles.
>
> In response to the comparison with baseline models such as  DVMP_{TF}, we presented the performance of SMI-Editor on DeepChem downstream task datasets, showing that SMI-Editor outperforms baseline models, including DVMP_{TF}. We also provided the model's performance on regression tasks such as FreeSolv and ESOL, with results indicating that SMI-Editor performs well on these regression tasks.
>
> Finally, regarding the exploration of applying MLM on fragment spans, we trained two different MLM models with fragment span supervision signals and compared their performance on downstream tasks. The results demonstrated that the training strategy of SMI-Editor is a more effective approach.
>
> **Please let us know if we have adequately addressed your concerns, and we are always open to further suggestions.**
>
> Thank you for your time and consideration.
>
> Authors

---

> ### Author Response · Authors · 2024-11-23
> **Looking Forward to Your Valuable Feedback**
>
> We greatly appreciate your time and effort in reviewing our paper, as your feedback is incredibly valuable in helping us improve the quality of our paper and also crucial to the evaluation process. We have carefully addressed all the concerns and comments you raised. As the deadline for this discussion is approaching, we kindly request you to prioritize submitting your feedback at your earliest convenience. Thank you again for your valuable suggestions. We look forward to any further discussions.

---

> ### Author Response · Authors · 2024-11-24
> **Eagerly seeking more discussion. We need your valuable feedback !**
>
> Thank you for providing us with valuable suggestions. Considering that only **two days** remain until the end of the discussion period and we have yet to receive any feedback, we would like to confirm once again whether our response has addressed all your concerns. We welcome any further discussion.
>
> As one of the reviewers for this year's ICLR conference as well, I deeply understand that reading others' papers and providing suggestions is a time-consuming and arduous task. Therefore, I sincerely thank every reviewer who thoughtfully provided feedback on my paper; it is your suggestion that helped make my work better. At the same time, as one of the authors submitting to this conference, I also deeply understand the hard work and dedication that every author puts into preparing their manuscripts and responding to each question raised by the reviewers. Therefore, I will take responsibility and actively engage with the author discussions for every paper I review.  **The paper you have read and the responses we have provided are the result of our team members' utmost effort, dedication, and hard work. It holds great significance for us, and we simply hope to be treated fairly.**
>
> Thanks for your understanding. Wishing you all the best !

---

> ### Author Response · Authors · 2024-11-25
> **Thanks for your reply!**
>
> Thank you for reading our rebuttal and for your supportive words! We're happy that we have addressed most of your concerns! We would like to once again thank you for your comments, which are super inspiring and have indeed helped greatly enhance our paper. Your response truly means a lot to us, and we warmly welcome any further communication.
>
> Authors

---

### Official Review · Reviewer_SiA5 · 2024-10-30

**Soundness:** 3
**Presentation:** 3
**Contribution:** 3
**Rating:** 8
**Confidence:** 4

**Summary:**

This paper proposes an edit-based pre-trained SMILES language model to learn 3D molecular representations and applies the model to several downstream tasks, such as molecular property prediction. The idea of using token edit operations to replace token masking introduces a novel approach with the potential to address the rapid saturation problem. The paper provides comprehensive details on model implementation and includes extensive ablation studies.

**Strengths:**

Novelty: The use of token edit operations instead of traditional token masking is innovative and could offer significant advantages in terms of model performance and stability.
Comprehensive Implementation: The paper provides detailed descriptions of the model architecture and training procedures.
Extensive Ablation Studies: The authors have conducted thorough ablation studies to validate the effectiveness of their approach.

**Weaknesses:**

1. Figure 4: The requirement for an expert to provide training signals is inefficient and limits the size of the training set. This could be a significant drawback in practical applications.
2. Figure 5: Figures 3 and 5 should be presented together for easier comparison. Additionally, the figure's purpose is not clearly explained.
3. Figure 6: The figure's ability to demonstrate scalability is questionable. It would be helpful to show whether larger models yield better results to substantiate claims of scalability.
4. Conclusion: The statement, "ablation studies confirm the advantages of its design over traditional MLMs in modeling molecular substructure semantics and training stability," is not fully supported by the evidence presented in Figure 5. The figure does not convincingly demonstrate that the model understands molecular substructure semantics.

**Questions:**

1. Figure 3: The purpose of this figure is unclear. It would be beneficial to include a comparison with other methods to highlight the advantages of the proposed approach.
2. Table 1: It is important to specify the amount of data used for fine-tuning to better understand the model's performance.
3. Generative Tasks: It would be interesting to explore whether this model can be applied to generative tasks, which could broaden its applicability.

---

> ### Author Response · Authors · 2024-11-19
> **Response to Reviewer SiA5 (Part 1)**
>
> We express our gratitude to Reviewer SiA5 for valuable suggestions. In the following, we will address all your concerns regarding training efficiency, further explanation of  analytical experiments, model's scalability, the amount of data used for fine-tuning and application on generative tasks. And we have also supplemented many new experimental results (statistics and analysis of training time cost for SMI-Editor, more detailed analytical experiments, the preformance of SMI-Editor with different scales, the statistics of datasets in downsteam tasks, performance of SMI-Editor on retrosynthesis prediction task. ) to address some of the issues you have raised. All the new results have been updated in the revised paper. We hope the replies can make our paper more clear. Any further comments are welcome!
>
> ---
>
> ### Reponse to: The Figure 6 does not convincingly demonstrate scalability. Showing whether larger models yield better results would substantiate claims about scalability.
>
> **A:** Thank you for this valuable suggestion. We have added results showing the performance of SMI-Editor and SMI-MLM models of varying sizes on downstream tasks, which further demonstrate SMI-Editor’s strong scalability. These results, included in the revised paper, are shown in Table D1. It is evident that while increasing model size has minimal impact on MLM models, larger SMI-Editor models show more consistent performance gains. **This confirms the claim that SMI-Editor has better scalability compared to MLM models.** We have also added these resuts to Appendix O in the revised paper.
>
>
>
> Table D1. Performance of SMI-Editor and SMI-MLM models of varying sizes on downstream tasks.
>
> |                   | BACE↑    | BBBP↑    | SIDER↑   | Tox21↑   | ToxCast↑ | Mean↑    |
> | ----------------- | -------- | -------- | -------- | -------- | -------- | -------- |
> | SMI-MLM (Small)   | 76.8     | 69.6     | 60.5     | 75.3     | 64.2     | 69.2     |
> | SMI-MLM (Base)    | 76.6     | 69.3     | 59.9     | 75.3     | 64.4     | 69.1     |
> | SMI-MLM (Big)     | 77.4     | 68.7     | 60.8     | 75.1     | 65.3     | 69.4     |
> | SMI-Editor (Small)| 78.3     | 72.6     | 59.4     | 75.6     | 65.1     | 70.2     |
> | SMI-Editor (Base) | 79.2     | 73.2     | **61.0** | 75.7     | 65.8     | 71.0     |
> | SMI-Editor (Big)  | **79.3** | **74.2** | 60.9     | **76.7** | **66.4** | **71.5** |

---

> > ### Comment · Reviewer_SiA5 · 2024-11-26
> >
> > Thanks for your response which clarifies some of my concerns. I will keep my score.

---

> > > ### Author Response · Authors · 2024-11-26
> > > **Thanks for your reply!**
> > >
> > > Thank you very much for your reply and your valuable suggestions regarding this paper. Your thoughtful suggestions mean a lot to us. If you have any further questions, please feel free to reach out to us. Once again, we all sincerely appreciate your time and effort.

---

> ### Author Response · Authors · 2024-11-19
> **Response to Reviewer SiA5 (Part 2)**
>
> ### Reponse to: The requirement for experts to provide training signals is inefficient and limits the training set size.
>
> **A:** Thank you for raising this insightful concern. The use of experts does increase training costs. We measured that the training cost of SMI-Editor is approximately three times that of MLM models (SMI-MLM) for the same model size, training hyperparameters, and data. However, **the training cost of SMI-Editor remains acceptable**. To better analyze the impact of training cost, we trained an MLM model with equivalent computational cost (SMI-MLM(More)). Results showed that SMI-MLM(More) performed worse than the original SMI-MLM and significantly lagged behind SMI-Editor, highlighting that merely increasing MLM training cost does not yield better results. We have also added these resuts to Appendix P in the revised paper. Below is a detailed analysis:
>
> 1. **Training cost comparison:**
>    Table D2 compares the training costs (measured in GPU hours) of SMI-Editor and SMI-MLM models with the same size and hyperparameters. SMI-Editor’s cost is approximately three times that of SMI-MLM.
>
> 2. **Reasons for higher training cost in SMI-Editor:**
>    SMI-Editor requires computing expert actions (using a computationally expensive dynamic programming algorithm) and modeling three different editing operations, which introduces additional overhead.
>
> 3. **Acceptable training cost:**
>    Training SMI-Editor on a dataset with 19M compounds using four RTX 3090 GPUs took approximately 24.6 hours. Scaling SMI-Editor to larger datasets (e.g., 100M+ compounds) is feasible, demonstrating its potential for broader applications.
>
> 4. **SMI-Editor performs better under the same training cost with MLM:**
>    We trained SMI-MLM(More) with the same computational cost as SMI-Editor by increasing its training steps from 120K to 360K. Table D3 shows that SMI-MLM(More) performs worse than the SMI-Editor and  original SMI-MLM . This is due to rapid saturation issues in MLM training on SMILES data. **This also indicates that the speed of model training is not the most important factor; what matters more is whether the model can efficiently extract high-quality semantic representations**. This highlights the importance of designing more powerful training schemes like SMI-Editor to effectively extract meaningful information from SMILES.
>
> 5. **Higher performance ceiling for SMI-Editor:**
>    Although the inclusion of Experts slows down the training speed of the SMI-Editor model, it also enriches the semantic information the model learns. This gives SMI-Editor greater scalability and a higher performance ceiling compared to SMI-MLM. As shown in Table D1 in part 1, SMI-Editor benefits more from increased model size and training cost. This makes SMI-Editor a better choice when given the same and enough training budget.
>
>
>
> Table D2. Training time comparison (RTX 3090 GPUs) between SMI-Editor and SMI-MLM.
>
> |              | SMI-MLM | SMI-Editor |
> | ------------ | ------- | ---------- |
> | GPU Time (h) | 33.2    | 98.35      |
>
> Table D3. Performance of MLM models with equivalent training costs (SMI-MLM(More)).
>
> |               | BACE↑    | BBBP↑    | SIDER↑   | Tox21↑   | ToxCast↑ | Mean↑    |
> | ------------- | -------- | -------- | -------- | -------- | -------- | -------- |
> | SMI-MLM(More) | 74.3     | 66.2     | 49.5     | 73.3     | 62.3     | 65.1     |
> | SMI-MLM       | 77.8     | 68.6     | 61.2     | 75.1     | 64.9     | 69.5     |
> | SMI-Editor    | **80.3** | **77.4** | **63.0** | **77.1** | **67.4** | **73.0** |

---

> ### Author Response · Authors · 2024-11-19
> **Response to Reviewer SiA5 (Part 3)**
>
> ### Reponse to: The purpose of Figure 3 and Figure 5 (Figure 4 in the revised draft) is unclear. Figure 5 does not convincingly demonstrate the model’s understanding of molecular substructure semantics.
>
> **A:** Thank you for your feedback. We have further supplemented the relevant information for Figures 3 and 5, providing additional explanation and analysis on the chemical implications of the datasets involved and the model behaviors. We also included a comparison with auto-regressive LM models to further illustrate the advantages of the SMI-Editor model in modeling molecular substructures. We have also added these resuts to Appendix Q in the revised paper. Below are the specific results and analyses:
>
> - **Conclusion first:** **The observed trends for the FreeSolv dataset are fully consistent with our expectations and align with the definition of its physical properties**. On the other hand, the performance on the ESOL dataset is influenced by additional factors such as molecular weight. We also designed more analytical experiments to further investigate the behavior of the SMI-Editor model, and **the results demonstrate that the model’s behavior aligns with expectations**. Detailed explanations are as follows.
>
>   - **For the FreeSolv dataset, the observed trends align with its physical property definitions.** FreeSolv reflects the hydration free energy of compounds, defined as the free energy change when a compound transitions from a non-dissolved state to a dissolved state. When hydrophilic groups in a molecule are reduced, the change in hydration free energy increases, leading to higher hydration free energy. Thus, when we remove hydrophilic groups from the molecule, the model predicts an increase in hydration free energy, consistent with the trend observed in Figure 5(b), which matches our expectations.
>
>   - **For the ESOL task, the model predictions are significantly influenced by molecular weight.** The ESOL dataset reflects compound solubility, which is strongly negatively correlated with molecular weight: the larger the molecular weight, the lower the solubility. We plotted a scatter diagram (figure line: [here](https://anonymous.4open.science/r/ICLR25-21E2/esol_vis.pdf), or the Figure 14 in our revised draft) showing the relationship between molecular weight and solubility in the ESOL training set. A clear negative correlation with a coefficient of \( R = -0.69 \) is observed. Consequently, when functional groups or atoms are removed from a molecule, its molecular weight decreases, leading the model to predict an increase in solubility. **This explains why, in Figure 5(a), the model predicts increased solubility regardless of whether hydrophilic groups or random groups are removed**. The increase is more significant with random deletions, demonstrating the model's ability to distinguish between hydrophilic group deletions and random deletions.
>
>   - **To eliminate the influence of molecular weight, we designed a hydrophilic group replacement scheme (HG Rep).** We replaced all hydrophilic groups in a molecule with non-hydrophilic groups of similar molecular weight (e.g., methyl, ethyl, propyl) and compared this hydrophilic group replacement scheme (HG Rep) with a random group replacement scheme (Rand Rep), where random groups were replaced with others of similar weight. The results, shown in Figure 15 in our revised draft (or used links to the figure: [SMI-Editor](https://anonymous.4open.science/r/ICLR25-21E2/SMI_Editor_ESOL_Rep.pdf), [SMI-MLM](https://anonymous.4open.science/r/ICLR25-21E2/MLM_ESOL_Rep.pdf)), reveal that **SMI-Editor effectively distinguishes between HG Rep and Rand Rep, demonstrating its ability to model key molecular group semantics.** It also correctly predicts that replacing hydrophilic groups reduces molecular solubility.
>
>
>
>   Furthermore, we plotted the distribution of predicted changes for MLM models and Auto-regressive language models (Auto-regressive LM) before and after these replacement operations (link to the figure: [here](https://anonymous.4open.science/r/ICLR25-21E2/AR_ESOL_Rep.pdf), or Figure 16 in our revised draft). **The results show that these models perform significantly worse than the SMI-Editor in distinguishing between random replacements and hydrophilic group replacements.** This further highlights the superiority of the SMI-Editor in modeling the semantics of molecular substructures.
>
>
>
>   Once again, we sincerely thank you for raising this issue, which has allowed us to more thoroughly explain the model behaviors and our analytical methods!
>
> ---
>
> ### Reponse to: Figures 3 and 5 should be presented together for easier comparison.
>
> **A:** Thank you very much for your suggestion. We have reorganized the structure of the paper and placed Figures 3 and 5 together to facilitate comparison for the readers in our revised draft.

---

> ### Author Response · Authors · 2024-11-19
> **Response to Reviewer SiA5 (Part 4)**
>
> ### Reponse to: Can the model be applied to generative tasks? Exploring this could broaden its applicability.
>
> **A:** Thank you very much for your suggestion. In fact, we are also highly interested in exploring the potential of SMI-Editor in generative tasks. We believe that this edit-based pretraining model is well-suited for such tasks. Considering that the original SMI-Editor is an encoder-only model and cannot be directly applied to generative tasks, we further pretrained a model based on an encoder-decoder architecture, referred to as **SMI-Editor-Gen**. **We tested its performance on the retrosynthesis prediction task, where it achieved state-of-the-art results**. We have also added these resuts to Appendix R in the revised paper. Below is a detailed discussion:
>
> 1. **Model Details of SMI-Editor-Gen**:
>    SMI-Editor-Gen adopts a transformer architecture with a base-sized scale [2] and the specific model size details are provided in Table D5. During pretraining, the input to the encoder consists of SMILES strings with missing molecular fragments, while the decoder’s pretraining task is to reconstruct the original SMILES. Following approaches commonly used in machine translation [2], the features extracted by the encoder are passed to the decoder through encoder-decoder attention [2]. Compared to SMI-Editor, **the most significant difference is that the encoder-decoder architecture enables SMI-Editor-Gen to perform sequence-to-sequence generative tasks**, allowing us to explore the model’s capabilities in such tasks.
>
> 2. **SMI-Editor-Gen Exhibits Strong Performance in Retrosynthesis Prediction Tasks**:
>    Following the experimental setup of EditRetro [1], we evaluated SMI-Editor-Gen on the retrosynthesis prediction task. During fine-tuning, we applied the same fine-tuning strategies and data augmentation techniques as EditRetro (hyperparameters are detailed in Table D5). The experimental results, shown in Table D4, demonstrate that SMI-Editor-Gen achieved strong performance on the USPTO-50K dataset (more details on the dataset are provided in the supplementary materials). This validates that the pretraining approach proposed by SMI-Editor also exhibits excellent performance and great potential in generative tasks.
>
>
>
> P.S. Details of USPTO-50K dataset: USPTO-50K is a high-quality dataset comprising approximately 50,000 chemical reactions extracted from U.S. patent literature. These reactions feature precise atom-to-atom mappings, clearly defining the correspondence between reactants and products. The dataset is categorized into 10 distinct reaction types, enabling detailed analysis and comparison with other existing methods. Widely used in prior research, this dataset is ideal for benchmarking our proposed method against current state-of-the-art approaches.  For our experiments on the USPTO-50K dataset, we adhere to the well-established data split scheme reported by Coley et al. Specifically, the dataset is divided into training, validation, and test sets comprising 40,000, 5,000, and 5,000 samples, respectively. This consistent division ensures comparability with previous studies and allows for a rigorous evaluation of our method. For the evaluation metric, we utilize the top-k exact match accuracy as our primary evaluation metric. This metric provides a rigorous assessment by comparing the canonical SMILES of the predicted reactants to the ground truth reactants in the test dataset.
>
>
>
> Table D4. Top-k exact match accuracy of SMI-Editor and baselines on USPTO-50k dataset.
>
> |             | Top-1↑    | Top-3↑    | Top-5↑    | Top-10↑   |
> | ----------- | --------- | --------- | --------- | --------- |
> | RetroPrime  | 51.4%     | 70.8%     | 74.0%     | 76.1%     |
> | Transformer | 42.4%     | 58.6%     | 63.8%     | 67.7%     |
> | SCROP       | 43.7%     | 60.0%     | 65.2%     | 68.7%     |
> | MEGAN       | 48.1%     | 70.7%     | 78.4%     | 86.1%     |
> | GTA         | 51.1%     | 67.6%     | 74.8%     | 81.6%     |
> | Retroformer | 53.2%     | 71.1%     | 76.6%     | 82.1%     |
> | Graph2Edits | 55.1%     | 77.3%     | 83.4%     | 89.4%     |
> | R-SMILE     | 56.3%     | 79.2%     | 86.2%     | **91.0%** |
> | EditRetro   | 60.8%     | 80.6%     | 86.0%     | 90.3%     |
> | SMI-Editor  | **61.2%** | **80.9%** | **86.4%** | 89.7%     |
>
>
>
> Table D5. Hyperparameters of SMI-Editor on USPTO-50k task.
>
> | Hyperparameter       | Value  |
> | -------------------- | ------ |
> | Learning Rate        | 0.0001 |
> | Max Tokens           | 16384  |
> | Max Updates          | 100000 |
> | Num of Augmentations | 20     |
> | Warmup Steps         | 4000   |
>
>
>
> [1] Han, Yuqiang, et al. "Retrosynthesis prediction with an iterative string editing model." *Nature Communications* 15.1 (2024): 6404.
>
> [2] Vaswani, A. "Attention is all you need." *Advances in Neural Information Processing Systems* (2017).

---

> ### Author Response · Authors · 2024-11-19
> **Response to Reviewer SiA5 (Part 5)**
>
> ### Reponse to: It is important to specify the amount of data used for fine-tuning.
>
> **A:** Thank you for your suggestion. We have provided the dataset size information for each task in Table 1 (as shown in Table D6) and added more detailed explanations for each dataset (as shown in Table D7). This information has been added to Appendix H in the paper. Additionally, we have enhanced the title of Table 1 to help readers better understand this section of information. Thank you again for your valuable feedback!
>
> Table D6. Dataset sizes used for fine-tuning.
>
> | Dataset | Train | Valid | Test | Tasks |
> | ------- | ----- | ----- | ---- | ----- |
> | BACE    | 1210  | 151   | 152  | 1     |
> | BBBP    | 1631  | 204   | 204  | 1     |
> | Tox21   | 6264  | 783   | 784  | 12    |
> | SIDER   | 1141  | 143   | 143  | 27    |
> | MUV     | 74469 | 9309  | 9309 | 17    |
> | ClinTox | 1182  | 148   | 148  | 2     |
> | ToxCast | 6860  | 858   | 858  | 617   |
>
> Table D7. Descriptions of downstream datasets.
>
> | Dataset | Describe                                         |
> | ------- | :----------------------------------------------- |
> | BACE    | Binding results of human BACE-1 inhibitors       |
> | BBBP    | Blood-brain barrier penetration                  |
> | Tox21   | Qualitative toxicity measurements                |
> | SIDER   | Adverse drug reactions to the 27 systemic organs |
> | MUV     | A subset of PubChem BioAssay                     |
> | ClinTox | Clinical trial toxicity and FDA approval status  |
> | ToxCast | Toxicology data based on in vitro screening      |

---

> ### Author Response · Authors · 2024-11-21
> **Seeking more discussion with Reviewer SiA5**
>
> Dear Reviewer SiA5,
>
> We are grateful for your insightful feedback, which has been instrumental in improving our manuscript. We have also provided an additional summary here to further facilitate our discussion.
>
> First, in response to your concerns regarding the scalability of the model, we provide a comparison of the performance of SMI-Editor and SMI-MLM models of different sizes on downstream tasks. The results demonstrate that larger-scale SMI-Editor models achieve better performance, with a more pronounced improvement in effectiveness as the model scale increases.
>
> To address the issue of training efficiency, we compare the training time costs of the SMI-Editor and SMI-MLM models under equivalent training settings. Additionally, we present the performance of both models on downstream tasks with equivalent computational costs.
>
> Regarding the issues raised about Figures 3 and 5 in the paper, we have reorganized the manuscript to place these two figures together for easier comparison. Further detailed analyses have been added to explain the consistency between the SMI-Editor model's performance on these two types of data and the underlying physical principles.
>
> In response to your suggestion about applying the model to generative tasks, we trained an Encoder-Decoder-based SMI-Editor model and applied it to molecular retrosynthesis prediction tasks. The model's performance demonstrates the significant potential of the SMI-Editor pre-training strategy for generative tasks.
>
> Finally, concerning the question of downstream task data volume, we have supplemented the paper with detailed statistics on the downstream task datasets and emphasized this information in the revised manuscript.
>
> **Please let us know if we have adequately addressed your concerns, and we are always open to further suggestions.**
>
> Thank you for your time and consideration.
>
> Authors

---

> ### Author Response · Authors · 2024-11-24
> **Eagerly seeking more discussion. We need your valuable feedback !**
>
> Thank you for providing us with valuable suggestions. Considering that only **two days** remain until the end of the discussion period and we have yet to receive any feedback, we would like to confirm once again whether our response has addressed all your concerns. We welcome any further discussion.
>
> As one of the reviewers for this year's ICLR conference as well, I deeply understand that reading others' papers and providing suggestions is a time-consuming and arduous task. Therefore, I sincerely thank every reviewer who thoughtfully provided feedback on my paper; it is your suggestion that helped make my work better. At the same time, as one of the authors submitting to this conference, I also deeply understand the hard work and dedication that every author puts into preparing their manuscripts and responding to each question raised by the reviewers. Therefore, I will take responsibility and actively engage with the author discussions for every paper I review.  **The paper you have read and the responses we have provided are the result of our team members' utmost effort, dedication, and hard work. It holds great significance for us, and we simply hope to be treated fairly.**
>
> Thanks for your understanding. Wishing you all the best !

---

### Official Review · Reviewer_mXJn · 2024-11-04

**Soundness:** 3
**Presentation:** 3
**Contribution:** 3
**Rating:** 8
**Confidence:** 4

**Summary:**

The paper first investigates the problem of MLM training, as train-inference mismatch. To address this, the authors propose the edit-based pre-trained language model, SMI-Editor, for SMILES. As another pre-training strategy, SMI-Editor drops the substructures in a valid SMILES sequence and tries to recover the original molecules.. The extensive experiments demonstrate the effectiveness of SMI-Editor.

**Strengths:**

1. The model is well-motivated to solve the train-inference mismatch problem in existing MLM paradigm.

2. Extensive experiments indicate the effectiveness of SMI-Editor.

3. The analysis of the rapid saturation problem of MLM and how SMI-Editor can solve this problem is generally convincing.

4. The paper is well-written and easy to follow.

**Weaknesses:**

1. The authors put a lot efforts to compared SMI-Editor with MLM paradigm. However, there are still other strategies for pre-training the model, for example, contrastive learning. Can the authors also discuss the relations between SMI-Editor and other pre-training paradigms?

2. As the authors state, increasing the mask ratios will influence the convergence in MLM. Would fragment drop ratio also influences the SMI-Editor?

**Questions:**

See weaknesses.

---

> ### Author Response · Authors · 2024-11-19
> **Response to Reviewer mXJn (Part 1)**
>
> We express our gratitude to Reviewer mXJn for valuable suggestions. In the following sections, we will thoroughly address all your concerns regarding Comparison with other pre-training paradigm, and the impact of  fragment drop ratio on SMI-Editor model. And we have also supplemented new experimental results (performance of other pre-training paradigm, the training curves of SMI-Editor with different fragment drop ratioand  the performance of SMI-Editor with different fragment drop ratio) to address some of the issues you have raised. All the new results have been updated in the revised paper. We welcome any further comments or feedback you may have.
>
> ---
>
> ### Response to: Can the authors elaborate on the relationships between SMI-Editor and these alternative paradigms?
>
> **A:** Thank you for your suggestion. In the revised draft, we have included more comparisons and discussions of different pretraining strategies, including contrastive learning and autoregressive language modeling in Appendix U and Appendix J of the revised draft. Below is a detailed discussion of their similarities and differences:
>
> Comparison between **SMI-Editor and contrastive learning**:
>
> - **Similarities:** Both **contrastive learning and SMI-Editor aim to learn alignment.**
>
>   - **Contrastive learning aligns representations of different views:** The core idea of contrastive learning is to bring the representations of different views of the same sample (positive pairs) closer while pushing representations of different samples (negative pairs) apart. Essentially, this process learns the correct alignment between views of the same sample.
>
>   - **SMI-Editor aligns representations of missing substructures and contexts:** As Fu et al. [1] noted, MLM models align the representations of contexts and missing words during training. Similarly, SMI-Editor aligns the representations of missing substructures and their contexts. For example, given the input `Nc1ccc(O)cc1`, the model need to predict the complete molecule `CC(=O)Nc1ccc(O)cc1`. SMI-Editor can effectively align the representation of the missing fragment `CC(=O)` with the context `Nc1ccc(O)cc1` through this process.
>
> - **Differences:** The alignment targets differ between the two paradigms.
>
>   - **Contrastive learning focuses on global information:** The representations to be aligned often correspond to different augmented views of the same molecule, such as through atom deletion, bond deletion, or subgraph deletion. These views typically preserve the molecule's overall structure and thus contain global information.
>
>   - **SMI-Editor emphasizes aligning local substructure information with global context:** In SMI-Editor, the context typically corresponds to the molecule's backbone, representing global information, while the missing substructures contain local information.
>
>   - **SMI-Editor is more sensitive to local structure information:** By aligning local substructures with global context, SMI-Editor learns finer-grained semantics from SMILES data, making it better suited to capturing detailed molecular information than contrastive learning.
>
> - **Performance comparison:**
>   In Table 1 of the main paper and Table A1 of the General Response, we compared SMI-Editor to MolCLR, a contrastive learning-based molecular representation model. SMI-Editor consistently outperforms MolCLR across most tasks, demonstrating its effectiveness.

---

> ### Author Response · Authors · 2024-11-19
> **Response to Reviewer mXJn (Part 2)**
>
> ### Reponse to: Would the fragment drop ratio similarly affect SMI-Editor?
>
> **A:** Thank you for the valuable suggestion. To investigate the impact of the fragment drop ratio on SMI-Editor, we trained SMI-Editor models with different drop ratios (15%, 30%, 45%) and analyzed their training curves and downstream task performance. The results indicate that increasing the drop ratio significantly raises training loss for SMI-Editor, suggesting that its pretraining task is more challenging than MLM. We have also added these results to Appendix L in the revised paper. Below are the detailed findings:
>
> 1. **Impact on SMI-Editor's convergence**:
>
>    - We plotted the training and validation loss curves for SMI-Editor with varying drop ratios (figure link: [train loss](https://anonymous.4open.science/r/ICLR25-21E2/DropRatio_Train_Loss.pdf), [valid loss](https://anonymous.4open.science/r/ICLR25-21E2/DropRatio_Valid_Loss.pdf), also shown in Figure 12 of the revised paper). **The results show that as the drop ratio increases, both training and validation losses rise significantly.** Compared to Figure 2(C) of the paper, the loss increase for SMI-Editor is more pronounced than for MLM, confirming that SMI-Editor's task is inherently more challenging.
>
> 2. **Impact on downstream task performance**:
>
>    - We evaluated the performance of SMI-Editor and MLM models with varying drop or mask ratios. The results are summarized in Table C1. From Table C1, it can be observed that as the mask ratio increases, the average performance of the SMI-MLM model shows no significant change, while the performance of the SMI-Editor model declines as the drop ratio increases. This indicates that SMI-Editor represents a more challenging training task.
>    - **Explanation:**
>      - SMI-Editor discards chemically meaningful substructures that often serve as standalone semantic units. This also makes predicting the discarded fragments more difficult than predicting individual masked tokens. **Dropping more substructures severely disrupts the molecular structure, making it harder for the model to reconstruct the original molecule.**
>      - MLM, on the other hand, randomly masks tokens in SMILES sequences. Since SMILES tokens often represent individual atoms or bonds, masking does not typically disrupt the molecular semantics significantly. For instance, masking one or two atoms of a functional group like `-COOH` still leaves enough contextual information to reconstruct it. Additionally, the probability of masking an entire functional group is low due to MLM's token-based masking mechanism. This explains why MLM performance is less sensitive to mask ratio increases, as also reflected in Figure 2(C) of the paper: **"Different Mask Ratios Cannot Alleviate Rapid Saturation."**
>
>
>
> Table C1. Performance of SMI-Editor and MLM with different drop or mask ratios on downstream tasks.
>
> |                 | BACE↑    | BBBP↑    | SIDER↑   | Tox21↑   | ToxCast↑ | Mean↑    |
> | --------------- | -------- | -------- | -------- | -------- | -------- | -------- |
> | SMI-MLM(15%)    | 77.8     | 68.6     | 61.2     | 75.1     | 64.9     | 69.5     |
> | SMI-MLM(30%)    | 78.3     | 70.2     | 58.2     | 76.0     | 63.7     | 69.3     |
> | SMI-MLM(45%)    | 78.4     | 66.1     | 59.3     | 76.4     | 65.5     | 69.1     |
> | SMI-Editor(15%) | 80.3     | **77.4** | **63.0** | 77.1     | **67.4** | **73.0** |
> | SMI-Editor(30%) | **81.6** | 73.3     | 59.6     | 77.0     | 66.8     | 71.7     |
> | SMI-Editor(45%) | 79.3     | 72.2     | 61.1     | **77.8** | 67.1     | 71.5     |

---

> ### Author Response · Authors · 2024-11-19
> **Response to Reviewer mXJn (Part 3)**
>
> Comparison between **SMI-Editor and autoregressive language models**:
>
> - **SMI-Editor (encoder-only) vs. Decoder-only autoregressive models:**
>   Autoregressive models, which use decoder-only architectures, are another common pretraining approach for molecules. We trained a model, SMI-GPT, using the autoregressive objective and the same architecture and size as SMI-Editor. The performance comparison is presented in Table A2 of the General Response. SMI-Editor outperforms SMI-GPT across tasks, showing that the encoder-only architecture is better suited for representation learning.
>
> - **Challenges with autoregressive models:**
>   Autoregressive models receive all preceding tokens as context when generating new tokens. For tokens at later positions, this rich context simplifies the prediction task, making the prediction of these tokens overly simple for the model. **This phenomenon leads to rapid convergence problem and limits the model's ability to extract richer information from the data**. A key difference between SMI-Editor and autoregressive models is that in SMI-Editor, each discarded token is predicted independently, with equal importance assigned to the prediction of each token. This enables SMI-Editor to better capture the complete semantic information encoded in the dropped tokens.
>
> **Summary**:  Compared to other pretraining strategies, SMI-Editor excels in representation learning, mitigating rapid saturation issues, and extracting richer substructure semantics from SMILES. The extensive experimental results further validate the advantages of SMI-Editor's pretraining approach.
>
> [1] Fu, Zhiyi, et al. "Contextual representation learning beyond masked language modeling." *arXiv preprint arXiv:2204.04163* (2022).

---

> ### Comment · Reviewer_mXJn · 2024-11-19
> **Thank you for your rebuttal**
>
> I went through the rebuttal from the authors and other reviews. The response has totally addressed my concerns. I believe this paper has conducted solid and comprehensive experiments demonstrating the effectiveness of SMI-Editor. Although I may have some concerns that the idea of SMI-Editor is somewhat straightforward, I am still willing to raise my score to 8.

---

> > ### Author Response · Authors · 2024-11-19
> > **Thank you for your timely reply !**
> >
> > Thank you very much for your prompt reply! It means a lot to us, and we warmly welcome any further communication in the future. Once again, thank you for your response and support!

---

### Official Review · Reviewer_uWiK · 2024-11-05

**Soundness:** 2
**Presentation:** 2
**Contribution:** 3
**Rating:** 5
**Confidence:** 3

**Summary:**

This paper identifies three key challenges in existing SMILES masked language models: the neglect of substructural information, overly simplistic training tasks, and a mismatch between training and inference procedures and introduce a SMILES language model employing edit-based, fragment-level supervision to address these challenges.

**Strengths:**

1.	This paper reveals the shortcomings of existing SMILES masked language models through an analysis of experimental results, thereby informing future research directions.
2.	This paper presents a novel SMILES language model that uses edit-based, fragment-level supervision. This approach improves performance on the molecule property prediction task.

**Weaknesses:**

1.	Edit-based models, as currently designed, focus solely on restoring removed substructures during pre-training. Other valuable sources of molecular information, such as correcting errors and removing extraneous components, may be neglected.
2.	The data processing and pre-training stages lack clarity. A detailed case study would significantly improve understanding of these processes.
3.	The description of the experimental tasks is inconsistent between the caption of Table 1 and the " 5.2 RESULTS ON MOLECULAR PROPERTY PREDICTION" section.

**Questions:**

1.	How to ensure that left-over fragments can be assembled into a valid molecule? Please provide an exact example.

---

> ### Author Response · Authors · 2024-11-19
> **Response to Reviewer uWiK (Part 1)**
>
> We thank reviewer uWiK for the helpful suggestions. In the following, we will address all your concerns regarding pre-training strategy, paper's clarity and inconsistent in the table's title. And we have also supplemented new experimental results (new pre-training strategy, examples and codes used to illustrate the data processing procedure. ) to address some of the issues you have raised. All the new results have been updated in the revised paper. We hope the replies can make our paper more clear. Any further comments are welcome!
>
> ---
>
> ### **Response to**: Other valuable sources of molecular information, such as correcting errors and removing extraneous components, may be overlooked.
>
> **A:** Thank you for your insightful suggestion. In the revised paper, we have included more discussions and comparisons regarding different pretraining strategies in Appendix K. In fact, we previously attempted to train the model to correct errors and remove extraneous components. **However, this approach did not lead to performance improvement**. Below are the detailed results and discussions:
>
> 1. **Training SMI-Editor to correct errors and remove extraneous components did not improve performance:**
>    We implemented a version of SMI-Editor that learns to correct erroneous functional groups and remove extraneous substructures, referred to as SMI-Editor-Cor. However, SMI-Editor-Cor did not outperform the original SMI-Editor on downstream tasks. Considering the increased complexity and training cost of SMI-Editor-Cor (due to longer input sequences), we focused on SMI-Editor in the submitted draft. Table B1 below compares the performance of SMI-Editor and SMI-Editor-Cor, showing that their performance is similar, demonstrating the limited benefit of incorporating these tasks.
>
> 2. **Analysis of SMI-Editor-Cor’s performance:**
>    We attribute SMI-Editor-Cor's lack of improvement to the following reasons:
>    - **Correcting errors and removing extraneous components provide limited additional training signals:**
>      SMI-Editor’s training comprises two major steps: deletion and insertion. During deletion, erroneous functional groups and extraneous substructures are removed, while the insertion step involves learning to recover the correct tokens in the appropriate positions. Thus adding erroneous functional groups or extraneous substructures affects only the deletion step, which is a simpler task providing limited information. Moreover, as shown in Table 3 of the main text, ablating the token deletion (TokDel) step has minimal performance impact.
>    - **Identifying erroneous functional groups and extraneous structures is too simple for the model:**
>      SMI-Editor-Cor constructs erroneous inputs through random substitutions, often resulting in chemically invalid SMILES that are easy for the model to identify. Consequently, the simplicity of the training task limits further performance improvement.
>
> 3. **Future directions for improvement:**
>    The results from SMI-Editor-Cor inspire us to explore chemical-aware SMILES corruption methods to construct chemically plausible erroneous inputs. **We are currently working on the next generation of SMI-Editor, which leverages real chemical reactions from databases to transform molecules into similar but chemically altered forms**. This model predicts how to restore the original molecule. However, implementing this version of the model is significantly more challenging as it requires defining additional chemical rules. Currently, we are still in the process of collecting the necessary data for this aspect.
>
>
>
> *Implementation details of SMI-Editor-Cor:*
>
> We first fragmente molecules in the training dataset using a BRICKS-based approach (splitting at BRICS bonds and detaching side chains). We then matche fragments based on their connectivity interfaces, identifying interchangeable fragments (e.g., `-CH3` and `-COOH` share a single-bond carbon interface). During training, fragments are randomly replaced with compatible fragments to generate corrupted molecules for the model.
>
>
>
> Table B1: Performance comparison between SMI-Editor-Cor and SMI-Editor.
>
> |                | BACE↑ | BBBP↑ | SIDER↑ | Tox21↑ | ToxCast↑ | Mean↑ |
> | -------------- | ----- | ----- | ------ | ------ | -------- | ----- |
> | SMI-Editor-Cor | 80.6  | 77.1  | 62.2   | 76.8   | 68.0     | 72.9  |
> | SMI-Editor     | 80.3  | 77.4  | 63.0   | 77.1   | 67.4     | 73.0  |

---

> > ### Comment · Reviewer_uWiK · 2024-11-25
> > **Thanks for the rebuttal**
> >
> > For the second and the third points, the authors have give responses to solve my concerns. However, for the first point, there may leave some spaces. I am not fully convinced about the explanations. These tasks though easy but not means they can not contribute. However, I just leave question here.
> > Thanks for the authors rebuttal.

---

> > > ### Author Response · Authors · 2024-11-25
> > > **Thanks for your reply!**
> > >
> > > Thank you very much for your response. In fact, we are also very interested in exploring how to improve model performance by enabling it to learn to identify incorrect groups in molecules. Since the deletion operation accounts for a relatively small portion of the learning in SMI-Editor (as shown in Figure 7 of the paper, the loss value for the TokDel process is significantly lower than the losses for the other two editing operations), and TokDel, compared to the other two training tasks, is a simpler binary classification task, the introduction of incorrect groups to enhance the model's recognition capability provides only limited additional training signals to TokDel training process. This, in turn, restricts the model's potential for further performance improvement (you could also understand this as an unbalanced problem among different tasks in a multi-task learning process).
> > >
> > > As mentioned earlier, in future work, we will further explore how to design more effective training signals for incorrect group identification to enhance the model's capability. Additionally, we aim to introduce more training signals of such editing operations to enrich the knowledge learned during the TokDel process. We deeply appreciate your constructive suggestions and welcome any further discussions.

---

> > > ### Author Response · Authors · 2024-11-26
> > > **Thanks for your reply and suggestions!**
> > >
> > > We look forward to your response and hope that the newly provided information and these discussions can help you reassess our paper, which is of great importance to us. Once again, thank you for your valuable suggestions, and we look forward to more communication in the future.

---

> ### Author Response · Authors · 2024-11-19
> **Response to Reviewer uWiK (Part 2)**
>
> ### Response to: The description of experimental tasks is inconsistent between the caption of Table 1 and the section's title.
>
> **A:** Thank you for pointing this out. The original title of Table 1 was meant to indicate that it belongs to molecular classification tasks. To improve accuracy, we have revised the title to: “The overall results on 7 molecular property classification datasets” and also update the section titles to the corresponding names in our revised draft. Additionally, we have included results for another type of task—molecular property regression tasks—in Table B2 (or  in Appendix M of the revised draft). These results demonstrate that SMI-Editor also performs well in regression tasks.
>
>
>
> Table B2: Results on molecular property regression tasks.
>
> |            | ESOL↓     | FreeSolv↓ | Lipo↓     |
> | ---------- | --------- | --------- | --------- |
> | MPNN       | 0.58      | 1.150     | 0.7190    |
> | DMP_{TF}   | 0.700     | -         | -         |
> | A-FP       | 0.503     | 0.736     | 0.578     |
> | SMI-MLM    | 0.576     | 0.709     | 0.642     |
> | SMI-Editor | **0.362** | **0.524** | **0.565** |
>
>
> ---
>
>
> ### Response to: How can the leftover fragments be assembled into a valid molecule? Please provide an exact example.
>
> **A:** Thank you for the question. In fact, we do not require the reassembled molecules to be chemically valid, as the corruption process is designed to disrupt the molecule's original structure, allowing the model to learn how to restore it. As described in Section 4.2 of the paper, “The resulting **corrupted molecule** is converted back into a SMILES representation and fed into the SMILES Encoder.” While the reassembled molecule is disrupted, it is represented as a valid SMILES sequence according to SMILES syntax rules.
>
> Here is a detailed example explaining this process, accompanied by an illustrative flowchart to provide a clear visual explanation (**figure link**: [here](https://anonymous.4open.science/r/ICLR25-21E2/molecule_frag_case.pdf), or the Figure 13 in our revised draft). **Additionally, we have provided executable code for you to explore and test this process directly** (**code link**: [here](https://anonymous.4open.science/r/ICLR25-21E2/molecule_frag.py)). We have also added these resuts to Appendix N in the revised paper.

---

> ### Author Response · Authors · 2024-11-19
> **Response to Reviewer uWiK (Part 3)**
>
> **Example Workflow with Paracetamol (SMILES: `CC(=O)Nc1ccc(O)cc1`):**
>
> 1. **Structure Overview:**
>    - An aromatic benzene ring with a hydroxyl group (`-OH`).
>    - An amide group (`-NH-C(=O)-CH₃`) attached to the benzene ring.
>
> 2. **Fragmentation Process:**
>    - Break the **amide bond** connecting the carbonyl carbon (`C=O`) and the nitrogen atom (`-NH-`).
>    - Detach the hydroxyl group from the benzene ring (break `-OH`).
>
> Resulting Fragments:
> - Fragment 1: `[1*]C(C)=O`
> - Fragment 2: `[*:1]N[*:2]`
> - Fragment 3: `[*:2]c1ccc([*:3])cc1`
> - Fragment 4: `[*:3]O`
>
> Here, `[*:n]` represents dummy atom labels used to record the topological information of the original molecule. Identical dummy atom labels across different fragments indicate that these positions were previously connected by chemical bonds. Additionally, we record the type of chemical bond corresponding to each dummy atom. This information enables us to correctly reassemble the molecule later.
>
> 3. **Reassembly Scenarios:**
>
>    Depending on the specific discarded fragment, the reassembled molecule will differ. Here, we demonstrate the reassembly process using fragments 1–4 as examples of discarded fragments.
>
>
>
>    **Case 1**: Discarding Fragment 1 `[1*]C(C)=O`
>
>    *Reassembly Process*: In this case, the remaining molecular fragments are still connected, with no isolated fragments. Therefore, we simply reconnect the fragments based on their dummy atom labels by forming chemical bonds between matching dummy atoms. Specifically, `[*:1]N[*:2]` and `[*:2]c1ccc([*:3])cc1` share the same dummy atom `[*:2]`, so these two fragments are connected.
>
>    *Result*:  `Nc1ccc(O)cc1`
>
>
>
>    **Case 2**: Discarding Fragment 2 `[*:1]N[*:2]`
>
>    *Reassembly Process*: Here, the molecule becomes disconnected. Following the rule of reconnecting fragments based on matching dummy atoms, the original molecule is split into two fragments: `[1*]C(C)=O` and `[2*]c1ccc(O)cc1`. These fragments are then reassembled in the order they originally appeared in the molecule, resulting in the final structure.
>
>    *Result*:  `C(C)=O.c1ccc(O)cc1`
>
>
>
>    **Case 3**: Discarding Fragment 3 `[*:2]c1ccc([*:3])cc1`
>
>    *Reassembly Process*: Similar to Case 2, the remaining molecular fragments are disconnected. The two fragments are reassembled in the order they originally appeared in the molecule, producing the final structure.
>
>    *Result*:  `NC(C)=O.O`
>
>
>
>    **Case 4**: Discarding Fragment 4 `[*:3]O`
>
>    *Reassembly Process*:  As in Case 1, the remaining molecular fragments remain connected. The molecule is reassembled by reconnecting the fragments based on their dummy atom labels.
>
>    *Result*:  `c1ccc(NC(C)=O)cc1`
>
>
>
> 4. **Final SMILES Representation:**
>    - The reassembled molecule is sanitized and converted to a canonical SMILES representation. Unmatched dummy atoms are removed to ensure validity.
>
> **Key Measures to Ensure Reassembly:**
>
> 1. **Avoid overly large fragments:** After fragmenting the molecule using the BRICKS algorithm, we further break the bonds between side chains and the rings they are attached to. This ensures that the resulting molecular fragments are not excessively large. By doing so, we can avoid discarding overly large molecular fragments during the subsequent fragment discarding process, preventing significant damage to the core structure of the molecule and facilitating the reassembly process.
> 2. **Track original bond information:** During fragmentation, bond connections and types are recorded, enabling correct reassembly based on this information.
>
> This ensures the correct reassembly of the left-over fragments. The canonical SMILES representation of the reassembled molecule is then input into the model, which edits it (via deletion or insertion) to restore the original SMILES information. This constitutes the core process of the pretraining workflow.

---

> ### Author Response · Authors · 2024-11-21
> **Seeking more discussion with Reviewer uWiK**
>
> Dear Reviewer uWiK,
>
> We are grateful for your insightful feedback, which has been instrumental in improving our manuscript. We have also provided an additional summary here to further facilitate our discussion.
>
> First, we present the performance results of the SMI-Editor model, which is trained using a strategy based on correcting errors and removing extraneous components, on downstream tasks to analyze the effectiveness of this pre-training strategy. A detailed analysis of these results and plans for future improvements are also provided.
>
> Second, regarding your suggestion about the title of Table 1, we have updated the titles of both Table 1 and Section 5.2 to enhance their consistency and accuracy of expression.
>
> Finally, for the pre-training and data processing procedures, we have provided a detailed case study and created a flowchart to visually illustrate the process. Additionally, we included an example code for molecular fragmentation and reassembly to further clarify the process.
>
> Thank you again for your valuable suggestions. We look forward to further discussions.
>
> **Please let us know if we have adequately addressed your concerns, and we are always open to further suggestions.**
>
> Thank you for your time and consideration.
>
> Authors

---

> ### Author Response · Authors · 2024-11-23
> **Looking Forward to Your Valuable Feedback**
>
> We greatly appreciate your time and effort in reviewing our paper, as your feedback is incredibly valuable in helping us improve the quality of our paper and also crucial to the evaluation process. We have carefully addressed all the concerns and comments you raised. As the deadline for this discussion is approaching, we kindly request you to prioritize submitting your feedback at your earliest convenience. Thank you again for your valuable suggestions. We look forward to any further discussions.

---

> ### Author Response · Authors · 2024-11-24
> **Eagerly seeking more discussion. We need your valuable feedback !**
>
> Thank you for providing us with valuable suggestions. Considering that only **two days** remain until the end of the discussion period and we have yet to receive any feedback, we would like to confirm once again whether our response has addressed all your concerns. We welcome any further discussion.
>
> As one of the reviewers for this year's ICLR conference as well, I deeply understand that reading others' papers and providing suggestions is a time-consuming and arduous task. Therefore, I sincerely thank every reviewer who thoughtfully provided feedback on my paper; it is your suggestion that helped make my work better. At the same time, as one of the authors submitting to this conference, I also deeply understand the hard work and dedication that every author puts into preparing their manuscripts and responding to each question raised by the reviewers. Therefore, I will take responsibility and actively engage with the author discussions for every paper I review.  **The paper you have read and the responses we have provided are the result of our team members' utmost effort, dedication, and hard work. It holds great significance for us, and we simply hope to be treated fairly.**
>
> Thanks for your understanding. Wishing you all the best !

---

> ### Author Response · Authors · 2024-11-29
> **New analysis of SMI-Editor-Cor and looking forward to further discussion!**
>
> Dear Reviewer uWiK,
>
> Hi, thank you for taking the time to provide feedback on our manuscript. We deeply appreciate your insights, and we have tried our best to address your concerns in our response. We have also provided some new results of SMI-Editor-Cor here to further facilitate our discussion.
>
> To further analyze the reasons behind the performance of SMI-Editor-Cor, we examined the average norm of the gradients for three different editing operations during the training process. The results are shown in the table below.
>
> Table B3: Gradient analysis of SMI-Editor-Cor.
> |           | Token Deletion | Placeholder Insertion | Token Prediction |
> | --------- | -------------- | --------------------- | ---------------- |
> | Grad Norm | 0.02215     | 0.09265            | 0.12070       |
>
> From the table, it can be observed that during the training process, the gradient norm for token deletion loss is significantly smaller than that of the other two operations. This finding is consistent with the conclusion we presented in Figure 7 of the paper, where we discussed the scale of the losses. This result also suggests that, since identifying and removing the incorrect parts in SMILES is straightforward and the SMI-Editor-Cor model trains the three editing operations simultaneously, token deletion has the smallest loss and gradient norm among them. Consequently, when we attempt to have the model distinguish erroneous functional groups during the token deletion step, the amount of knowledge provided to the model is limited. We will also include these new results in the final version of the paper.
>
> Once again, we appreciate your valuable feedback, which has supported the improvement of our paper. If you have any further questions, please feel free to ask. **Additionally, if our response has helped address your previous concerns, we would appreciate it if you could reconsider the paper's score. Your support is very very crucial to us.** Thank you!

---

> ### Author Response · Authors · 2024-12-03
> **Looking forward to Further Discussion**
>
> Dear Reviewer uWiK,
>
> Thank you so much for your constructive feedback! In our previous response, we have made every effort to address all the concerns you raised regarding the performance of SMI-Editor-Cor, providing the gradient analysis during the training process for further analysis of the model's behavior. Considering that there are only 12 hours left before the discussion ends, if you have any further questions, please feel free to raise them. We look forward to any further discussions to assist you in making a more comprehensive evaluation of this paper.
>
> Best,
>
> Authors

---

### Author Response · Authors · 2024-11-19
**General Reponses (Part 1)**

# General Response (Part 1)

We would like to express our sincere gratitude to all the reviewers for dedicating their time to thoroughly evaluate our work. We are delighted to receive valuable suggestions from the reviewers. We appreciate **uWiK** and **SiA5** for acknowledging the innovation of using edit-based, fragment-level supervision in SMI-Editor; **mXJn** and **Rnmw** for recognizing the well-motivated design and insightful analysis of the train-inference mismatch and rapid saturation problems; **SiA5** and **AGu3** for acknowledging the detailed implementation and significant performance improvements of SMI-Editor on molecular property prediction and valid SMILES generation; and **mXJn** and **Rnmw** for commending the clarity and readability of the paper. We sincerely thank all the reviewers for their insightful comments and constructive feedback.

**We have also carefully addressed all concerns and questions raised by each reviewer.** Incorporating the feedback, we have conducted additional experiments to strengthen our explanations and support our claims. All new results, analyses, and revisions have been included in the updated version of the paper. In addition, all new images, example codes, and other files supplemented during the rebuttal stage have been packaged and uploaded to the supplementary materials. They can be downloaded and viewed directly if needed. **If further clarifications are needed, please do not hesitate to let us know.**

Before addressing specific questions and concerns, we would like to share two significant new experimental findings that may interest the reviewers:

P.S: Due to the inclusion of a substantial number of new results during this rebuttal stage, we have labeled all tables in the rebuttal as Table A, Table B, etc., to avoid conflicts with the original paper's numbering (e.g., Table 1, Table 2).


## Part 1:Excellent Performance of SMI-Editor on DeepChem Data

We re-evaluated the performance of SMI-Editor on various downstream tasks of MoleculeNet benchmark using the data splits provided by DeepChem. Previously, our experiments were based on a different data split, which made it difficult to compare our model against others built on this dataset. Therefore, we re-tested SMI-Editor on DeepChem splits and included comparisons with more baseline models. Detailed results are presented in Table A1. As shown in Table A1, **SMI-Editor achieves significant performance gains over baseline models, reaching state-of-the-art levels with noticeable average performance improvements.** We have also added these resuts to Appendix I in the revised paper. Below is a detailed analysis of these results:

1. **SMI-Editor outperforms models trained with various paradigms:** On average, SMI-Editor surpasses molecular representation learning models like MolCLR and DMP_{TF}, which use contrastive pretraining, as well as models like ChemBerta and SMI-MLM, which use masked language modeling. It also outperforms autoregressive language models like Galactica and graph-based models like MolCLR, MGSSL, and MoMu. These results highlight the potential of SMILES language models.

2. **SMI-Editor achieves competitive performance with less training data:** SMI-Editor outperforms DMP_{TF}, which is trained on over 100 million compounds, despite using only 19 million compounds for training. This demonstrates SMI-Editor's higher data efficiency, enabled by its ability to effectively leverage substructure information from SMILES sequences.

Table A1. Overall results on MoleculeNet datasets using DeepChem splits ([DeepChem GitHub](https://github.com/deepchem/deepchem)). ROC-AUC scores (higher is better) are reported for all tasks. The best results are **bolded**.

|            | BBBP↑    | Tox21↑   | ClinTox↑ | HIV↑     | BACE↑    | SIDER↑   | Mean↑    |
| ---------- | -------- | -------- | -------- | -------- | -------- | -------- | -------- |
| GEM        | 72.4     | 78.1     | 90.1     | 80.6     | 85.6     | 67.2     | 79.0     |
| ChemBerta  | 64.3     | -        | 90.6     | 62.2     | -        | -        | -        |
| MolCLR     | 73.6     | 79.8     | 93.2     | 80.6     | 89.0     | 68.0     | 80.7     |
| MGSSL      | 70.5     | 76.5     | 80.7     | 79.5     | 79.7     | 61.8     | 74.8     |
| DMP_{TF}   | 78.1     | 78.8     | 95.0     | 81.0     | 89.3     | 69.2     | 81.9     |
| Galactica  | 66.1     | 68.9     | 82.6     | 74.5     | 61.7     | 63.2     | 69.5     |
| MoMu       | 70.5     | 75.6     | 79.9     | 76.2     | 77.1     | 60.5     | 73.3     |
| SMI-MLM    | 89.4     | 76.2     | 90.6     | 79.8     | 86.6     | 66.5     | 81.5     |
| SMI-Editor | **93.5** | **81.4** | **95.2** | **81.6** | **89.9** | **69.8** | **85.2** |

---

> ### Author Response · Authors · 2024-11-19
> **General Response (Part 2)**
>
> ## Part 2:Performance Advantages of SMI-Editor Over Auto-regressive Models
>
> To comprehensively compare SMI-Editor with autoregressive models, we trained a decoder-only model with identical architecture and size to SMI-Editor using an autoregressive language modeling objective, referred to as SMI-GPT. We evaluated SMI-GPT's performance across several downstream tasks, with results shown in Table A2. **The findings indicate that SMI-Editor can perform better than SMI-GPT**. We have also added these resuts to Appendix J in the revised paper. Below is an analysis of these results:
>
> 1. **Implementation details for SMI-GPT(NT) and SMI-GPT(Emb):**
> To more comprehensively compare performance, we implemented two methods of applying the SMI-GPT model to downstream molecular property classification tasks.
>    - **SMI-GPT(NT):** This approach uses next-token prediction for downstream classification tasks by appending a special token (e.g., `$Label_{0}$`, `$Label_{1}$`) at the end of each SMILES sequence to denote the classes of sample's label. The model learns to predict the correct label token during fine-tuning.
>    - **SMI-GPT(Emb):** The representations of each token in the SMILES string extracted by the SMI-GPT model are processed using mean pooling. The resulting pooled representation is then fed into a classification head, which predicts the class of the SMILES.
>
> 2. **Advantages of the encoder-only SMI-Editor architecture:**
>    - As shown in Table A2, SMI-Editor consistently outperforms SMI-GPT(Emb) and SMI-GPT(NT), highlighting its superior semantic learning capabilities.
>    - SMI-GPT(Emb) achieves better performance than SMI-GPT(NT), suggesting that pretraining-based feature transfer is preferable for molecular property prediction tasks. Therefore, the encoder-only pre-trained model is highly suitable for molecular property prediction tasks.
>
> 3. **Rapid convergence in autoregressive LMs:**
>    We provide the training curve of the SMI-GPT model (figures: [training loss](https://anonymous.4open.science/r/ICLR25-21E2/gpt_loss.svg), [training ppl](https://anonymous.4open.science/r/ICLR25-21E2/gpt_ppl.svg), or the Figure 11 in our revised draft), which shows that the loss decreases rapidly during the early stages of training. Similarly, the perplexity also drops quickly, reaching approximately 1.6 at the 40K training step. By the end of training, the model's Perplexity falls below 1.6, which is significantly lower than the perplexity typically observed for GPT models trained on text data.
>
> 4. **Why does this phenomenon occur?**
>
>    For auto-regressive language models, each time a new token is generated, it receives all preceding tokens as prefix input. This means that when the model generates tokens at later positions, it has access to more comprehensive contextual information (i.e., a longer prefix and more complete sequence information). As a result, **the prediction difficulty for tokens in later positions is significantly reduced, allowing the model to converge more easily**. A key difference between SMI-Editor and SMI-GPT is that in SMI-Editor, each discarded token is predicted independently, with equal importance assigned to the prediction of each token. This enables SMI-Editor to better capture the complete semantic information encoded in the tokens.
>
> In summary, compared to LLMs on text data, GPT models on SMILES data converge significantly faster and achieve much lower perplexity. This indicates that SMILES data is inherently easier to fit than text. Therefore, it is crucial to design effective methods to extract richer semantic information from SMILES. **SMI-Editor represents a meaningful and successful exploration in this direction, highlighting the importance of leveraging substructural fragment information within SMILES data.**
>
>
>
> Table A2. Results of SMI-Editor and SMI-GPT on MoleculeNet datasets using DeepChem splits. ROC-AUC scores (higher is better) are reported for all tasks. The best results are **bolded**.
>
> |              | BBBP↑    | Tox21↑   | ClinTox↑ | HIV↑     | BACE↑    | SIDER↑   | Mean↑    |
> | ------------ | -------- | -------- | -------- | -------- | -------- | -------- | -------- |
> | SMI-GPT(NT)  | 88.5     | 74.3     | 88.9     | 68.8     | 76.2     | 63.7     | 76.7     |
> | SMI-GPT(Emb) | 91.2     | 75.1     | 91.4     | 79.4     | 86.2     | 67.1     | 81.7     |
> | SMI-MLM      | 89.4     | 76.2     | 90.6     | 79.8     | 86.6     | 66.5     | 81.5     |
> | SMI-Editor   | **93.5** | **81.4** | **95.2** | **81.6** | **89.9** | **69.8** | **85.2** |

---

### Meta-Review · Area_Chair_Zfab · 2024-12-20

**Metareview:**

This paper introduces SMI-Editor, an edit-based pre-trained SMILES language model aimed at improving molecular representation learning. The authors propose a novel pre-training strategy, which involves randomly corrupting SMILES strings by dropping substructures and training the model to restore the original molecules, providing fragment-level supervision and better leveraging molecular substructure semantics.

Overall, this paper is well motivated and well written. By using edit-based, fragment-level supervision instead of traditional token masking, SMI-Editor effectively tackles the rapid saturation and train-inference mismatch problems. The authors provide the comprehensive experiments to validate the effectiveness of the proposed pre-training strategy.

Meanwhile, some concerns were raised regarding the clarity of data processing and pre-training stages, suggesting that detailed explanations and case studies would enhance understanding.  Additionally,  the literature review of this paper is not sufficient. It dose not include the broader pre-training strategies of molecules, including molecular graph pre-training and contrastive pre-training approaches.

Minor issue: at Line 82, RELATD should be RELATED.

In a summary,  The technical novelty is somehow limited. Extending the mask strategy to the fragment-level is a minor improvement. But I still recommend the acceptance of this paper based on its comprehensive experiments.

**Additional Comments On Reviewer Discussion:**

The AC has organized  the internal discussions about this paper. After several round discussions, the reviewers decided to keep their original score.

---

### Decision · Program_Chairs · 2025-01-22

Accept (Poster)